# Finite Sample Analyses for Continuous-time Linear Systems: System Identification and Online Control

**Hongyi Zhou**[*]
IIIS, Tsinghua University
Shanghai Qizhi Institute
zhouhong24@mails.tsinghua.edu.cn

**Jingwei Li**[*]
IIIS, Tsinghua University
Shanghai Qizhi Institute
ljw22@mails.tsinghua.edu.cn

**Jingzhao Zhang**[†]
IIIS, Tsinghua University
Shanghai Qi zhi Institute
jingzhaoz@mail.tsinghua.edu.cn

## Abstract

Real world evolves in continuous time but computations are done from finite samples. Therefore, we study algorithms using finite observations in continuous-time linear dynamical systems. We first study the system identification problem, and propose a first non-asymptotic error analysis with finite observations. Our algorithm identifies system parameters without needing integrated observations over certain time intervals, making it more practical for real-world applications. Further we propose a lower bound result that shows our estimator is provably optimal up to constant factors. Moreover, we apply the above algorithm to online control regret analysis for continuous-time linear system. Our system identification method allows us to explore more efficiently, enabling the swift detection of ineffective policies. We achieve a regret of $\mathcal{O}(\sqrt{T})$ over a single $T$-time horizon in a controllable system, requiring only $\mathcal{O}(T)$ observations of the system.

## 1 Introduction

Finding optimal control policies requires accurately modelling the system [18]. However, real-world environments often involve unknown system parameters. In such cases, estimating unknown parameters from exploration becomes essential to identify the unseen dynamics. This process is recognized as system identification, a fundamental tool employed in various research fields, including time-series analysis [20], control theory [21], robotics [16], and reinforcement learning [28].

The identification of linear systems has long been studied because linear systems, as one of the most fundamental systems in both theoretical frameworks and practical applications, has wide applications ranging from natural physical processes to robotics. Most classical results provide only *asymptotic* convergence guarantees for parameter estimation [3, 24, 5].

On the other hand, with the rapid increase in data scale, there is a growing concern for statistical efficiency. Consequently, the non-asymptotic convergence of *discrete-time* linear system identification has emerged as another pivotal topic in this field. Investigations into this matter delve into understanding how estimation confidence is influenced by the sample complexity of trajectories [9], or the running time on a single trajectory [33, 29]. Furthermore, many of these studies operate

---

[*]Equal Contribution
[†]Corresponding Author

39th Conference on Neural Information Processing Systems (NeurIPS 2025).

under the common assumption of stochastic noise, there has been a parallel exploration into the identification of discrete-time linear dynamical systems with diverse setups. This includes scenarios where perturbations are adversarial [15] or when only black-box access is available [7].

In contrast to studies in discrete time system, there have been relatively fewer non-asymptotic results addressing parameter identification for *continuous-time systems*. Two problems exist for continuous time analysis. First, nonasymptotic analysis in continuous system without noise can be degenerate, as a short time interval can contain infinite pieces of information. Second, if we consider the non-degenerate case when finite noisy observations are available, then the analyses require concentration results that become known only as in [33, 9, 29]. Recently [4] provides novel analyses for estimating system parameters, which relies on continuous data collection and interaction with the environment. Motivated by progress in these works, our first goal is to answer the question below:

> *Can we design a continuous-time stochastic system identification algorithm that provides nonasymptotic error bounds with only a finite number of samples?*

We will introduce our system identification algorithms tailored to meet the above requirements. As expected, we discretize time into small intervals, thereby reducing the problem to a discrete system. The interesting part involves ensuring that the discretization remains bijective and that the inversion is unbiased. Our algorithm identifies the continuous system using only a finite number of samples from the discrete system. We further propose a information theoretic lower bound that shows our algorithm is optimal.

As an application of our system identification methods, we study an online continuous-time linear control problem as introduced in [30]. In this context, exploration is essential for estimating unknown parameters, with the goal of identifying a more optimal control policy that narrows the performance gap. The primary challenge involves finding the right balance between exploration and exploitation. Leveraging our identification method for more efficient parameter estimation allows us to effectively manage exploration and exploitation, achieving an expected regret of $\mathcal{O}(\sqrt{T})$ over a single trajectory with only $\mathcal{O}(T)$ samples in time horizon $T$. This surpasses the previously best known result of $\mathcal{O}(\sqrt{T}\log(T))$, which needs continuous data collection from the system.

We summarize our contributions below.

1. When the system can be stabilized by a known controller, we establish an algorithm with $\mathcal{O}(T)$ samples that achieves estimation error $\mathcal{O}(\sqrt{1/T})$ on a single trajectory with running time $T$, which is shown in Theorem 2. We also provide Theorem 4 which shows that the estimation error of our system identification method is optimal up to constant factors.

2. When a stable controller is not available, we can use $N$ independent short trajectories to obtain estimators with error $\mathcal{O}(\sqrt{1/N})$, as is shown in Theorem 5 .

3. We apply our system identification method to an online continuous linear control algorithm, which only requires $\mathcal{O}(T)$ samples and achieves $\mathcal{O}(\sqrt{T})$ regret on a single trajectory with lasting time $T$ (Theorem 6), improving upon the best known result $\mathcal{O}(\sqrt{T}\log(T))$ in [30].

## 2 Related Works

Control of both discrete and continuous linear dynamical systems have been extensively studied in various settings, such as linear quadratic optimal control [27], $H_2$ stochastic control [10], $H_\infty$ robust control [34, 17] and system identification [21, 24]. Below we introduce some of the important results on both system identification and optimal control for linear dynamical systems.

**System Identification**    Earlier literature focused primarily on the asymptotic convergence of system identification [6, 25]. Recently, there has been a resurgence of interest in non-asymptotic system identification for *discrete-time* systems. [9] studied the sample complexity of multiple trajectories, with $\mathcal{O}(\sqrt{1/N})$ estimation error on $N$ independent trajectories. For systems with dynamics $x_{t+1} = Ax_t + w_t$(without controllers), [33] established an analysis for $\mathcal{O}(\sqrt{1/T})$ estimation error on a single stable trajectory with running time $T$, while [13] and [29] extended to more general discrete-time systems.

Non-asymptotic analyses for continuous-time linear system are less studied. Recently, [4] examined continuous-time linear quadratic control systems with standard brown noise and unknown system dynamics. Our algorithm is specifically designed for finite observations, achieving an error rate that cannot be attained through the direct discretization of integrals as done in [4].

**Regret Analysis of Online Control**   In online control, if the system's parameters are known, achieving the optimal control policy in this setup can be straightforward [34, 35]. However, when the system parameters are unknown, identifying the system incurs regret. [1] achieved an $\mathcal{O}(\sqrt{T})$ regret for discrete-time online linear control, which has been proven optimal in $T$ under that setting in [32]. Subsequent works have extended this setup, focusing on worst-case analysis with adversarial noise and cost, including [26, 8, 22, 32, 2]. These analyses are limited to discrete systems. For continuous-time systems, works of [31, 30, 23] established algorithms for online continuous control that achieves $\mathcal{O}\left(\sqrt{T}\log(T)\right)$ regret.

# 3   Problem Setups and Notations

In this section, we introduce the background and notation for linear dynamical systems and online control.

## 3.1   Linear Dynamical Systems

We first introduce discrete-time linear dynamical systems as follows: Let $x_k \in \mathbb{R}^d$ represent the state of the system at time $k$, and let $u_k \in \mathbb{R}^p$ denote the action at time $k$. Then, for some linear time-invariant dynamics characterized by $A \in \mathbb{R}^{d \times d}$ and $B \in \mathbb{R}^{d \times p}$, the transition of the system to the next state can be represented as:

$$x_{k+1} = Ax_k + Bu_k + w_k, \tag{1}$$

where $w_k \in \mathbb{R}^d$ are i.i.d. Gaussian random vectors with zero means and certain covariance.

Similarly, a continuous-time linear dynamical system with stochastic disturbance at time $t$ is defined by a differential equation, instead of a recurrence relation:

$$dX_t = AX_t dt + BU_t dt + dW_t. \tag{2}$$

In this context, we use $X_t$ and $U_t$ to represent the state and action in the continuous-time linear system, distinguishing them from $x_t$ and $u_t$ in discrete-time systems. $W_t$ denotes the stochastic noise, which is modeled by standard Brownian motion.

For a continuous control problem, an important question of a linear dynamical system is whether such system can be stably controlled. Below we define the concepts of stable dynamics and stabilizers.

**Definition 1.** For any square matrix $A$, define $\alpha(A) = \max_i \{\Re(\lambda_i) | \lambda_i \in \lambda(A)\}$, where $\Re(\lambda)$ represents the real part of complex number $\lambda$, $\lambda(A)$ is the set of all eigenvalues of $A$.

**Definition 2.** A matrix $A \in \mathbb{R}^{d \times d}$ is stable if $\alpha(A) < 0$. A control matrix $K \in \mathbb{R}^{p \times d}$ is said to be a stabilizer for system $(A, B)$ if $A + BK$ is stable.

Under the above definition, a stable dynamic guarantees that the state can automatically go to the origin when no external forces are added, while applying a stabilizer as the dynamic for controller will also ensure that the state does not diverge.

## 3.2   Continuous-time LQR Problems and Optimal Control

For continuous-time linear systems disturbed by stochastic noise, as introduced in 3.1, we denote the strategy of applying control to such systems through a specific causal policy, $f : X \to U$. This policy maps states $X$ to control inputs $U$, where the policy at time $t$ can only depend on the states and actions prior to $t$.

The optimal controls in linear systems are often linear [34, 35], which takes the following form

$$U_t = K_t X_t,$$

where $K_t \in \mathbb{R}^{p \times d}$ represents the linear parameterization at time $t$ under some policy $f(X) = KX$. Additionally, we define the cost function of applying the action $U_t = K_t X_t$ with linear quadratic regulator (LQR) control. Given predefined symmetric positive definite matrices $Q \in \mathbb{R}^{d \times d}$ and $R \in \mathbb{R}^{p \times p}$, along with the initial state $X_0$, the cost during $t \in [0, T]$ is denoted by $J_T$, as represented in the following equation:

$$J_T = \mathbb{E}\left[\int_{t=0}^{T} \left(X_t^{\mathrm{T}} Q X_t + U_t^{\mathrm{T}} R U_t\right) dt\right]. \tag{3}$$

Here the expectation is taken over the randomness of $X_t$.

Among all the polices there exists an optimal mapping $f_*$ which minimizes $J_T$. When the system is dominated by dynamics $(A, B)$, with the state transits according to (2), such optimal $K_t$ can be computed via the Lyapunov matrix $P_t$ that solves the Ricatti differential equation [35]:

$$\frac{d}{dt} P_t = P_t^{\mathrm{T}} B R^{-1} B^{\mathrm{T}} P_t - A^{\mathrm{T}} P_t - P_t^{\mathrm{T}} A - Q, \quad P_T = 0. \tag{4}$$

Then, under $f_*$ the action dynamic is set to be $K_t = -R^{-1} B^{\mathrm{T}} P_t$.

When $T \to +\infty$, the starting dynamic $P_0$ converges to some special dynamic $P_*$ satisfying

$$P_*^{\mathrm{T}} B R^{-1} B^{\mathrm{T}} P_* - A^{\mathrm{T}} P_* - P_*^{\mathrm{T}} A - Q = 0, \tag{5}$$

and the optimal control policy for infinite time horizon is by setting $K_* = -R^{-1} B^{\mathrm{T}} P_*$ and apply the action by $U_t = K_* X_t$.

**Online Control Problems.** Online learning aims to find a strategy to output a sequence of controls $\{U_t\}$ that minimizes the cost $J_T$ without knowing the system parameters $A, B$. In this scenario, the algorithms explore to obtain valuable information, such as estimators $(\hat{A}, \hat{B})$ for $(A, B)$, while simultaneously exploit gathered information to avoid large instantaneous cost.

To quantify the progress in an online learning problem with horizon $T$, one quantity of interest is the regret $R_T$, which quantifies the performance gap between the control taken $U_t = f(X_t)$ and a baseline optimal policy which takes $U_t = K_* X_t = -R^{-1} B^{\mathrm{T}} P_* X_t$, where $K_*$ is defined in (5). Formally, by denoting $J_T$ be the expected cost under $f$, and $J_T^*$ be the expected cost under the baseline optimal policy, the regret $R_T$ is represented as:

$$R_T = J_T - J_T^*. \tag{6}$$

**Other Notations** Denote the d-dimensional unit sphere $\mathcal{S}^{d-1} = \{v \in \mathbb{R}^d, \|v\|_2 = 1\}$, where $\|\cdot\|_2$ is the $L_2$ norm. For any matrix $A \in \mathbb{R}^{m \times n}$, denote $\|A\|$ be the spectral norm of $A$, or equivalently,

$$\|A\| = \sup_{v \in \mathcal{S}^{n-1}} \|Av\|_2 = \sup_{u \in \mathcal{S}^{m-1}, v \in \mathcal{S}^{n-1}} u^{\mathrm{T}} A v.$$

## 4 The Proposed System Identification Method

In this section we propose our system identification method. Before presenting our method, we first introduce the formal definition of system identification and the finite observation setting.

### 4.1 System Identification and Finite Observation

We start with the definition of system identification.

**Definition 3** (System Identification). The system identification task aims to recover the true system dynamics matrices $A$ and $B$ by observing the system's response over time. Specifically, one selects a time horizon $T$ and a sequence of actions $U$, observes the resulting states $X$, and computes estimates $\hat{A}$ and $\hat{B}$ of the true dynamics. The goal is to design an effective algorithm that achieves the following non-asymptotic estimation bound:

$$\|\hat{A} - A\|, \|\hat{B} - B\| \leq f(T),$$

for some function $f$ depending on $T$. In particular, as $T \to \infty$, we expect the estimation error $f(T)$ to converge to zero.

Next, we introduce the *finite observation assumption*. Under this setting, the number of observed states $N$ grows at most linearly with the trajectory running time $T$. In other words, for any trajectory of length $T$, we can only access a finite set of states $\{X_1, X_2, \ldots, X_N\}$ to identify the system, where $N = O(T)$ and does not exhibit superlinear growth.

To analyze the continuous-time system, we need to discretize it. Prior works [4, 31, 30] commonly approximate the dynamics using

$$X_{t+h} \approx (I + hA)X_t + hBU_t + (W_{t+h} - W_t).$$

However, this approximation introduces a discretization error between the approximated and true dynamics. The error term, characterized by $(e^{hA} - I)/h - A$, is of order $O(h)$. Consequently, the sampling interval $h$ must be chosen as $O(1/\sqrt{T})$ to ensure that discretization error does not dominate. This leads to a super-linear sampling complexity of $m = T/h = \Omega(T^{3/2})$, which violates the finite observation assumption and significantly increases computational demands.

In contrast, our method overcomes this limitation by directly estimating the matrix exponential $e^{Ah}$ in Lemma 1, and subsequently recovering $(A, B)$ from this estimate. As a result, our approach avoids discretization error entirely, allowing the sampling interval to depend solely on system parameters rather than the total sampling time $T$. This innovation reduces the sampling complexity to grow linearly with $T$, offering significant computational advantages.

## 4.2 Algorithm Design

Then we introduce our algorithm. We choose a small sampling time interval $h$ across a single trajectory of time length $T$. We then divide the time into small intervals and consider the state evolution within each interval. We get the following Lemma:

**Lemma 1.** *In the time interval $[t, t + h]$, the following transition function holds:*

$$X_{t+h} = e^{Ah}X_t + \int_{s=0}^{h} e^{A(h-s)}BU_{t+s}ds + w_t\,,$$

Here, $w_t$ is Gaussian noise $\mathcal{N}(0, \Sigma)$ with covariance $\Sigma = \int_{s=0}^{h} e^{As}e^{A^{\mathrm{T}}s}ds$. The formal proof of this Lemma is deferred to the Appendix A.2.

This transition equation connects continuous-time and discrete-time systems. In our method, the whole trajectory is partitioned into intervals with proper determined length $h$. During time $t \in [kh, (k+1)h]$, we observe a state $x_k$ at time $t = kh$, and fix the action $U_t \equiv u_k$ in this interval. Denoting $A^{'} = e^{Ah}$ and $B^{'} = \left[\int_{s=0}^{h} e^{A(h-s)}ds\right]B$, then the set of observations $\{x_k|k = 0, 1, 2, ...\}$ and actions $\{u_k|k = 0, 1, 2, ...\}$ follow the standard discrete-time linear dynamical system:

$$x_{k+1} = A^{'}x_k + B^{'}u_k + w_k.$$

Then we can apply the system identification method of discrete-time system [33, 9]. However, different from classical discrete-time systems, continuous-time systems present new challenges. The crucial one is that knowing $e^{Ah}$ is not sufficient to determine $A$, because the matrix exponential function $f(X) = e^X$ is not one-to-one. This means we might obtain an incorrect estimator $\hat{A}$ by solving $e^{\hat{A}h} = M$, where $M$ is the estimate of $e^{Ah}$. From the above analysis, we introduce our assumptions of the algorithm.

**Assumption 1** (Assumptions for Algorithm 1 and Theorem 2). We assume

1. The linear dynamic $A$ is stable, with $\alpha(A) < 0$ (see Definition 1). This is equivalent to assuming the existence of a stable controller $K$ and then set $A \leftarrow A + BK$.

2. $\|A\| \leq \kappa_A$, $\|B\| \leq \kappa_B$ for some known $\kappa_A, \kappa_B$ ($\kappa_A, \kappa_B$ need not be closed to $\|A\|, \|B\|$).

3. The sample interval $h$ is chosen to be $h = \frac{1}{15\kappa_A}$.

With the above assumptions, we design our algorithm as described in Algorithm 1. In the $k$-th interval of length $h$, the state $x_k$ is observed at the beginning, and a randomly selected action $u_k$ is applied

---

**Algorithm 1** System identification algorithm for stable system

---

**Input:** Running time $T$, sample interval $h$ satisfying the condition in Assumption 1.
Define the number of samples $T_0 = \lceil T/h \rceil$.
**for** $k = 0, \ldots, T_0 - 1$ **do**
    Sample the action $u_k \overset{\text{i.i.d.}}{\sim} \mathcal{N}(0, I_p)$.
    Use the action $U_t \equiv u_k$ during the time period $t \in [kh, (k+1)h]$.
    Observe the new state $x_{k+1}$ at time $(k+1)h$.
**end for**
Compute system estimates $(\hat{A}, \hat{B})$ via (8).

---

uniformly throughout the interval. The state-action pair $x_k, u_k$ is then used to estimate the discretized dynamics via:

$$(\widetilde{A})^{\mathrm{T}} = \left[ \sum_{k=0}^{T_0-1} x_k x_k^{\mathrm{T}} \right]^{\dagger} \sum_{k=0}^{T_0-1} x_k x_{k+1}^{\mathrm{T}} , (\widetilde{B})^{\mathrm{T}} = \left[ \sum_{k=0}^{T_0-1} u_k u_k^{\mathrm{T}} \right]^{\dagger} \sum_{k=0}^{T_0-1} u_k \left( x_{k+1} - \widetilde{A} x_k \right)^{\mathrm{T}} . \quad (7)$$

The continuous-time dynamics $(A, B)$ are then recovered from $(\widetilde{A}, \widetilde{B})$. Under the condition $\|A\|h \ll 1$, we employ Taylor expansion to compute $\hat{A}h = \log(\widetilde{A})$, approximating $Ah$. The estimators $(\hat{A}, \hat{B})$ are given by:

$$\hat{A} = \frac{1}{h} \sum_{k \geq 1} \frac{(-1)^{k-1}}{k} (\widetilde{A} - I)^k, \hat{B} = \left[ \int_{t=0}^{h} e^{\hat{A}t} dt \right]^{-1} \widetilde{B} . \quad (8)$$

We now establish the efficiency of our algorithm and derive the main theorem as follows.

**Theorem 2** (Upper bound). *In Algorithm 1, there exists a constant $C \in poly\left(|\alpha(A)|^{-1}, \kappa_A, \kappa_B\right)$ such that, $\forall 0 < \delta < \frac{1}{2}$, when $T \geq C\left(\|X_0\|_2^2 + \log^2 1/\delta\right)$, with probability at least $1 - \delta$, we have:*

$$\|\hat{A} - A\|, \|\hat{B} - B\| \leq C \sqrt{\frac{\log(1/\delta)}{T}} . \quad (9)$$

We defer the proof of the theorem to Appendix A.4 and highlight the key idea below. The key idea of the proof is to analyze the error transformation from the discrete system to the original system. We prove Lemma 3, which shows that the errors in the discrete and original systems differ only by a constant factor. This allows us to focus solely on the discrete system identification problem.

**Lemma 3.** *In Algorithms 1, suppose we obtain the relative error $\|\widetilde{A} - A^{'}\|, \|\widetilde{B} - B^{'}\| \leq \epsilon$ for some $\epsilon \leq \frac{1}{15}$ and $\|Ah\| \leq \frac{1}{15}$. Then, the relative error in the original system satisfies:*

$$\|\hat{A} - A\|, \|\hat{B} - B\| \leq \frac{1}{h} \left( 2 + \frac{\kappa_B}{\kappa_A} \right) \epsilon . \quad (10)$$

From this lemma, it follows that if we develop a system identification algorithm for the discrete system that produces dynamics estimates $\widetilde{A}$ and $\widetilde{B}$ with minimal error, we can obtain accurate estimates for the original system. The remaining task is to analyze the discrete system with the transition function $x_{k+1} = A x_k + B u_k + w_k$, which has been discussed in previous works such as [33].

### 4.3 Lower Bound

In this section, we discuss the lower bound of the problem. We prove Theorem 4 and establish that this method has already attained the optimal convergence rate for parameter estimation. The theorem primarily asserts that, given a single trajectory lasting for time $T$, any algorithm that estimates system parameters solely based on *an arbitrarily large number of finite observed states* cannot guarantee an estimation error of $o(\sqrt{1/T})$.

**Theorem 4** (Lower bound). *Suppose $T \geq 1$ be the running time of a single trajectory of continuous-time linear differential system, represented as in (2). Then there exist constants $c_1, c_2$ independent*

*of $d$ such that, for any finite set of observed points $\{t_0 = 0, t_1, t_2, ..., t_N = T\}$, and any (possibly randomized) estimator function $\phi : \{X_{t_0}, X_{t_1}, ..., X_{t_N}\} \to \mathbb{R}^{d \times d}$, there exists system parameter $A, B$ satisfying $\mathbb{P}\left[\|\phi(\{X_{t_i}\}_{i \leq N}) - A\| \geq \frac{c_1}{\sqrt{T}}\right] \geq c_2$. Here the probability is with respect to noise.*

In Theorem 4, the mapping $\phi$ can refer to the output of any algorithm that exclusively relies on the finite set of states $X_{t_0}, X_{t_1}, ..., X_{t_N}$. *The interesting observation is that the lower bound does not decrease with a larger observation number $N$.*

We defer the proof of the theorem to the Appendix A.6 and provide a proof sketch below. We consider two sets of dynamics, $(A, 0)$ and $(\bar{A}, 0)$, where both $A$ and $\bar{A}$ are stable, and $|A - \bar{A}| = \frac{2c_1}{\sqrt{T}}$. Our key observation is that for the two distributions of observed states $S_k = \{X_{t_0}, X_{t_1}, ..., X_{t_k}\}$ and $\bar{S}_k = \{\bar{X}_{t_0}, X_{t_1}, ..., X_{t_k}\}$, where $X$ corresponds to the linear dynamic $A$ and $\bar{X}$ corresponds to $\bar{A}$, the KL divergence between $S_{k+1}$ and $\bar{S}_{k+1}$ increases by at most $\frac{c}{T}(t_{k+1} - t_k)$. Here, $c$ is a universal constant independent of $t_k$ and $t_{k+1}$. Thus, regardless of how the observation times are selected, the KL divergence between the observed states remains bounded.

**Remark 1** (The Discussion of Lower Bound). The construction in Theorem 4 involves matrices $A$ and $\bar{A}$ that depend on $T$, specifically with $\|A - \bar{A}\| = \frac{2c_1}{\sqrt{T}}$. One might be concerned that such a $T$-dependent construction lacks interpretability since the true system parameters are independent of $T$. However, as shown in Appendix A.6, the matrices are taken as $A = -I_d$ and $\bar{A} = -I_d - U$, where $U$ has only one nonzero entry at position $(1, 1)$ equal to $\frac{1}{5\sqrt{T}}$. For these matrices, the key constants in the upper bound remain uniformly bounded: the inverse stability margin $\frac{1}{|\alpha(A)|}$ equals 1 for $A = -I_d$ and is at most $\frac{1}{1 + \frac{1}{5\sqrt{T}}} \leq 1$ for $\bar{A}$; the condition number $\kappa(A)$ equals 1 for $A = -I_d$ and is at most $1 + \frac{1}{5\sqrt{T}} \leq 2$ for $\bar{A}$. Thus both quantities are controlled by universal constants, independent of $T$, ensuring that the lower and upper bounds are comparable up to a constant factor.

## 4.4 Finding an Initial Stable Controller

While previous work on continuous-time system identification [4, 30] always assumes a known stable controller, our method extends to cases where a stabilizer is not known in advance. For general $(A, B)$, where a stabilizer is not predetermined, relying on a single trajectory is not feasible, as the state may diverge rapidly before obtaining a stable controller is obtained. Instead, we first find a stable controller $K$ using multiple short-interval trajectories and then employ it in Algorithm 3 for online control. Below, we list the assumptions on system parameters.

**Assumption 2** (Assumptions for Algorithm 2 and Theorem 5). We assume

1. The constants $\kappa_A, \kappa_B, h$ follow the same assumptions as in 1.

2. The running time $T$ for each trajectory is small, say, $T = T_0 h$ where $T_0 \in \mathbb{N}$ and $T_0 \leq 10$.

Then, we employ multiple short trajectories to identify $A$ and $B$ as outlined in Algorithm 2. Similar to what is demonstrated in [9], this procedure results in an $\mathcal{O}(H^{-1/2})$ estimation error on the trajectory number $H$.

**Theorem 5.** *In Algorithm 2, there exists a constant $C \in poly(\kappa_A, \kappa_B)$ such that w.p. at least $1 - \delta$, the estimation error of $(\hat{A}, \hat{B})$ from $H$ trajectories satisfies:*

$$\|\hat{A} - A\|, \|\hat{B} - B\| \leq C\sqrt{\frac{\log(1/\delta)}{H}} .$$

The proof details are shown in the Appendix A.5.

## 5 A Continuous Online Control Algorithm with Improved Regret

In this section, we apply our system identification method to a continuous LQR online control algorithm. Recall the setup introduced in Section 3.2 where we want to minimize the regret $R_T$ defined in (6). We will show in this section that with $\mathcal{O}(T)$ samples, our algorithm achieves $\mathcal{O}(\sqrt{T})$

---

**Algorithm 2** Multi-trajectory system identification algorithm

---

**Input:** $T, T_0, h$ as in Assumption 2, number of trajectories $H$.
**for** $l = 1, \dots, H$ **do**
    **for** $k = 0, \dots, T_0 - 1$ **do**
        Sample the action $u_k^l \overset{\text{i.i.d.}}{\sim} \mathcal{N}(0, I_p)$, use the action $U_t \equiv u_k^l$ during $t \in [kh, (k+1)h]$.
        Observe the new state $x_{k+1}^l$ at time $(k+1)h$.
    **end for**
**end for**
Compute $(\widetilde{A}, \widetilde{B})$ by $(\widetilde{A}, \widetilde{B}) \in \arg\min_{(A,B)} \frac{1}{2} \sum_{l=1}^{H} \left\| x_{T_0}^l - A x_{T_0-1}^l - B u_{T_0-1}^l \right\|_2^2$.
Compute $\bar{A}, \bar{B}$ as in (8), let $(\hat{A}, \hat{B}) = (\bar{A}, \bar{B})$ be estimates for system dynamics $(A, B)$.

---

expected regret on a single trajectory, thereby improving upon the previous $\mathcal{O}\left(\sqrt{T} \log(T)\right)$ result. We list the assumption for the online LQR problems below.

**Assumption 3** (Assumptions for Algorithm 3 and Theorem 6). We assume that:

1. A stabilizer $K$ for $(A, B)$ (see Definition 2) with $\alpha(A + BK) < 0$ is known in advance.

2. Sample distance $h$ satisfies $h = \frac{1}{15\kappa}$, where $\kappa \geq \|A\| + \|B\|\|K\| \geq \|A + BK\|$ is known.

3. Denote $P_*$ be the solution in (5) and $K_* = -R^{-1}B^\mathrm{T}P_*$ be the baseline control dynamic.

4. $Q, R$ are positive-definite symmetric matrices with bounded spectral norms $\|Q\|, \|R\| \leq M$ and for some $\mu > 0$, $\mu I \preceq Q, \mu I \preceq R$.

## 5.1 An $\mathcal{O}(\sqrt{T})$ Regret Algorithm for Continuous Online Control

Our online continuous control algorithm is outlined in Algorithm 3, and we provide a detailed description below. Algorithm 3 is divided into two phases, exploration and exploitation. For the first exploration phase, a previously known stabilizer $K$ is applied to prevent the state from diverging. During the $k$-th interval, by setting $U_t = KX_t + u_k$, the state $X_t$ transits according to

$$dX_t = (A + BK)X_t dt + Bu_k dt + dW_t.$$

Since $A + BK$ is stable, through replacing $A$ in Theorem 2 by $A + BK$ in Algorithm 3, we can obtain a set of estimators $(\hat{A}, \hat{B})$ for $(A, B)$ with small error. This further allows us to accurately estimate $(A, B)$, thereby a controller $\bar{K} = -R^{-1}(\hat{B})^\mathrm{T}P$ closed to $K_*$ is obtained.

During exploitation phase, the near-optimal controller is deployed to minimize the cost, resulting in a regret of $\mathcal{O}(\sqrt{T})$ (see Theorem 6). However, as we lack direct feedback on whether $\bar{K}$ is a stabilizer, we need to detect its stability. Our approach involves replacing it with the known stabilizer $K$ whenever the state deviates too far. Then we introduce the theorem of the regret analysis:

**Theorem 6.** *Let $J_T$ be the expected LQR cost introduced in* (3) *that takes the action $U_t$ as in Algorithm 3. Then for some constant $C \in poly\left(\kappa, M, \mu^{-1}, |\alpha(A + BK)|^{-1}, |\alpha(A + BK_*)|^{-1}\right)$, the regret satisfies:*

$$R_T = J_T - J_T^* \leq C\sqrt{T}.$$

**Proof Sketch of Theorem 6**   We analyze the two phases of our algorithm. During the exploration phase, the stabilizing controller $K$ effectively bounds the trajectory's radius, ensuring the average cost per unit time is within $\mathcal{O}(1)$, resulting in a total exploration cost of $C\sqrt{T}$. In the subsequent exploitation phase, we analyze two scenarios separately. The first scenario occurs when the estimators $(\hat{A}, \hat{B})$ have large errors or when $\|X_t\|_2 \geq T^{1/5}$ for some $t \in [\sqrt{T}, T]$. This situation is rare and contributes a limited expected cost that can be bounded by a constant. The second scenario occurs when $(\hat{A}, \hat{B})$ are accurately estimated, and the control $U_t = -R^{-1}(\hat{B})^\mathrm{T}PX_t$ is applied throughout the exploitation phase. In this case, the trajectory's performance is straightforward to analyze, and the expected cost is bounded by $\mathcal{O}(\sqrt{T}) + J_T^*$.

---

**Algorithm 3** Continuous online control algorithm

---

**Input:** $K, h$ which follows Assumption 3, running time $T$
**for** $k = 0, \ldots, [\frac{\sqrt{T}}{h}] - 1$ **do**

    Sample the action $u_k \overset{\text{i.i.d.}}{\sim} \mathcal{N}(0, I_p)$.
    For $t \in [kh, (k+1)h]$, set $U_t = KX_t + u_k$.
    Observe the new state $x_{k+1}$ at time $(k+1)h$.
**end for**
**Do system identification and estimate dynamics:**
Compute $(\widetilde{A}, \widetilde{B})$ according to (7) by using $\{x_k, u_k\}$.
Compute $\bar{A}, \bar{B}$ by (8) with $\widetilde{A}, \widetilde{B}$, and estimators $(\hat{A}, \hat{B})$ by $\hat{A} = \bar{A} - \bar{B}K, \hat{B} = \bar{B}$.
If $\hat{A}$ is stable, compute $P$ by (5) with estimated $\hat{A}, \hat{B}$, and set $\bar{K} = -R^{-1}(\hat{B})^{\mathrm{T}}P$.
If $\hat{A}$ is not stable or $P$ computed above satisfies $\|P\| \geq T^{\frac{1}{5}}$, then set $\bar{K} = K$.
**Perform exploitation:**
For $t \in [\sqrt{T}, T]$, set $U_t = \bar{K}X_t$.
**Detect bad policy and prevent the trajectory from diverging:**
If for some $t_0 \geq \sqrt{T}$, $\|X_{t_0}\| \geq T^{\frac{1}{5}}$, then set $U_t = KX_t$ for $t \in [t_0, T]$.

---

By summing the expected costs, the total exploration cost is bounded by $\mathcal{O}(\sqrt{T})$, and the exploitation cost is bounded by $J_T^* + \mathcal{O}(\sqrt{T})$. By the definition of regret, $R_T = J_T - J_T^*$, the total regret is $\mathcal{O}(\sqrt{T})$, leading to the result of Theorem 6.

Our result is closely related to the result in [30], along with its similar version [12]. They achieve $\mathcal{O}(\sqrt{T} \log(T))$ regret. However, they further assumes a known stabilization set for obtaining a stable controller, which is stronger compared with ours. Such difference exists because our approach detects divergence and avoids sticking to a controller which is not stable. Morever, in [30], the exploration and exploitation is simultaneous, where a random matrix is added to the near-optimal controller so that both $A$ and $B$ can be identified. This causes an extra $\log(T)$ factor to the regret. In contrast, our algorithm follows an explore-then-commit structure, which is enabled by the efficient system identification results presented previously. Finally, we additionally considered the setup of finite observation, which is not discussed in [30].

## 5.2 Experiments

In this section, we conduct simulation experiments for the baseline algorithm and our proposed algorithm. The baseline algorithm follows the work of [30]. We set $d = p = 3$ for simplicity. Each element of $A$ is sampled uniformly from $[-1, 1]$, making $A$ unstable with high probability. The matrix $B, Q, R$ are set as the identity matrix $I_3$. The sampling interval is set to $h = \frac{1}{30}$.

First, we run Algorithm 1 for system identification. We plot the expected Frobenius norms of the error matrices $\|\hat{A} - A\|_F^2$ and $\|\hat{B} - B\|_F^2$. The results demonstrate that our algorithm can identify $A$ and $B$ within sufficient running time or number of trajectories.

Next, we compare Algorithm 3 with the baseline algorithm. We analyze the normalized regret $R(T)/T^{1/2}$ for different $t \in [600, 10000]$ and plot the results in Figure 1. The results show that our online control algorithm with system identification achieves constant normalized regret (i.e., $O(\sqrt{T})$ regret) and outperforms the baseline algorithm when $T$ is sufficiently large.

## 6 Conclusions, Limitations and Future Directions

In this work, we establish a novel system identification method for continuous-time linear dynamical systems. This method only uses a finite number of observations and can be applied to an algorithm for online LQR continuous control which achieves $\mathcal{O}(\sqrt{T})$ regret on a single trajectory. Compared with existed works, our work not only eases the requirement for data collection and computation, but achieves fast convergence rate in identifying the unknown dynamics as well.

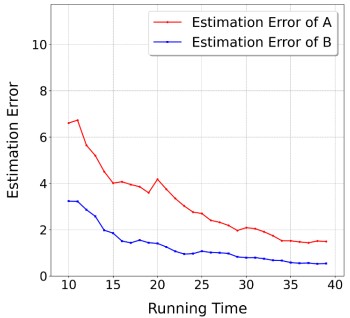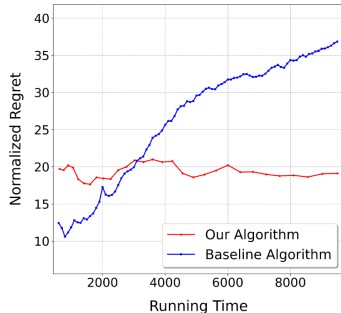

**Figure 1: The empirical validation of our algorithm.** Left: Identification of system dynamics using a single trajectory. Right: The normalized regret $R(T)/T^{1/2}$ of the baseline algorithm and our algorithm. The results show that our algorithm achieves small identification error and is more efficient than the baseline algorithm.

Although our method achieves near-optimal results in system identification and LQR online control for continuous systems with stochastic noise, many questions remain unsolved. First, it is unclear whether our system identification approach can be extended to more challenging setups, such as deterministic or adversarial noise. Additionally, many practical models are non-linear, raising the question of under what conditions discretization methods are effective. We believe these questions are crucial for real-world applications.

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

# A  System Identification for Continuous-time Linear System

In this section, we analysis our system identification method in Algorithm 1 and Algorithm 2. As a preparation, we establish some properties of matrix exponentials and their inverses.

## A.1  Matrix Exponential

For a matrix exponential $e^{At}$, where the largest real component of $A$'s eigenvalues is denoted by $\alpha(A)$, the spectral norm of $e^{At}$ can be well-bounded [14], as demonstrated in Lemma 7.

**Lemma 7.** *Suppose an $n \times n$ matrix $A$ satisfies that $0 > \alpha(A) = \max\{\Re(\lambda_i)|\lambda_i \in \lambda(A)\}$. Let $Q^H A Q = \mathrm{diag}(\lambda_i) + N$ be the Schur decomposition of $A$, and let $M_S(t) = \sum_{k=0}^{n-1} \frac{\|Nt\|_2^k}{k!}$. Then for $t > 0$, we have:*

$$\|e^{At}\| \leq e^{\alpha(A)t M_s(t)}, \tag{11}$$

$$\frac{\|e^{(A+E)t} - e^{At}\|}{\|e^{At}\|} \leq t\|E\|_2 (M_s(t))^2 e^{(t M_S(t)\|E\|_2)}. \tag{12}$$

In a special case where $\alpha(A) \leq 0$, since $M_s(t) \geq 1$ for all $t$, we obtain

$$\|e^{At}\| \leq e^{\alpha(A)t}.$$

We also show some properties of matrix inverse in the following Lemma 8.

**Lemma 8** (Matrix inverse). *For any $A \in \mathbb{R}^{d \times d}$ and $t$ such that $0 < \|At\| \leq \frac{1}{10}$, we have the following estimation of $e^{At}$:*

$$\|e^{At} - I_d\| \leq e^{\|At\|} - 1,$$

*and if we denote $A_1 = e^{At}$, then $A$ also satisfies that*

$$A = \frac{1}{t} \sum_{k \geq 1} \frac{(-1)^{k+1}}{k}(A_1 - I_d)^k.$$

*Proof.* We expand $e^{At}$ by

$$e^{At} = \sum_{k \geq 0} \frac{1}{k!}(At)^k,$$

which follows that

$$\|e^{At} - I_d\| = \left\| \sum_{k \geq 1} \frac{1}{k!}(At)^k \right\| \leq \sum_{k \geq 1} \frac{1}{k!}\|At\|^k = e^{\|At\|} - 1 \leq \frac{1}{9}.$$

Since $\|A_1 - I_d\| < 1$, the progression $A_2 = \sum_{k \geq 1} \frac{(-1)^{k+1}}{kt}(A_1 - I_d)^k$ converges, and thus $e^{A_2 t} = e^{At}$. Furthermore, it can be computed that

$$\|A_2 t\| \leq \sum_{k \geq 1} \left\| \frac{1}{k}(A_1 - I_d) \right\| \leq \sum_{k \geq 1} \frac{1}{k}(\frac{1}{9})^k \leq \frac{1}{8}.$$

Now we show that $A_2 = A$. We have already known that $\|At\|$ and $\|A_2 t\|$ are small. We also note that the function $f : X \to e^X$ $\left(\|X\| \leq \frac{1}{8}\right)$ constitutes a one-to-one mapping. This assertion is supported by the observation that for any $X_1, X_2$ such that $\|X_1\|, \|X_1 + X_2\| \leq \frac{1}{8}$, we have $\|X_2\| \leq \frac{1}{4}$, implying that

$$\|e^{X_1 + X_2} - e^{X_1} - X_2\| = \left\| \sum_{k \geq 2} \frac{1}{k!}(X_1 + X_2)^k - X_1^k \right\| \tag{13}$$

$$\leq \sum_{k \geq 2} \frac{1}{k!} \frac{2^k - 1}{4^{k-1}}\|X_2\| \tag{14}$$

$$\leq \frac{1}{2}\|X_2\|. \tag{15}$$

Then $\left\|e^{X_1+X_2} - e^{X_1}\right\| \geq \frac{1}{2}\|X_2\|$, which means $f$ is one-to-one, and thereby leading that $A_2 = A$.

$\square$

## A.2 Proof of Lemma 1

**Lemma 1.** *In the time interval $[t, t+h]$, the following transition function holds:*

$$X_{t+h} = e^{Ah}X_t + \int_{s=0}^{h} e^{A(h-s)}BU_{t+s}ds + w_t\,,$$

*Proof.* Using Newton-Leibniz formula, we have

$$X_{t+h} = X_t + \int_0^h AX_{t+t_1} + BU_{t+t_1} + \frac{dW_{t+t_1}}{dt}d_{t_1}\,.$$

Let $w_{t+t_1} = BU_{t+t_1} + \frac{dW_{t+t_1}}{dt}$, we have:

$$X_{t+h} = X_t + \int_0^h AX_{t+t_1} + w_{t+t_1}dt_1$$

$$= (I + Ah)X_t + \int_0^h w_{t+t_1}dt_1 + A\int_0^h (X_{t+t_1} - X_t)dt_1$$

$$= (I + Ah)X_t + \int_0^h w_{t+t_1}dt_1 + A\int_0^h \int_0^{t_1} AX_{t+t_2} + w_{t+t_2}dt_2dt_1$$

$$= (I + Ah + \frac{1}{2}A^2h^2)X_t + \int_0^h (I + A(h - t_1))w_{t+t_1}dt_1 + A^2\int_0^h \int_0^{t_1}(X_{t+t_2} - X_t)dt_2dt_1\,,$$

where the last equality we use the Fubini theorem to change the integral order of $t_1$ and $t_2$ to calculate the second term.

Suppose we already have the following equality for integer $m$ (The case $m = 2$ has been checked above):

$$X_{t+h} = (I + \sum_{k=1}^{m} \frac{(hA)^k}{k!})X_t + \int_0^h [I + \sum_{k=1}^{m-1} \frac{((h - t_1)A)^k}{k!}]w_{t+t_1}dt_1$$

$$+ A^m \int_{0 \leq t_m \leq \ldots \leq h} (X_{t+t_m} - X_t)dt_1dt_2...d_{t_m}\,.$$

Then, replace $X_{t+t_m} - X_t$ by $\int_0^{t_m}[AX_t + A(X_{t+t_{m+1}} - X_t) + w_{t_{m+1}}]d_{t_{m+1}}$, we get:

$$A^m \int_{0 \leq t_m...\leq h} (X_{t+t_m} - X_t)dt_1dt_2...d_{t_m}$$

$$= A^{m+1}X_t \int_{0 \leq t_{m+1} \leq ... \leq h} dt_1dt_2...d_{t_{m+1}} + A^m \int_{0 \leq t_{m+1} \leq ... \leq h} w_{t+t_{m+1}}dt_1dt_2...d_{t_m}$$

$$+ A^{m+1} \int_{0 \leq t_{m+1} \leq ... \leq h} (X_{t+t_{m+1}} - X_t)dt_1dt_2...d_{t_{m+1}}\,.$$

Using the property that

$$\int_{0 \leq x_1 \leq x_2 \leq ... \leq x_m \leq h} dx_1dx_2...dx_m = \frac{h^m}{m!}\,.$$

We finally get

$$A^m \int_{0 \leq t_m \leq t_{m-1} \leq \dots \leq t_1 \leq h} (X_{t+t_m} - X_t) dt_1 dt_2 \dots d_{t_m}$$

$$= \frac{h^{m+1}}{(m+1)!} A^{m+1} X_t + A^m \int_0^h \frac{(h - t_{m+1})^m}{m!} w_{t+t_{m+1}} dt_{m+1}$$

$$+ A^{m+1} \int_{0 \leq t_{m+1} \leq \dots \leq h} (X_{t+t_{m+1}} - X_t) dt_1 \dots d_{t_{m+1}}.$$

In the calculation of the second term we use the Fubini theorem to change the integral order of $t_{m+1}$ and $t_1, t_2 \dots t_m$.

So the induction hypothesis is true. For any positive integer m, we have the following equality:

$$X_{t+h} = (I + \sum_{k=1}^m \frac{(hA)^k}{k!}) X_t + \int_{t_1=0}^h [I + \sum_{k=1}^{m-1} \frac{((h-t_1)A)^k}{k!}] w_{t+t_1} dt_1$$

$$+ A^m \int_{0 \leq t_m \dots \leq h} (X_{t+t_m} - X_t) dt_1 dt_2 \dots d_{t_m}.$$

For the time interval $\tilde{t} \in [t, t+h]$, by the continuity of $X_{\tilde{t}}$ we know that $X_{\tilde{t}}$ is uniformly bounded by some constant $C$. Therefore we have the convergence of the third term in the RHS:

$$\lim_{m \to \infty} \| A^m \int_{0 \leq t_m \dots \leq h} (X_{t+t_m} - X_t) dt_1 dt_2 \dots d_{t_m} \| \leq \lim_{m \to \infty} \frac{2C(\kappa_A h)^m}{m!} = 0.$$

Therefore, we finally get:

$$X_{t+h} = e^{Ah} X_t + \int_0^h e^{A(h-s)} w_{t+s} ds \tag{16}$$

Now, we use $w_{t+s} = BU_{t+s} + \frac{dW_{t+s}}{dt}$, we get:

$$X_{t+h} = e^{Ah} X_t + \int_0^h e^{A(h-s)} BU_{t+s} ds + \int_0^h e^{A(h-s)} dW_{t+s} \tag{17}$$

$$= e^{Ah} X_t + \int_0^h e^{A(h-s)} BU_{t+s} ds + w_t, \tag{18}$$

where $w_t$ is Gaussian noise $\mathcal{N}(0, \Sigma)$ with covariance $\Sigma = \int_0^h e^{As} e^{A^\mathrm{T} s} ds$.

$\square$

## A.3 Proof of Lemma 3

We restate Lemma 3 and provide the proof here.

**Lemma 3** In Algorithm 1, 2, suppose we have obtained the relative error $\|\widetilde{A} - A^{'}\|, \|\widetilde{B} - B^{'}\| \leq \epsilon$ for some $\epsilon \leq \frac{1}{15}$ and $\|Ah\| \leq \frac{1}{15}$, then we have the following relative error of the primal system:

$$\|\hat{A} - A\|, \|\hat{B} - B\| \leq \frac{C}{h}\epsilon, \tag{19}$$

where $C$ is a constant independent of $h$.

*Proof.* Firstly, according to Lemma 8, the estimated $\widetilde{A}$ is not too far away from $I_d$, as we have:

$$\left\| \widetilde{A} - I_d \right\| \leq \left\| \widetilde{A} - e^{Ah} \right\| + \left\| e^{Ah} - I_d \right\| \leq \epsilon + e^{\|A\|h} - 1 \leq \frac{1}{7},$$

Then, from (8) we can bound the matrix norm $\left\|\hat{A}h\right\|$ by

$$\left\|\hat{A}h\right\| = \left\|\sum_{k\geq 1}\frac{(-1)^{k-1}}{k}(\widetilde{A}-I)^k\right\| \leq \sum_{k\geq 1}\frac{1}{k}(\frac{1}{7})^k \leq \frac{1}{6}\,.$$

Now, let's denote $A_1 = Ah$ and $A_2 = \hat{A}h - A_1$, satisfying the relations $A' = e^{A_1}$ and $\widetilde{A} = e^{A_1+A_2}$. It is given that $\|A_1\| \leq \frac{1}{15}$ and $\|A_2\| \leq \|A_1\| + \|\hat{A}h\| \leq \frac{1}{4}$, so by (13), we obtain that $\|\hat{A}-A\|h = \|A_2\| \leq 2\|\widetilde{A}-A'\|$, which follows that $\|\hat{A}-A\| \leq \frac{2}{h}\|\widetilde{A}-A'\| \leq \frac{2}{h}\epsilon$. $\qquad\square$

Next, we will upper bound the estimation error of $B$. Let $A_h = \int_{t=0}^h e^{At}dt$ and $\bar{A}_h = \int_{t=0}^h e^{\hat{A}t}dt$, satisfying

$$\|A_h - hI\| = \left\|\int_{t=0}^h (e^{At}-I)dt\right\| \leq \int_{t=0}^h \|e^{At}-I\|\,dt \leq \int_{t=0}^h (e^{\|A\|t}-1)dt \leq \frac{1}{20}h\,,$$

$$\|\bar{A}_h - A_h\| = \left\|\int_{t=0}^h e^{\hat{A}t}-e^{At}dt\right\| \leq \int_{t=0}^h \left\|e^{\hat{A}t}-e^{At}\right\|dt \leq \frac{3}{2}\int_{t=0}^h \|\hat{A}-A\|t\,dt \leq \frac{3}{4}h\epsilon\,.$$

This follows that

$$\|A_h^{-1}\| = \frac{1}{h}\left\|\left[I+(\frac{A_h}{h}-I)\right]^{-1}\right\| \leq \frac{1}{h}\sum_{k\geq 0}\left\|\frac{A_h}{h}-I\right\|^k \leq \frac{20}{19h}\,,$$

$$\|(\bar{A}_h)^{-1} - A_h^{-1}\|$$
$$= \left\|A_h^{-1}\right\|\left\|\left[I+(\bar{A}_h-A_h)A_h^{-1}\right]^{-1}-I\right\| \leq \|A_h^{-1}\|\frac{1}{1-\|(\bar{A}_h-A_h)A_h^{-1}\|} \leq \frac{1}{h}\epsilon\,.$$

Since $B$ and its estimator $\hat{B}$ satisfy that

$$B = (A_h)^{-1}B', \hat{B} = (\bar{A}_h)^{-1}\widetilde{B}\,,$$

we can upper bound the estimation error $\left\|\hat{B}-B\right\|$ by

$$\left\|\hat{B}-B\right\| \leq \|(\bar{A}_h)^{-1}-A_h^{-1}\|\left\|B'\right\| + \|(\bar{A}_h)^{-1}\|\left\|\widetilde{B}-B'\right\| \leq \frac{\|B'\|}{h}\epsilon + \frac{2}{h}\epsilon \leq (2\|B\|+\frac{2}{h})\epsilon\,,$$

where the last inequality is because $\|B'\| \leq \|A_h\|\|B\| \leq 2h\|B\|$.

Since $2\|B\| \leq 2\kappa_B \leq \frac{1}{h}\cdot\frac{2\kappa_B}{15\kappa_A} \leq \frac{\kappa_B}{\kappa_A}$, we obtain Lemma 3.

### A.4 Proof of Theorem 2

In this section, we derive the proof of Theorem 2. We upper bound the estimation errors of intermediate dynamics $(A', B')$, obtained as in (7). We first prove Lemma 9 below, providing system identification results on a single trajectory with a stable controller.

**Lemma 9.** *Consider the trajectory* $x_{k+1} = Ax_k + Bu_k + w_k$ *with* $A \in \mathbb{R}^{d\times d}$, $\|A\| < 1$, $B \in \mathbb{R}^{d\times p}$; $u_k \sim \mathcal{N}(0, I_p)$ *and* $w_k \sim \mathcal{N}(0, \Sigma)$ *are i.i.d. random variables. Suppose we compute* $(\hat{A}, \hat{B})$ *by*

$$(\hat{A})^{\mathrm{T}} = \left[\sum_{k=0}^{T_0-1}x_k x_k^{\mathrm{T}}\right]^\dagger \sum_{k=0}^{T_0-1}x_k x_{k+1}^{\mathrm{T}}, (\hat{B})^{\mathrm{T}} = \left[\sum_{k=0}^{T_0-1}u_k u_k^{\mathrm{T}}\right]^\dagger \sum_{k=0}^{T_0-1}u_k\left(x_{k+1}-\hat{A}x_k\right)^{\mathrm{T}}. \quad (20)$$

*Then there exists a constant $C$ (depending only on $A$, $B$, $d$, $p$ and $\Sigma$) such that for $T \geq C\left(\|X_0\|_2^2 + \log^2(1/\delta)\right)$, w.p. at least $1-\delta$:*

$$\|\hat{A}-A\|, \|\hat{B}-B\| \leq C\sqrt{\frac{\log(1/\delta)}{T}}\,, \quad (21)$$

We first provide Lemma 10, which is used as the base of Lemma 9.

**Lemma 10.** *Consider $A \in \mathbb{R}^{d \times d}$ such that $\rho(A) < 1$ and the system $X_{k+1} = AX_k + w_k$ with $w_k \sim \mathcal{N}(0, \Sigma)$ be i.i.d. random variables. Suppose we estimate $A$ as in (7). Then there exists a constant $C$ depending on $A$, $\Sigma$ and $d$ such that for $T \geq C(\|X_0\|_2^2 + \log(1/\delta))$, w.p. at least $1 - \delta$, we have:*

$$\|\hat{A} - A\| \leq C\sqrt{\frac{\log(1/\delta)}{T}} .$$

The work of [33] has discussed such systems in their Theorem 2.4, and we list it below:

**Theorem 11.** *Fix $\epsilon, \delta \in (0, 1)$, $T \in \mathbb{N}$ and $0 \prec \Gamma_{sb} \prec \bar{\Gamma}$. Then if $(X_t, Y_t)_{t \geq 1} \in (\mathbb{R}^d \times \mathbb{R}^n)^T$ is a random sequence such that (a) $Y_t = A_* X_t + \eta_t$, where $\eta_t | \mathcal{F}_t$ is $\sigma^2$-sub-Gaussian and mean zero, (b) $X_1, ..., X_T$ satisfies the $(k, \Gamma_{sb}, p)$-small ball condition, and (c) such that $\mathbb{P}\left[\sum_{t=1}^{T} X_t X_t^{\mathrm{T}} \not\succeq T\bar{\Gamma}\right] \leq \delta$. Then if*

$$T \geq \frac{10k}{p^2}\left(\log(1/\delta) + 2d\log(10/p) + \log\det(\bar{\Gamma}\Gamma_{sb}^{-1})\right) ,$$

*we have*

$$\mathbb{P}\left[\|\hat{A} - A_*\| > \frac{90\sigma}{p}\sqrt{\frac{n + d\log\frac{10}{p} + \log\det(\bar{\Gamma}\Gamma_{sb}^{-1}) + \log(\frac{1}{\delta})}{T\lambda_{\min}(\Gamma_{sb})}}\right] \leq 3\delta .$$

*Here, the $(k, \Gamma_{sb}, p)$-small ball condition is defined as follows. Let $(Z_t)_{t \geq 1}$ be an $\mathcal{F}_{t t \geq 1}$-adapted random process taking values in $\mathbb{R}$. We say $(Z_t)_{t \geq 1}$ satisfies the $(k, \nu, p)$-block martingale small-ball (BMSB) condition if, for any $j \geq 0$, one has $\frac{1}{k}\sum_{i=1}^{k}\mathbb{P}(|Z_{j+i}| \geq \nu | \mathcal{F}_j) \geq p$ almost surely. Given a process $(X_t)_{t \geq 1}$ taking values in $\mathbb{R}^d$, we say that it satisfies the $(k, \Gamma_{sb}, p)$-BMSB condition for $\Gamma_{sb} \succ 0$ if, for any fixed $w \in \mathcal{S}^{d-1}$, the process $Z_t := \langle w, X_t \rangle$ satisfies $(k, \sqrt{w^{\mathrm{T}}\Gamma_{sb}w}, p)$-BMSB.*

In the work of [33], they have discussed the case when $X_0 = 0$, and now we modify it to a general starting state $X_0$. From (7), we derive the estimation error of $A$ as

$$\hat{A}^{\mathrm{T}} - A^{\mathrm{T}} = \left[\sum_{k=0}^{T-1} X_k X_k^{\mathrm{T}}\right]^{\dagger} \sum_{k=0}^{T-1} X_k X_{k+1}^{\mathrm{T}} - A^{\mathrm{T}}$$

$$= \left[\sum_{k=0}^{T-1} X_k X_k^{\mathrm{T}}\right]^{\dagger} \sum_{k=0}^{T-1} X_k (AX_k + w_k)^{\mathrm{T}} - A^{\mathrm{T}}$$

$$= \left[\sum_{k=0}^{T-1} X_k X_k^{\mathrm{T}}\right]^{\dagger} \sum_{k=0}^{T-1} X_k w_k^{\mathrm{T}} .$$

For the first term, consider any $v \in \mathcal{S}^{d-1}$, we lower bound $v^{\mathrm{T}}\left(\sum_{k=0}^{T-1} X_k X_k^{\mathrm{T}}\right)v$. Let $a_k = v^{\mathrm{T}}X_k$, then $a_k = v^{\mathrm{T}}AX_{k-1} + v^{\mathrm{T}}w_k$. We claim that for any $k \geq 1$, $\mathbb{P}\left[|a_k| \geq \frac{1}{2}|X_{k-1}\right] \geq \frac{1}{2}$. Let $b_k = v^{\mathrm{T}}w_k$, which is independent of $X_{k-1}$. It suffices to show that for any $c \in \mathbb{R}$, $\mathbb{P}\left[b_k \in [c, c+1]\right] \leq \frac{1}{2}$. Since $\|v\|_2 = 1$ and $w_k \sim \mathcal{N}(0, I_d)$, we have $b_k \sim \mathcal{N}(0, 1)$, from which we estimate the probability as

$$\mathbb{P}\left[b_k \in [c, c+1]\right] = \int_{x=c}^{c+1} \frac{1}{\sqrt{2\pi}}e^{-\frac{1}{2}x^2}dx \leq \frac{1}{\sqrt{2\pi}} \leq \frac{1}{2} . \tag{22}$$

Based on (22), we can simply choose $k = 1$, $\Gamma_{sb} = \frac{1}{4}I_d$ and $p = \frac{1}{2}$, then the random sequence $(X_i)_{i \geq 0}$ satisfies the $(k, \Gamma_{sb}, p)$-BMSB condition. It remains to choose a proper $\bar{\Gamma}$ that meets the condition (c) in Theorem 11.

Since $X_k = A^k X_0 + \sum_{i=1}^{k} A^{k-i} w_i$, we have:

$$\mathbb{E}\left[\sum_{k=0}^{T-1} X_k X_k^{\mathrm{T}}\right] = \mathbb{E}\left[\sum_{k=0}^{T-1}\left(A^k X_0 + \sum_{i=1}^{k} A^{k-i} w_i\right)\left(A^k X_0 + \sum_{i=1}^{k} A^{k-i} w_i\right)^{\mathrm{T}}\right]$$

$$= \sum_{k=0}^{T-1} A^k X_0 X_0^{\mathrm{T}} (A^k)^{\mathrm{T}} + \mathbb{E}\left[\sum_{k=0}^{T-1}\sum_{i=0}^{k} A^k X_0 w_i^{\mathrm{T}} (A^{k-i})^{\mathrm{T}}\right]$$

$$+ \mathbb{E}\left[\sum_{k=0}^{T-1}\sum_{i=0}^{k} A^{k-i} w_i X_0^{\mathrm{T}} (A^k)^{\mathrm{T}}\right] + \mathbb{E}\left[\sum_{k=0}^{T-1}\sum_{i,j=0}^{k} A^{k-i} w_i w_j^{\mathrm{T}} (A^{k-j})^{\mathrm{T}}\right]$$

$$= \sum_{k=0}^{T-1} A^k X_0 X_0^{\mathrm{T}} (A^k)^{\mathrm{T}} + \sum_{k=0}^{T-1}\sum_{i=0}^{k} A^{k-i} \Sigma (A^{k-i})^{\mathrm{T}}.$$

Let $\Gamma_\infty = \sum_{k\geq 0} A^k \Sigma (A^k)^{\mathrm{T}}$ which is bounded and $C_1$ be a constant such that $C_1 \geq \sum_{k\geq 0} \|A^k\|^2$. We then show that for $\bar{\Gamma} = \left(\frac{C_1 \|X_0\|_2^2}{T} dI_d + d\|\Gamma_\infty\| I_d\right)/\delta$, the condition (c) in Theorem 11 is satisfied. This is because $\mathbb{E}\left[\mathrm{tr}\left(\sum_{k=0}^{T-1} X_k X_k^{\mathrm{T}}\right)\right] = \mathrm{tr}\left(\mathbb{E}\left[\sum_{k=0}^{T-1} X_k X_k^{\mathrm{T}}\right]\right) \leq \frac{T\delta}{d}\mathrm{tr}(\bar{\Gamma})$ so that $\mathbb{P}\left[\mathrm{tr}(\sum_{k=0}^{T-1} X_k X_k^{\mathrm{T}}) \geq \frac{1}{d}T\mathrm{tr}(\bar{\Gamma})\right] \leq \delta$. Furthermore, a necessary condition for $\sum_{k=0}^{T-1} X_k X^{\mathrm{T}} \not\prec T\bar{\Gamma}$ is $\mathrm{tr}(\sum_{k=0}^{T-1} X_k X^{\mathrm{T}}) \geq \frac{1}{d}T\mathrm{tr}(\bar{\Gamma})$.

Now, we apply such $\bar{\Gamma}$ to Theorem 11. It can be computed that

$$\log \det(\bar{\Gamma}\Gamma_{sb}^{-1}) = d\log\left(4d(C_1\|X_0\|_2^2/T + \|\Gamma_\infty\|)\right) + d\log(1/\delta).$$

Then when $T \geq C_1\|X_0\|^2$ as well as $T \geq 40\left(2d\log(20) + d\log(4d(1 + \|\Gamma_\infty\|)) + 2d\log(1/\delta)\right)$, we have:

$$\mathbb{P}\left[\|\hat{A} - A\| > 360\sqrt{\frac{d + d\log(20) + d\log(4d(1 + \|\Gamma_\infty\|)) + 2d\log(\frac{1}{\delta})}{T}}\right] \leq 3\delta.$$

This implies our Lemma 10.

**Proof of Lemma 9**   As for the estimation error $\|\hat{A} - A\|$, let $w_k' = Bu_k + w_k \sim \mathcal{N}(0, \Sigma + BB^{\mathrm{T}})$, which form a sequence of $i.i.d$ random variables. With the results in Lemma 10, there exist some constants $C_1, C_2$ such that, as long as $T \geq C_1\left(\|X_0\|_2^2 + \log(1/\delta)\right)$ we have:

$$\|\hat{A} - A\| \leq C_2\sqrt{\frac{\log(1/\delta)}{T}}.$$

Now we upper bound the estimation error $\|\hat{B} - B\|$. With the expression in (7), we obtain:

$$\|\hat{B} - B\| = \left\|\left[\sum_{k=0}^{T-1} u_k u_k^{\mathrm{T}}\right]^{\dagger} \sum_{k=0}^{T-1} u_k \left[(A - \hat{A})X_k + w_k\right]^{\mathrm{T}}\right\|$$

$$\leq \lambda_{\min}^{-1}\left(\sum_{k=0}^{T-1} u_k u_k^{\mathrm{T}}\right)\left[\left\|\sum_{k=0}^{T-1} u_k X_k^{\mathrm{T}}\right\| \|\hat{A} - A\| + \left\|\sum_{k=0}^{T-1} u_k w_k^{\mathrm{T}}\right\|\right].$$

For the quantities $\lambda_{\min}^{-1}(\sum_{k=0}^{T-1} u_k u_k^{\mathrm{T}})$ and $\|\sum_{k=0}^{T-1} u_k w_k^{\mathrm{T}}\|$, we apply Lemma 2.1. and Lemma 2.2. in the work of [9], where they present the following results.

**Lemma 12.** *Let $N \geq 2\log(1/\delta)$. Suppose $f_k \in \mathbb{R}^m$, $g_k \in \mathbb{R}^n$ are independent vectors such that $f_k \sim \mathcal{N}(0, \Sigma_f)$ and $g_k \sim \mathcal{N}(0, \Sigma_g)$ for $1 \leq k \leq N$. With probability at least $1 - \delta$,*

$$\left\|\sum_{k=1}^{N} f_k g_k^{\mathrm{T}}\right\| \leq 4\|\Sigma_f\|_2^{1/2}\|\Sigma_g\|_2^{1/2}\sqrt{N(m+n)\log(9/\delta)}.$$

**Lemma 13.** *Let $X \in \mathbb{R}^{N \times n}$ have i.i.d. $\mathcal{N}(0,1)$ entries. With probability at least $1 - \delta$,*

$$\sqrt{\lambda_{\min}(X^{\mathrm{T}}X)} \geq \sqrt{N} - \sqrt{n} - \sqrt{2\log(1/\delta)}.$$

With these two lemmas, we can conclude that if $T \geq 32(d+p)\log(4/\delta)$, then both $\lambda_{\min}(u_k u_k^{\mathrm{T}}) \geq \frac{1}{2}T$ and $\left\|\sum_{k=0}^{T-1} u_k w_k^{\mathrm{T}}\right\| \leq 4\|\Sigma\|_2^{1/2} \sqrt{T(d+p)\log(18/\delta)}$, w.p. at least $1 - \delta$.

Now we concentrate on the term $\left\|\sum_{k=0}^{T-1} u_k X_k^{\mathrm{T}}\right\|$. Since $w_i' = Bu_i + w_i \sim \mathcal{N}(0, \Sigma + BB^{\mathrm{T}})$, it can be directly computed that, w.p. at least $1 - \delta/T$, $\left\|w_i'\right\|_2 \leq 2\left\|d(\Sigma + BB^{\mathrm{T}})\right\|_2^{1/2} \sqrt{\log(T/\delta)}$. Then by union bound we get $\mathbb{P}\left[\sup_{0 \leq i \leq T-1} \left\|w_i'\right\|_2 \leq 2\|\Sigma + BB^{\mathrm{T}}\|_2^{1/2}\sqrt{d\log(T/\delta)}\right] \leq \delta$. Furthermore, when $\sup_{0 \leq i \leq T-1} \left\|w_i'\right\|_2 \leq 2\left\|\Sigma + BB^{\mathrm{T}}\right\|_2^{1/2}\sqrt{d\log(T/\delta)}$, we must have

$$\|X_k\|_2 = \left\|A^k X_0 + \sum_{i=0}^{k-1} A^{k-1-i} w_i\right\| \leq \|A\|^k\|X_0\|_2 + \frac{2}{1-\|A\|}\left\|\Sigma + BB^{\mathrm{T}}\right\|_2^{1/2}\sqrt{d\log(T/\delta)}. \tag{23}$$

For any $u \in \mathcal{S}^{p-1}$ and $v \in \mathcal{S}^{d-1}$, let $x_i = u^{\mathrm{T}}u_i (0 \leq i \leq T-1)$. Then, $x_i$ follows a normal distribution $x_i \sim \mathcal{N}(0,1)$ and $\{x_i\}$ is a sequence of independent random variables. Furthermore, $x_k$ is also independent of $(X_i)_{0 \leq i \leq k}$. On the other hand, denote $y_k = X_k^{\mathrm{T}}v$, (23) implies that w.p. at least $1 - \delta$, for all $k$ we have $|y_k| \leq \|X_0\|_2 + \frac{2}{1-\|A\|}\left\|\Sigma + BB^{\mathrm{T}}\right\|_2^{1/2}\sqrt{d\log(T/\delta)} := Y$. Let

$$Z_k := \sum_{i=0}^{k} u^{\mathrm{T}}\left(u_k X_k^{\mathrm{T}}\right)v \cdot 1_{\|X_k\|_2 \leq Y} = \sum_{i=0}^{k} x_k y_k \cdot 1_{\|X_k\|_2 \leq Y},$$

and let $\mathcal{F}_0, \mathcal{F}_1, ..., \mathcal{F}_T$ be the filtration of $X_0, X_1, ..., X_T$, then for any $\alpha \geq 0$,

$$\mathbb{E}\left[e^{\frac{\alpha Z_{k+1}}{Y}}|\mathcal{F}_k\right] = e^{\frac{\alpha Z_k}{Y}}\mathbb{E}_{X_{k+1}}\left[\mathbb{E}_{x \sim \mathcal{N}(0,1)}\left[\exp\left(\frac{\alpha x y_{k+1} \cdot 1_{\|X_{k+1}\|_2 \leq Y}}{Y}\right)\right]\right] \leq e^{\frac{1}{2}\alpha^2}e^{\frac{\alpha Z_k}{Y}},$$

implying that $\mathbb{E}\left[e^{\frac{\alpha Z_{k+1}}{Y}}\right] \leq e^{\frac{1}{2}\alpha^2}\mathbb{E}\left[e^{\frac{\alpha Z_{k+1}}{Y}}\right]$ So we have: $\mathbb{E}\left[e^{\frac{\alpha Z_{T-1}}{Y}}\right] \leq e^{\frac{1}{2}\alpha^2 T}$. By choosing $\alpha = \pm\sqrt{\frac{1}{T}}$, we obtain that

$$\mathbb{P}\left[|Z_{T-1}| \geq 2Y\sqrt{T\log(4/\delta)}\right] \leq \delta$$

For $\mathcal{T}_d$ be a $\frac{1}{4}$-net of $\mathcal{S}^{d-1}$ and $\mathcal{T}_p$ be a $\frac{1}{4}$-net of $\mathcal{S}^{p-1}$, we use union bound on them and obtain that, w.p. at least $1 - \delta$

$$|Z_{T-1}| \leq 2Y\sqrt{T\log(4|\mathcal{T}_p||\mathcal{T}_d|/\delta)} \leq 2Y\sqrt{T[4(d+p) + \log(4/\delta)]}.$$

Where the last inequality is because $|\mathcal{T}_p| \leq 9^p$ and $|\mathcal{T}_d| \leq 9^d$

Next we upper bound $\left\|\sum_{k=0}^{T-1} u_k X_k^{\mathrm{T}}\right\|$. For any $u_* \in \mathcal{S}^{p-1}$ and $v_* \in \mathcal{S}^{p-1}$, with some $u \in \mathcal{T}_p$ and $v \in \mathcal{T}_d$ s.t. $\|u - u_*\|_2, \|v - v_*\|_2 \leq \frac{1}{2}$, we have:

$$\left|u_*^{\mathrm{T}}(\sum_{k=0}^{T-1} u_k X_k^{\mathrm{T}})v_*\right|$$

$$\leq \left|u^{\mathrm{T}}(\sum_{k=0}^{T-1} u_k X_k^{\mathrm{T}})v\right| + \left|(u_* - u)^{\mathrm{T}}(\sum_{k=0}^{T-1} u_k X_k^{\mathrm{T}})v_*\right| + \left|u^{\mathrm{T}}(\sum_{k=0}^{T-1} u_k X_k^{\mathrm{T}})(v - v_*)\right|$$

$$\leq \sup_{u \in \mathcal{T}_p, v \in \mathcal{T}_d}\left|u^{\mathrm{T}}(\sum_{k=0}^{T-1} u_k X_k^{\mathrm{T}})v\right| + \frac{1}{2}\left\|\sum_{k=0}^{T-1} u_k X_k^{\mathrm{T}}\right\|.$$

This leads $\left\| \sum_{k=0}^{T-1} u_k X_k^{\mathrm{T}} \right\| \leq 2 \sup_{u \in \mathcal{T}_p, v \in \mathcal{T}_d} \left| u^{\mathrm{T}} (\sum_{k=0}^{T-1} u_k X_k^{\mathrm{T}}) v \right|$. Therefore, for any $\delta \in (0, \frac{1}{2})$, we have:

$$\mathbb{P}\left[ \left\| \sum_{k=0}^{T-1} u_k X_k^{\mathrm{T}} \right\|_2 \geq 4Y\sqrt{T[4(d+p) + \log(4/\delta)]} \right]$$

$$\leq \mathbb{P}\left[ \sup_{u \in \mathcal{T}_d, v \in \mathcal{T}_p} \left| u^{\mathrm{T}} \left( \sum_{k=0}^{T-1} u_k X_k^{\mathrm{T}} 1_{\|X_k\|_2 \leq Y} \right) v \right| \geq 2Y\sqrt{T[4(d+p) + \log(4/\delta)]} \right]$$

$$+ \mathbb{P}\left[ \exists\, 0 \leq k \leq T-1, \|X_k\|_2 \geq Y \right]$$

$$\leq 2\delta .$$

We choose constant $C$ depending on $A, B, d, p$ such that for all $T \geq C\left(\|X_0\|_2^2 + \log^2(1/\delta)\right)$,

$$4Y\sqrt{T[4(d+p) + \log(4/\delta)]} \leq T ,$$

and we further have: whenever $T \geq C\left(\|X_0\|_2^2 + \log^2(1/\delta)\right)$, w.p. at least $1 - 3\delta$,

$$\left\| \sum_{k=0}^{T-1} u_k X_k^{\mathrm{T}} \right\| \|\hat{A} - A\| \leq C_2 \sqrt{\log(1/\delta)T} .$$

Finally, when $T \geq \max\left(C\left(\|X_0\|_2^2 + \log^2(1/\delta)\right), 32(d+p)\log(4/\delta)\right)$, we combine this upper bound with $\mathbb{P}\left(\lambda_{\min}(\sum_{k=0}^{T-1} u_k u_k^{\mathrm{T}}) \leq \frac{1}{2}T\right) \leq \delta$, and obtain Lemma 9.

### A.5 Proof of Theorem 5

Now, we aim to establish Theorem 5. The analysis of system identification for discrete-time linear dynamical systems with multiple trajectories has been thoroughly investigated by [9]. We hereby cite their findings, denoting the relevant result as Lemma 14.

**Lemma 14.** *Suppose we have $N$ i.i.d. trajectories $X_k^i$, each is defined by $X_{(k+1)h}^i = AX_k^i + Bu_k^i + w_k^i$, where $T_0$ is any integer, $u_k^i \sim \mathcal{N}(0, I_p)$ and $w_k^i \sim \mathcal{N}(0, \Sigma)$ are two sets of i.i.d. random variables. Then, for the estimator $(\hat{A}, \hat{B})$ of*

$$(\hat{A}, \hat{B}) \in \arg\min_{(A,B)} \frac{1}{2} \sum_{i=1}^{N} \left\| X_{T_0}^i - AX_{T_0-1}^i - Bu_{T_0-1}^i \right\|_2^2 \tag{24}$$

*with probability at least $1 - \delta$, we have:*

$$\|\hat{A} - A\|, \|\hat{B} - B\| \leq \mathcal{O}\left( \sqrt{\frac{\log(1/\delta)}{N}} \right) .$$

Combining Lemma 14 with Lemma 3, we directly obtain Theorem 5.

### A.6 Lower Bound of System Identification with Finite Observation

We restate and provide the proof of Theorem 4.

**Theorem 4** Suppose $T \geq 1$ be the running time of a single trajectory of continuous-time linear differential system, represented as in (2). Then there exist constants $c_1, c_2$ independent of $d$ such that, for any finite set of observed points $\{t_0 = 0, t_1, t_2, ..., t_N = T\}$, and any (possibly randomized) estimator function $\phi : \{X_{t_0}, X_{t_1}, ..., X_{t_N}\} \to \mathbb{R}^{d \times d}$, there exists bounded $A, B$ satisfying $\mathbb{P}\left[ \|\phi(\{X_i\}_{i \leq N}) - A\| \geq \frac{c_1}{\sqrt{T}} \right] \geq c_2$. Here the probability corresponds to the dynamical system dominated by $(A, B)$.

*Proof.* Firstly, we consider a special case where $d = 1$, and let $A = [-1]$ and $\bar{A} = [-1 - \delta]$. We show that when $\delta = \frac{1}{5\sqrt{T}}$, for the two dynamical systems $\psi_\theta : dX_t = AX_t dt + dW_t$ and $\psi_{\bar{\theta}} : dX_t = \bar{A}X_t dt + dW_t$, any algorithm $\mathcal{A}$ that outputs according only to $\{X_{t_0}, X_{t_1}, ..., X_{t_N}\}$ satisfies:

$$\max\left\{\mathbb{P}\left[\|\mathcal{A}(X_{t_0}, X_{t_1}, ..., X_{t_N}) - A\| \geq \frac{1}{10\sqrt{T}}\right], \mathbb{P}\left[\|\mathcal{A}(X_{t_0}, X_{t_1}, ..., X_{t_N}) - \bar{A}\| \geq \frac{1}{10\sqrt{T}}\right]\right\}$$

$$\geq \frac{1}{4e^3}.$$

We note that this special case can be easily generalized to any dimension $d$, since we can consider $A = -I_d$ and $\bar{A}$ satisfies $\bar{A}_{1,1} = A_{1,1} - \delta$, and for any $(i, j) \neq (1, 1)$, $\bar{A}_{i,j} = A_{i,j}$. In this case the last $d - 1$ dimension is independent of the first dimension, so it is essentially the same as the simplest one-dimensional case.

Denote $X = \{X_{t_0}, X_{t_1}, ..., X_{t_N}\}$ and $g(X), \bar{g}(X)$ be the probability density of $\psi_\theta$ and $\psi_{\bar{\theta}}$, respectively. For these two probability densities we have:

$$g(X) = \prod_{i=1}^{N} \frac{1}{\sqrt{2\pi\Gamma(t_i - t_{i-1})}} exp\left(-\frac{1}{2\Gamma(t_i - t_{i-1})}(X_{t_i} - e^{-(t_i - t_{i-1})}X_{t_{i-1}})^2\right),$$

and

$$\bar{g}(X) = \prod_{i=1}^{N} \frac{1}{\sqrt{2\pi\bar{\Gamma}(t_i - t_{i-1})}} exp\left(-\frac{1}{2\bar{\Gamma}(t_i - t_{i-1})}(X_{t_i} - e^{-(1+\delta)(t_i - t_{i-1})}X_{t_{i-1}})^2\right).$$

Where

$$\Gamma(t) = \int_{s=0}^{t} e^{-2s}ds = \frac{1}{2}(1 - e^{-2t}) \quad \bar{\Gamma}(t) = \int_{s=0}^{t} e^{(-2-2\delta)s}ds = \frac{1}{2+2\delta}(1 - e^{-(2+2\delta)t}).$$

Denote $\alpha_i = \sqrt{\frac{1}{\Gamma(t_i - t_{i-1})}}(X_{t_i} - e^{-(t_i - t_{i-1})}X_{t_{i-1}})$,
$\beta_i = \sqrt{\frac{1}{\Gamma(t_i - t_{i-1})}}(e^{-(t_i - t_{i-1})} - e^{-(1+\delta)(t_i - t_{i-1})})X_{t_{i-1}}$ and $\gamma_i = \sqrt{\frac{\Gamma(t_i - t_{i-1})}{\bar{\Gamma}(t_i - t_{i-1})}}$. Then

$$\ln\left(\frac{g(X)}{\bar{g}(X)}\right) = \sum_{i=1}^{N} -\ln(\gamma_i) + \frac{1}{2}\gamma_i^2(\alpha_i + \beta_i)^2 - \frac{1}{2}\alpha_i^2.$$

Next we show that $\left|\ln\left(\frac{g(X)}{\bar{g}(X)}\right)\right|$ is not large with high probability when $X$ follows the probability density of $g$. Consider the following subsets of $X$: $\mathcal{E}_1 = \left\{X\Big|\left|\sum_{i=1}^{N} -\ln(\gamma_i) + \frac{1}{2}(\gamma_i^2 - 1)\alpha_i^2\right| \leq 1\right\}$.
$\mathcal{E}_2 = \left\{X\Big|\left|\sum_{i=1}^{N} \gamma_i^2\alpha_i\beta_i\right| \leq 1\right\}$ and $\mathcal{E}_3 = \left\{X\Big|\frac{1}{2}\sum_{i=1}^{N} \gamma_i^2\beta_i^2 \leq 1\right\}$. When $X$ lies in the intersection of these three sets, $\left|\ln\left(\frac{g(X)}{\bar{g}(X)}\right)\right|$ is guaranteed to be not very large.

Let $\mathbb{P}$ be the probability with respect to density $g$. We will explicitly show that $\mathbb{P}[X \in \mathcal{E}_k] \geq \frac{5}{6}(k = 1, 2, 3)$.

**Lower bound** $\mathbb{P}[X \in \mathcal{E}_1]$ Firstly, we estimate $\sum_{i=1}^{N} \frac{1}{2}(\gamma_i^2 - 1) - \ln(\gamma_i)$. We first prove the following inequality:

$$0 \leq \gamma_i^2 - 1 \leq 2\delta \min\{1, t_i - t_{i-1}\}. \tag{25}$$

Let $t = t_i - t_{i-1}$. Then $\gamma_i^2 = (1+\delta)\frac{1-e^{-2t}}{1-e^{-(2+2\delta)t}}$.

The left hand side of this inequality is because $\Gamma_t \geq \bar{\Gamma}_t$, due to the reason that $e^{-2s} \geq e^{-(2+2\delta)s}$ for all $s \geq 0$ and when $f(x) \geq g(x)$ for any $x \in I$ we have: $\int_{x \in I} f(x)dx \geq \int_{x \in I} g(x)dx$. Now we consider the right hand side of the inequality.

**Case 1:** When $t \geq 1$, we directly use the fact that $1 - e^{-2t} \leq 1 - e^{-(2+2\delta)t}$ and obtain $\gamma_i \leq 1 + \delta$.

**Case 2:** When $t \in (0,1]$, it suffices to show that

$$(1+\delta)(1-e^{-2t}) \leq (1+2\delta t)(1 - e^{-(2+2\delta)t}).$$

Let $h(t) = (1+\delta)(1-e^{-2t}) - (1+2\delta t)(1-e^{-(2+2\delta)t})$, then

$$\begin{aligned} h(t) &= \delta(1-2t) - e^{-2t}[1+\delta - (1+2\delta t)e^{-2\delta t}] \\ &\leq \delta(1 - 2t - e^{-2t}) \\ &\leq 0. \end{aligned}$$

Where for the first inequality we use the relation that $e^{-2\delta t} \leq \frac{1}{1+2\delta t}$. The second inequality is obtained by the relation that $e^{-2t} \geq 1 - 2t$.

Now we bound $\frac{1}{2}(\gamma_i^2 - 1) - \ln(\gamma_i)$. We first show that

$$0 \leq \frac{1}{2}(\gamma_i^2 - 1) - \ln(\gamma_i) \leq \frac{1}{4}(\gamma_i^2 - 1)^2.$$

Let $x = \gamma_i^2 - 1$ and we obtain $\frac{1}{2}(\gamma_i^2 - 1) - \ln(\gamma_i) = \frac{1}{2}[x - \ln(1+x)]$, and the inequality is obtained directly since we have $x \geq \ln(1+x) \geq x - \frac{1}{2}x^2 \ (x \geq 0)$.

Then we can bound $\sum_{i=1}^N \frac{1}{2}(\gamma_i^2 - 1) - \ln(\gamma_i)$ as

$$\begin{aligned} 0 \leq \sum_{i=1}^N \frac{1}{2}(\gamma_i^2 - 1) - \ln(\gamma_i) &\leq \sum_{i=1}^N \frac{1}{4}(\gamma_i^2 - 1)^2 \\ &\leq \sum_{i=1}^N \delta^2 \min(1, (t_i - t_{i-1}))^2 \\ &\leq \sum_{i=1}^N \delta^2(t_i - t_{i-1}) \\ &\leq \delta^2 T \\ &\leq \frac{1}{25}. \end{aligned}$$

Now we bound $\sum_{i=1}^N \frac{1}{2}(\gamma_i^2 - 1)(\alpha_i^2 - 1)$. Notice that this variable has zero mean, so we can bound its variance and then apply Markov inequality to obtain a high probability bound.

At first, consider the variance of $\alpha_i^2 - 1$, denoted as $Var(\alpha_i^2 - 1)$. By noticing that $\alpha_i \sim \mathcal{N}(0,1)$, we can directly calculate that

$$Var(\alpha_i^2 - 1) = \int_{x \in \mathbb{R}} \frac{1}{\sqrt{2\pi}} e^{-\frac{1}{2}x^2}(x^2 - 1)^2 dx = 2.$$

Since all the $\alpha_i$'s are independent, we have:

$$Var\left(\sum_{i=1}^{N}\frac{1}{2}(\gamma_i^2-1)(\alpha_i^2-1)\right) = \sum_{i=1}^{N}\frac{1}{4}(\gamma_i^2-1)^2 Var(\alpha_i^2-1)$$

$$\leq \frac{1}{2}\sum_{i=1}^{N}(\gamma_i^2-1)^2$$

$$\leq 2\delta^2\sum_{i=1}^{N}\min(1, t_i-t_{i-1})^2$$

$$\leq 2\delta^2 T$$

$$\leq \frac{2}{25}.$$

By Markov inequality, we have:

$$\mathbb{P}\left[\left|\sum_{i=1}^{N}\frac{1}{2}(\gamma_i^2-1)(\alpha_i^2-1)\right| \geq \frac{4}{5}\right] \leq Var\left(\sum_{i=1}^{N}\frac{1}{2}(\gamma_i^2-1)(\alpha_i^2-1)\right) / \left(\frac{4}{5}\right)^2 \leq \frac{1}{8}.$$

Finally, for the subset $\mathcal{E}_1 = \left\{X \Big| \left|\sum_{i=1}^{N} -\ln(\gamma_i) + \frac{1}{2}(\gamma_i^2-1)\alpha_i^2\right| \leq 1\right\}$, we have:

$$\mathbb{P}\left[x \in \mathcal{E}_1\right] \geq 1 - \mathbb{P}\left[\left|\sum_{i=1}^{N}\frac{1}{2}(\gamma_i^2-1)(\alpha_i^2-1)\right| \geq \frac{4}{5}\right] \geq \frac{7}{8}.$$

**Lower bound** $\mathbb{P}[X \in \mathcal{E}_2]$    Since all the $\alpha_i$'s are independent, and $\alpha_i$ is independent of $\{\beta_1, ..., \beta_i\}$ and $\{\gamma_1, ..., \gamma_N\}$, we obtain that

$$\mathbb{E}\left[\left(\sum_{i=1}^{N}\gamma_i^2\alpha_i\beta_i\right)^2\right] = \mathbb{E}\left[\sum_{i=1}^{N}(\gamma_i^2\alpha_i\beta_i)^2\right]$$

$$= \mathbb{E}\left[\sum_{i=1}^{N}(\gamma_i^2\beta_i)^2\right]$$

$$= \sum_{i=1}^{N}\mathbb{E}\left[(\gamma_i^2\beta_i)^2\right].$$

We have shown that $\gamma_i^2 \leq 1 + 2\delta$. Then for $T \geq 1$ we have: $\gamma_i^4 \leq (1+\frac{2}{5})^2 \leq 2$. Therefore, we obtain:

$$\mathbb{E}\left[\left(\sum_{i=1}^{N}\gamma_i^2\alpha_i\beta_i\right)^2\right] \leq 2\sum_{i=1}^{N}\mathbb{E}\left[\beta_i^2\right].$$

Now we upper bound $\mathbb{E}\left[\beta_i^2\right]$, where $\beta_i = \sqrt{\frac{1}{\Gamma(t_i-t_{i-1})}}(e^{-(t_i-t_{i-1})} - e^{-(1+\delta)(t_i-t_{i-1})})X_{t_{i-1}}$

Firstly, we show that

$$\sqrt{\frac{1}{\Gamma(t_i-t_{i-1})}}(e^{-(t_i-t_{i-1})} - e^{-(1+\delta)(t_i-t_{i-1})}) \leq \delta\sqrt{t_i-t_{i-1}}. \tag{26}$$

Again denote $t = t_i - t_{i-1}$. By using $\Gamma_t = \frac{1}{2}(1 - e^{-2t})$, it suffices to show that

$$e^{-t} - e^{-(1+\delta)t} \leq \delta\sqrt{\frac{1}{2}t(1 - e^{-2t})}.$$

By multiplying $e^t$ on both sides, the inequality is equivalent to

$$1 - e^{-\delta t} \leq \delta\sqrt{\frac{1}{2}t(e^{2t} - 1)}.$$

This is true since $e^{-\delta t} \geq 1 - \delta t$, and $e^{2t} \geq 1 + 2t$, implying that

$$1 - e^{-\delta t} \leq \delta t \leq \delta\sqrt{\frac{1}{2}t(e^{2t} - 1)}.$$

With this result, we can upper bound $2\sum_{i=1}^{N} \mathbb{E}\left[\beta_i^2\right]$ by

$$2\sum_{i=1}^{N} \mathbb{E}\left[\beta_i^2\right] \leq \sum_{i=1}^{N} 2\delta^2(t_i - t_{i-1})\mathbb{E}\left[X_{t_{i-1}}^2\right].$$

Finally, since $X_t \sim \mathcal{N}(0, \Gamma(t))$, for all $t \geq 0$,

$$\mathbb{E}\left[X_t^2\right] = \Gamma_t = \frac{1}{2}(1 - e^{-2t}) \leq 1.$$

Therefore, we obtain

$$\mathbb{E}\left[\left(\sum_{i=1}^{N} \gamma_i^2 \alpha_i \beta_i\right)^2\right] \leq 2\sum_{i=1}^{N} \mathbb{E}\left[\beta_i^2\right] \leq \sum_{i=1}^{N} 2\delta^2(t_i - t_{i-1})\mathbb{E}\left[X_{t_{i-1}}^2\right] \leq \sum_{i=1}^{N} 2\delta^2(t_i - t_{i-1}) = 2T\delta^2 = \frac{2}{25}$$

Again by using Markov inequality, we obtain:

$$\mathbb{P}\left[|\sum_{i=1}^{N} \gamma_i^2 \alpha_i \beta_i| > 1\right] \leq \frac{2}{25}.$$

Which follows that

$$\mathbb{P}\left[X \in \mathcal{E}_2\right] = 1 - \mathbb{P}\left[|\sum_{i=1}^{N} \gamma_i^2 \alpha_i \beta_i| \geq 1\right] \geq \frac{23}{25}.$$

**Lower bound $\mathbb{P}[X \in \mathcal{E}_3]$**   We have shown that $\gamma_i^2 \leq 2, \forall i$ and $\sum_{i=1}^{N} \mathbb{E}\left[\beta_i^2\right] \leq \delta^2 T$. Therefore,

$$\mathbb{E}\left[\frac{1}{2}\sum_{i=1}^{N} \gamma_i^2 \beta_i^2\right] \leq \delta^2 T \leq \frac{2}{25}.$$

And we also have

$$\mathbb{P}\left[X \in \mathcal{E}_3\right] = 1 - \mathbb{P}\left[\frac{1}{2}\sum_{i=1}^{N} \gamma_i^2 \beta_i^2 > 1\right] \geq \frac{23}{25}.$$

Now we come back to prove the theorem. With lower bounds of $\mathbb{P}[X \in \mathcal{E}_1], \mathbb{P}[X \in \mathcal{E}_2], \mathbb{P}[X \in \mathcal{E}_3]$, we have

$$\mathbb{P}\left[X \in \mathcal{E}_1 \cap \mathcal{E}_2 \cap \mathcal{E}_3\right] \geq 1 - (1 - \mathbb{P}[X \in \mathcal{E}_1]) - (1 - \mathbb{P}[X \in \mathcal{E}_2]) - (1 - \mathbb{P}[X \in \mathcal{E}_3]) \geq \frac{1}{2}.$$

With this bound, we have:

$$\mathbb{E}_{X \sim g}\left[1\left(|\phi(X) - A| \geq \frac{1}{10\sqrt{T}}\right)\right] + \mathbb{E}_{X \sim \bar{g}}\left[1\left(|\phi(X) - \bar{A}| \geq \frac{1}{10\sqrt{T}}\right)\right]$$

$$\geq \int_{X \in \mathcal{E}_1 \cap \mathcal{E}_2 \cap \mathcal{E}_3} g(X) \mathbb{E}\left[1\left(\|\phi(X) - A\| \geq \frac{1}{10\sqrt{T}}\right) \mid X\right] + \bar{g}(X) \mathbb{E}\left[1\left(\|\phi(X) - \bar{A}\| \geq \frac{1}{10\sqrt{T}}\right) \mid X\right] dX$$

$$\geq \int_{X \in \mathcal{E}_1 \cap \mathcal{E}_2 \cap \mathcal{E}_3} \min\{g(X), \bar{g}(X)\} dX$$

$$\geq \int_{X \in \mathcal{E}_1 \cap \mathcal{E}_2 \cap \mathcal{E}_3} \frac{1}{e^3} g(X) dX$$

$$\geq \frac{1}{2e^3}.$$

Where the second inequality is because $\|\phi(X) - A\| + \|\phi(X) - \bar{A}\| \geq \|A - \bar{A}\| = \frac{1}{5\sqrt{T}}$ so we cannot have both $\|\phi(X) - A\| \leq \frac{1}{10\sqrt{T}}$ and $\|\phi(X) - \bar{A}\| \leq \frac{1}{10\sqrt{T}}$. The third inequality is because for any $X \in \mathcal{E}_1 \cap \mathcal{E}_2 \cap \mathcal{E}_3$, we have

$$\left|\ln \frac{g(X)}{\bar{g}(X)}\right| = \left|\sum_{i=1}^{N} -\ln(\gamma_i) + \frac{1}{2}\gamma_i^2(\alpha_i + \beta_i)^2 - \frac{1}{2}\alpha_i^2\right|$$

$$\leq \left|\sum_{i=1}^{N} -\ln(\gamma_i) + \frac{1}{2}(\gamma_i^2 - 1)\alpha_i^2\right|$$

$$+ \left|\sum_{i=1}^{N} \gamma_i^2 \alpha_i \beta_i\right|$$

$$+ \frac{1}{2}\sum_{i=1}^{N} \gamma_i^2 \beta_i^2$$

$$\leq 3,$$

implying that $\bar{g}(X) \geq \frac{1}{e^3} g(X)$.

Therefore, we have:

$$\max\left\{\mathbb{P}_{X \sim g}\left[|\phi(X) - A| \geq \frac{1}{10\sqrt{T}}\right], \mathbb{P}_{X \sim \bar{g}}\left[|\phi(X) - \bar{A}| \geq \frac{1}{10\sqrt{T}}\right]\right\} \geq \frac{1}{4e^3}.$$

This means that for any algorithm, it cannot achieve $\frac{1}{10\sqrt{T}}$ estimation error with success probability $1 - \frac{1}{4e^3}$ for at least one of the systems controlled by $(A, 0)$ and $(\bar{A}, 0)$.

$\square$

## B  Regret Analysis

Having demonstrated the results of system identification for continuous-time linear systems, we leverage these findings to establish upper bounds on the regret for Algorithm 3. Elaborations on the details will be presented in the subsequent sections.

## B.1 Convergence of $P$ and the Estimation Error of $K$

In this section we provide the following Lemma 15, along with its proof, which shows that $\|P - P_*\|$ converges at the same speed as $\|\hat{A} - A\| + \|\hat{B} - B\|$.

**Lemma 15.** *There exist constants $\epsilon_0 > 0$ and $C_2 > 0$ such that as long as $\|\hat{A} - A\|, \|\hat{B} - B\| \leq \epsilon$ for some $0 < \epsilon < \epsilon_0$, with $P$ obtained from (5) we have:*

$$\|P - P_*\| \leq C_2 \epsilon. \tag{27}$$

Recall that the optimal dynamic is $K_* = -R^{-1}B^{\mathrm{T}}P_*$ with $P_*$ obtained from equation (5). Now we consider the distance between it and the sub-optimal dynamic $\bar{K} = -R^{-1}B^{\mathrm{T}}P$ with $P$ obtained from (5) with $(\hat{A}, \hat{B})$. Denote $\Delta A = \hat{A} - A$ and $\Delta B = \hat{B} - B$, along with $\|\Delta A\|, \|\Delta B\| \leq \epsilon$ where $\epsilon \in [0, \epsilon_0]$ with some $\epsilon_0$ determined later. We establish the proof by constructing a sequence of matrices $(P_k)_{k \geq 0}$, and we will prove that such sequence converges to the unique symmetric solution $P$ satisfying

$$P\hat{B}R^{-1}\hat{B}^{\mathrm{T}}P - \hat{A}^{\mathrm{T}}P - P\hat{A} - Q = 0.$$

At first we introduce a solution of a particular kind of matrix equation [19].

**Lemma 16.** *Suppose $A$ satisfies $\alpha(A) = \max\{\Re(\lambda_i)|\lambda_i \in \lambda(A)\} < 0$. $Q$ is a symmetric matrix. Consider such a function*

$$A^{\mathrm{T}}X + XA + Q = 0. \tag{28}$$

*Then, the unique symmetric solution $X$ of this equation can be expressed as:*

$$X = \int_{t \geq 0} e^{A^{\mathrm{T}}t} Q e^{At} dt. \tag{29}$$

Now we consider the relation between $P$ and $P_*$. The core is iteratively constructing a sequence of matrices $P_k$ such that $P_0 = P_*$ and $\lim_{k \to +\infty} P_k = P$. Such matrices follows the relation $P_{k+1} = P_k + \Delta P_k$ where $\Delta P_k$ converges rapidly. As for the starting case, consider the expansion

$$
\begin{aligned}
&(P_* + \Delta P)(B + \Delta B)R^{-1}(B + \Delta B)^{\mathrm{T}}(P_* + \Delta P) \\
&- (A + \Delta A)^{\mathrm{T}}(P_* + \Delta P) - (P_* + \Delta P)(A + \Delta A) - Q \\
&= \left[ (B + \Delta B)R^{-1}(B + \Delta B)^{\mathrm{T}}P_* - A - \Delta A \right]^{\mathrm{T}} \Delta P \\
&+ \Delta P \left[ (B + \Delta B)R^{-1}(B + \Delta B)^{\mathrm{T}}P_* - A - \Delta A \right] \\
&+ \left[ P_* B R^{-1}B^{\mathrm{T}}P_* - A^{\mathrm{T}}P_* - P_* A - Q \right] + P_* \left[ \Delta B \left( R^{-1}(B + \Delta B)^{\mathrm{T}} \right) + B R^{-1}\Delta B \right] P_* \\
&+ \Delta P (B + \Delta B)R^{-1}(B + \Delta B)^{\mathrm{T}} \Delta P.
\end{aligned}
$$

Define

$$
\begin{aligned}
A_0 &= A + \Delta A - (B + \Delta B)R^{-1}(B + \Delta B)^{\mathrm{T}}P_*, \\
F_0 &= -P_* \left[ \Delta B \left( R^{-1}(B + \Delta B)^{\mathrm{T}} \right) + B R^{-1}\Delta B \right] P_*.
\end{aligned}
$$

We set $\Delta P_0$ be a solution of

$$A_0^{\mathrm{T}}\Delta P_0 + \Delta P_0 A_0 + F_0 = 0.$$

which satisfies that (see Lemma 16)

$$\Delta P_0 = \int_{t \geq 0} e^{A_0^{\mathrm{T}}t} F_0 e^{A_0 t} dt,$$

$$\|\Delta P_0\| \leq \int_{t \geq 0} e^{2\alpha(A_0)t} \|F_0\| dt = \frac{1}{-2\alpha(A_0)} \|F_0\| \leq \frac{1}{-\alpha(A_0)} \|P_*\|^2 (\|BR^{-1}\|\epsilon + \|R^{-1}\|\epsilon^2).$$

This $\Delta P_0$ also satisfies

$$
\begin{aligned}
&(P_* + \Delta P_0)(B + \Delta B)R^{-1}(B + \Delta B)^{\mathrm{T}}(P_* + \Delta P_0) \\
&- (A + \Delta A)^{\mathrm{T}}(P_* + \Delta P) - (P_* + \Delta P)(A + \Delta A) - Q \\
&= \Delta P_0(B + \Delta B)R^{-1}(B + \Delta B)^{\mathrm{T}}\Delta P_0 \,.
\end{aligned}
$$

An important thing is to guarantee that $A_0$ is stable, and $|\alpha(A_0)|$ can not be too closed to zero. For any $\epsilon_1 \in (0,1)$ and $C_1 = \|R^{-1}\|\|P_*\| + 1 + 2\|BR^{-1}\|\|P_*\|$, as long as $\epsilon \le \epsilon_1$, $\|A_0 - (A - BR^{-1}B^{\mathrm{T}}P_*)\| \le C_1\epsilon$. Furthermore, there exists $\epsilon_2 > 0$ such that if $\|X - (A - R^{-1}B^{\mathrm{T}}P_*)\| \le \epsilon_2$, then $\alpha(X) \le \frac{1}{2}\alpha(A - R^{-1}B^{\mathrm{T}}P_*)$(the work of [30] shows this result). We can further let this $\epsilon_2$ satisfies that, as long as $\|\Delta A\|, \|\Delta B\|, \|\Delta P\| \le \epsilon_2$, we always have:

$$
\alpha\left(A + \Delta(A) - (B + \Delta B)R^{-1}(B + \Delta B)^{\mathrm{T}}(P_* + \Delta P)\right) \le \frac{1}{2}\alpha(A - BR^{-1}B^{\mathrm{T}}P_*) \,. \tag{30}
$$

Now we additionally set $\epsilon_1$ satisfying $\epsilon_1 \le \frac{1}{2C_1}\epsilon_2$ and $\|R^{-1}\|\epsilon_1 \le 1$, then for all $\epsilon \le \epsilon_1$,

$$
\|\Delta P_0\| \le \frac{2}{-\alpha(A - BR^{-1}B^{\mathrm{T}}P_*)}\|P_*\|^2(1 + \|BR^{-1}\|)\epsilon \,.
$$

Denote $P_1 = P_0 + \Delta P_0$, $C_2 = \frac{2}{-\alpha(A - BR^{-1}B^{\mathrm{T}}P_*)}\|P_*\|^2(1 + \|BR^{-1}\|)$, and set some constant $C_3$ satisfying $C_3 \ge \|BR^{-1}B^{\mathrm{T}}\| + 2\|BR^{-1}\| + \|R^{-1}\|$. We then inductively define $P_{k+1}$ and $\Delta P_k$ ($k \ge 1$). For defined $\Delta P_{k-1}$, we set $P_k = P_{k-1} + \Delta P_{k-1}$, which satisfies

$$
\begin{aligned}
&P_k(B + \Delta B)R^{-1}(B + \Delta B)^{\mathrm{T}}P_k - (A + \Delta A)^{\mathrm{T}}P_k - P_k(A + \Delta(A)) - Q \\
&= \Delta P_{k-1}(B + \Delta B)R^{-1}(B + \Delta B)^{\mathrm{T}}\Delta P_{k-1} \,.
\end{aligned}
$$

Then we denote $A_k = A + \Delta A - (B + \Delta B)R^{-1}(B + \Delta B)^{\mathrm{T}}P_k$, and set $\Delta P_k$ satisfying:

$$
A_k^{\mathrm{T}}\Delta P_k + \Delta P_k A_k = \Delta P_{k-1}(B + \Delta B)R^{-1}(B + \Delta B)^{\mathrm{T}}\Delta P_{k-1} \,.
$$

By the hypothesis of $\epsilon_2$, as long as $\|P_k - P_*\| \le \epsilon_2$, we have $\alpha(A_k) \ge \frac{1}{2}\alpha(A - BR^{-1}B^{\mathrm{T}}P_*)$. By using (29) we obtain that $\|\Delta P_k\| \le C_4\|\Delta P_{k-1}\|^2$, where $C_4 = \frac{2}{-\alpha(A - BR^{-1}B^{\mathrm{T}}P_*)}C_3$. Now if we define $P_{k+1} = P_k + \Delta P_k$, $P_{k+1}$ also satisfies:

$$
\begin{aligned}
&P_{k+1}(B + \Delta B)R^{-1}(B + \Delta B)^{\mathrm{T}}P_{k+1} - (A + \Delta A)^{\mathrm{T}}P_{k+1} - P_{k+1}(A + \Delta(A)) - Q \\
&= \Delta P_k(B + \Delta B)R^{-1}(B + \Delta B)^{\mathrm{T}}\Delta P_k \,,
\end{aligned}
$$

Then these sequences $\Delta P_k$ and $P_k$ are well defined, along with the relation that $P_{k+1} = P_k + \Delta P_k$. Furthermore, when $\|P_k - P_*\| \le \epsilon_2$, we have $\|\Delta P_{k+1}\| \le C_4\|\Delta P_k\|^2$. Note that for the base case we have $\|\Delta P_0\| \le C_2\epsilon$.

Finally, it remains to constrain $\|P_k - P_*\|$. By choosing $\epsilon \le \min(\frac{1}{2C_2C_4}, \frac{1}{2C_2}\epsilon_2, 1)$, we obtain $\|\Delta P_0\| \le C_2\epsilon$. We can also see that if for all $0 \le k \le m$, $\|\Delta P_k\| \le 2^{-k}C_2\epsilon$, then $\|P_m - P_*\| \le 2(1 - 2^{-m+1})C_2\epsilon \le \epsilon_2$ so that $\|\Delta P_{m+1}\| \le C_4\|\Delta P_m\|^2 \le 2^{-m-1}C_2\epsilon$. So by induction we see that $\|\Delta P_k\| \le 2^{-k}C_2\epsilon$ for any $k$.

On the other hand, since $\|\Delta P_k\| \le 2^{-k}\|\Delta P_0\|$, $\lim_{k \to +\infty} P_k = P_\infty$ exists, and such $P_\infty$ is the unique symmetric solution of

$$
P(B + \Delta B)R^{-1}(B + \Delta B)^{\mathrm{T}}P - (A + \Delta A)^{\mathrm{T}}P - P(A + \Delta(A)) - Q = 0 \,,
$$

such that $(A + \Delta A) - (B + \Delta B)R^{-1}(B + \Delta B)^{\mathrm{T}}P$ is stable (recall the stable margin in (30), which implies that $(A + \Delta A) - (B + \Delta B)R^{-1}(B + \Delta B)^{\mathrm{T}}P_\infty$ is stable).

So $P_\infty$ is exactly $P$, satisfying $\|P - P_*\| \le 2C_2\epsilon$.

Therefore, we conclude that there exists some $\epsilon_0 > 0$ and constant $C$, both depending on $A, B, K, d, p$ such that for any $\epsilon \in [0, \epsilon_0]$, $\|P - P_*\| \le C\epsilon$ as long as $\|\hat{A} - A\|, \|\hat{B} - B\| \le \epsilon$.

Then we apply our results for system identification to establish an upper bound for $\|\bar{K} - K_*\|$.

Based on Lemma 15, fix constant $\epsilon_1 > 0$ and constant $C_1 \geq 0$ so that we have $\|P - P_*\| \leq C_1 \left( \|\hat{A} - A\| + \|\hat{B} - B\| \right)$ whenever $\|\hat{A} - A\| + \|\hat{B} - B\| \leq \epsilon_1$

We set $C_2 \geq 1$ be two times the constant $C$ in Lemma 9, and obtain that, when $\log^2(1/\delta) \leq \frac{T^{1/2}}{C_2}$ and $T^{1/2} \geq C_2\|X_0\|_2^2$, we have:

$$\mathbb{P}\left[ \|\hat{A} - A\| + \|\hat{B} - B\| \leq 2C_2\sqrt{\frac{\log(1/\delta)}{T^{1/2}}} \right] \geq 1 - \delta \,.$$

Then, for $\log(1/\delta) \leq \min\left\{ \frac{T\epsilon_1^2}{4C_2^2}, \frac{T^{1/4}}{C_2^{1/2}} \right\} \leq \frac{T^{1/4}\epsilon_1^2}{4C_2^2}$, we have:

$$\mathbb{P}\left[ \|P - P_*\| \leq 2C_1C_2\sqrt{\frac{\log(1/\delta)}{T^{1/2}}} \right] \geq 1 - \delta \,. \tag{31}$$

Finally, since $\bar{K} = -R^{-1}(\hat{B})^{\mathrm{T}}P$, $K_* = -R^{-1}B^{\mathrm{T}}P_*$, we have:

$$\|\bar{K} - K_*\| \leq \|R^{-1}\| \left[ \|\hat{B} - B\|\|P\| + \|B\|\|P - P_*\| \right] \,.$$

We can reset $C_1$ such that $\|\bar{K} - K_*\| \leq C_1 \left( \|\hat{A} - A\| + \|\hat{B} - B\| \right)$ whenever $\|\hat{A} - A\| + \|\hat{B} - B\| \leq \epsilon_1$, and combine this with (31), we have: for any $\log(1/\delta) \leq \frac{T^{1/4}\epsilon_1^2}{4C_2^2}$

$$\mathbb{P}\left[ \|\bar{K} - K_*\| \leq 2C_1C_2\sqrt{\frac{\log(1/\delta)}{T^{1/2}}} \right] \geq 1 - \delta \,. \tag{32}$$

With this probability bound on $\|\bar{K} - K_*\|$, we can further upper bound the regret, shown in the following part.

### B.2  Key Lemmas

We first upper bound the radius of a single trajectory with stable controller, for which we introduce and provide a proof for the following lemma:

**Lemma 17.** *Consider the continuous system $dX_t = AX_t dt + dW_t$ such that $\alpha(A) < 0$ where $\alpha(A)$ is the largest real component of $A$ and $W$ is a standard Brownian noise. Then, w.p. at least $1 - \delta$:*

$$\sup_{0 \leq t \leq T} \left( \|X_t\|_2 - e^{\alpha(A)t}\|X_0\|_2 \right) \leq C\sqrt{d \log((1+T)/\delta)} \,.$$

Then we concentrate on how the error $\|P - P_*\|$ will influence the regret during the exploitation phase. For a dynamic $U$ with $\alpha(A + BU) < 0$, we define a cost function:

$$cost(U) = \mathrm{tr}\left( \int_{t \geq 0} (e^{(A+BU)t})^{\mathrm{T}}(Q + U^{\mathrm{T}}RU)e^{(A+BU)t}dt \right) \,.$$

The convergence rate of this cost function is stated in the following lemma:

**Lemma 18.** *Let $U_*$ minimize $cost(U)$. Then, there exists $\epsilon_0 \geq 0$ such that for any $\|\Delta U\| = 1$ and $\epsilon \in [0, \epsilon_0]$, we have:*

$$cost(U_* + \epsilon\Delta U) - cost(U_*) \leq C_1\epsilon^2 \,.$$

The above result shows the average cost per unit time when applying fixed controller for infinite time.

Then we further consider the case when the running time is finite. We derive the following lemma:

**Lemma 19.** *Let $U_*$ follows the same definition as in Lemma 18. Then, for some $\epsilon > 0$, there exist constants $C_2$ and $C_3$ (independent of $U$) such that for all $T > 0$ and any $U$ such that $\|U - U_*\| \leq \epsilon$,*

$$|J_T - cost(U)T| \leq C_2\|x\|_2^2 + C_3 \,.$$

*Here $J_T$ is the expected cost of the policy that takes action by $U_t = UX_t$ ($t \in [0, T]$), with initial state $X_0 = x$.*

With this lemma, by definition of $U_*$, we actually have $U_* = K_*$, where $K_* = -R^{-1}B^{\mathrm{T}}P_*$ and $P_*$ is the solution of (4). Since such $C_2, C_3$ also satisfy:

$$|J_T^* - cost(U_*)T| \leq C_2\|x\|_2^2 + C_3\,,$$

so it follows that

$$R_T = J_T - J_T^* \leq 2C_2\|x\|_2^2 + 2C_3\,. \tag{33}$$

### B.3 Proof of Lemma 17

We first upper bound the radius of a single trajectory with stable controller, for which we introduce and provide a proof for the following lemma:

**Lemma 17.** *Consider the continuous system $dX_t = AX_t dt + dW_t$ such that $\alpha(A) < 0$ where $\alpha(A)$ is the largest real component of $A$ and $W$ is a standard Brownian noise. Then, w.p. at least $1 - \delta$:*

$$\sup_{0 \leq t \leq T}\left(\|X_t\|_2 - e^{\alpha(A)t}\|X_0\|_2\right) \leq C\sqrt{d\log((1+T)/\delta)}\,.$$

*Proof.* The trajectory $X_t$ with differential equation $dX_t = AX_t + dW_t$ can be derived as

$$X_t = e^{At}X_0 + \int_{s=0}^{t} e^{A(t-s)}dW_t\,.$$

Lemma 7 tells that when $A$ is stable, $\left\|e^{At}X_0\right\|_2 \leq e^{\alpha(A)t}\|X_0\|_2$. So it suffices to show that

$$\mathbb{P}\left[\sup_{0 \leq t \leq T}\left\|\int_{s=0}^{t} e^{A(t-s)}dW_t\right\|_2 \geq C\sqrt{d\log(1+T)/\delta}\right] \leq \delta\,.$$

Let $T = T_0 h$ with $T_0$ be an integer. We first consider the set of points $\{X_{kh}\}$. Denote $w_k := \int_{t=0}^{kh} e^{A(kh-t)}dW_t$, then $w_k \sim \mathcal{N}(0, \Sigma_k)$ with $\Sigma_k = \int_{t=0}^{kh} e^{At}e^{A^{\mathrm{T}}t}dt$. This $\Sigma_h$ also satisfies

$$\|\Sigma_k\| \leq \int_{t=0}^{h}\left\|e^{At}\right\|^2 dt \leq \int_{t=0}^{kh} e^{2\alpha(A)t}dt \leq \frac{1}{2|\alpha(A)|}\,.$$

Which follows that $\sup_{0 \leq k \leq T_0}\|w_k\|_2 \leq 2\sqrt{\frac{d}{|\alpha(A)|}\log((1+T_0)/\delta)}$, w.p. at least $1 - \delta$.

Next we consider any $X_{kh+t}$ with $t \in [0, h]$. Bounding such terms requires the Doob's martingale inequality [11], stated as in Lemma 20. We denote $x_t^k = \int_{s=0}^{t} e^{A(t-s)}dW_{kh+s}ds$ with corresponding filtration $\mathcal{F}_t$. We also define $Z_t^k := e^{\lambda\left\|e^{-At}x_t^k\right\|_2^2}$ with $\lambda \geq 0$. Then $Z_t^k$ is a submartingale under the filtration $\mathcal{F}_t$, since for any $t \geq s$,

$$\mathbb{E}\left[Z_t^k|\mathcal{F}_s\right] = \mathbb{E}\left[\exp\left(\lambda\left\|e^{-As}x_s^k + \int_{t_1=s}^{t} e^{-At_1}dW_{kh+t_1}\right\|_2^2\right)\bigg|x_s^k\right] \geq e^{\lambda\left\|e^{-As}x_s^k\right\|_2^2} = Z_s^k\,.$$

Where we notice that $\mathbb{E}\left[\left\|e^{-As}x_s^k + \int_{t_1=s}^{t} e^{-At_1}dW_{kh+t_1}\right\|_2^2\bigg|x_s^k\right] \geq \left\|e^{-As}x_s^k\right\|_2^2$, and apply Jensen's inequality on the non-decreasing convex function $f(x) = e^{\lambda x}$ to obtain the above inequality.

Now we apply Lemma 20 and get

$$\mathbb{P}\left[\sup_{t \in [0,h]}\left\|e^{-At}x_t^k\right\|_2 \geq C\right] \leq e^{-\lambda C^2}\mathbb{E}[Z_h^k]\,. \tag{34}$$

We next estimate $\mathbb{E}(Z_h^k)$. Since $e^{-Ah}x_h^k = \int_{t=0}^{h} e^{-At}dW_{kh+t}$, we obtain that $e^{-Ah}x_h^k \sim \mathcal{N}(0, \bar{\Sigma})$, where

$$\bar{\Sigma} = \int_{t=0}^{h} e^{-At}e^{-A^{\mathrm{T}}t}dt\,.$$

By setting $\lambda = \frac{1}{4\|\bar{\Sigma}\|}$, it can be computed that

$$\mathbb{E}\left[e^{\lambda\|e^{-Ah}x_h^k\|_2^2}\right] = \int_{x\in\mathbb{R}^d} \frac{1}{(2\pi)^{d/2}\sqrt{\det(\bar{\Sigma})}}e^{-\frac{1}{2}x^{\mathrm{T}}\Sigma_1^{-1}x}e^{\lambda x^{\mathrm{T}}I_d x}dx$$

$$= \sqrt{\frac{1}{\det(\bar{\Sigma})\det(\Sigma_1^{-1}-2\lambda I_d)}}$$

$$= \sqrt{\frac{1}{\det(I_d-2\lambda\bar{\Sigma})}}$$

$$\leq 2^{d/2},$$

where the last inequality is because $I_d - 2\lambda\bar{\Sigma} \succeq \frac{1}{2}I_d$.

We combine this result with (34) and obtain:

$$\mathbb{P}\left[\sup_{0\leq k\leq T_0-1, 0\leq t\leq h}\|x_t^k\|_2 \geq 2e^{\|A\|h}\|\bar{\Sigma}\|^{1/2}\sqrt{\log(2^{d/2}T_0/\delta)}\right]$$

$$\leq \sum_{k=0}^{T_0-1}\mathbb{P}\left[\sup_{t\in[0,h]}Z_t^k \geq 2^{d/2}\frac{T_0}{\delta}\right]$$

$$\leq \sum_{k=0}^{T_0-1}\mathbb{P}\left[\sup_{t\in[0,h]}Z_t^k \geq \frac{T_0}{\delta}\mathbb{E}(Z_h^k)\right]$$

$$\leq \delta.$$

Finally, since $X_{kh+t} = e^{A(kh+t)}X_0 + e^{At}w_k + x_t^k$, it follows that

$$\|X_{kh+t}\|_2 \leq \left\|e^{A(kh+t)}X_0\right\|_2 + \left\|e^{At}w_k\right\|_2 + \left\|x_t^k\right\|_2$$

$$\leq e^{\alpha(A)(kh+t)}\|X_0\|_2 + \|w_k\|_2 + \left\|x_t^k\right\|_2.$$

By applying union bound on $\|w_k\|_2$ and $\left\|x_t^k\right\|_2$ we finally obtain Lemma 17. $\qquad\square$

**Lemma 20** (Doob's martingale inequality). *Let $X_1, \ldots, X_n$ be a discrete-time submartingale relative to a filtration $\mathcal{F}_1, \ldots, \mathcal{F}_n$ of the underlying probability space, which is to say:*

$$X_i \leq \mathbb{E}\left[X_{i+1} \mid \mathcal{F}_i\right].$$

*The submartingale inequality says that*

$$\mathbb{P}\left[\max_{1\leq i\leq n} X_i \geq C\right] \leq \frac{\mathbb{E}\left[\max\left(X_n, 0\right)\right]}{C}$$

*for any positive number $C$.*

*Moreover, let $X_t$ be a submartingale indexed by an interval $[0, \mathrm{T}]$ of real numbers, relative to a filtration $F_t$ of the underlying probability space, which is to say:*

$$X_s \leq \mathrm{E}\left[X_t \mid \mathcal{F}_s\right]$$

*for all $s < t$. The submartingale inequality says that if the sample paths of the martingale are almost-surely right-continuous, then*

$$\mathbb{P}\left[\sup_{0\leq t\leq T} X_t \geq C\right] \leq \frac{\mathbb{E}\left[\max\left(X_T, 0\right)\right]}{C}$$

*for any positive number $C$.*

## B.4 Proof of Lemma 18

In this section, we proof Lemma 18 which refers to the convergence rate of the cost function:

**Lemma 18.** *Let $U_*$ minimize $cost(U)$. Then, there exists $\epsilon_0 \geq 0$ such that for any $\|\Delta U\| = 1$ and $\epsilon \in [0, \epsilon_0]$, we have:*

$$cost(U_* + \epsilon \Delta U) - cost(U_*) \leq C_1 \epsilon^2 .$$

*Proof.* For any $\|\Delta U\| = 1$ and $\epsilon > 0$, consider $U = U_* + \epsilon \Delta U$, we show that as $\epsilon \to 0$, there exists $V \in \mathbb{R}^d$ such that $\text{tr}(V) = 0$, and

$$\int_{t \geq 0} e^{(A+BU)^{\mathsf{T}}t}(Q + U^{\mathsf{T}}RU)e^{(A+BU)t}dt - \int_{t \geq 0} e^{(A+BU_*)^{\mathsf{T}}t}(Q + U_*^{\mathsf{T}}RU_*)e^{(A+BU_*)t}dt$$
$$= \epsilon V + \mathcal{O}(\epsilon^2).$$

Let $D(\epsilon, t) = e^{(A+B(U_*+\epsilon \Delta U))t} - e^{(A+BU_*)t}$. The most important intuition is that $D(\epsilon, t)$ can be represented by the form of $D(\epsilon, t) = \epsilon D_1(t) + \epsilon^2 D_2(\epsilon, t)$, where $D_1(t)$ does not depend on $\epsilon$, and the residual $D_2(\epsilon, t)$ can be well bounded. Now we find such $D_1(t)$ and upper bound $\|D_2(\epsilon, t)\|$. For $t \leq t_0 = \frac{1}{\max\{\|A+BU_*\|, \|B\|\}}$ and $\epsilon < 1$, the Taylor expansion of $e^{(A+B(U_*+\epsilon \Delta U))t}$ can be represented as follows:

$$D(\epsilon, t) = \sum_{k \geq 1} \frac{1}{k!} \left[ (A + BU_* + \epsilon B\Delta U)^k t^k - (A + BU_*)^k t^k \right]$$

$$= \sum_{k \geq 1} \frac{1}{k!} \left[ \left( \sum_{i=0}^{k-1}(A + BU_*)^i (B\Delta U_*)(A + BU_*)^{k-1-i} \right) \epsilon + D_1(\epsilon, k)\epsilon^2 \right] t^k ,$$

where $D_1(\epsilon, k)$ is the residual of $(A + BU + \epsilon B\Delta U)^k - (A + BU)^k$ with order at least $\epsilon^2$. This sequence of matrices are expressed and bounded as follows.

$$D_1(k, \epsilon) = \sum_{i=2}^{k} \epsilon^i \sum_{j_1+...+j_{i+1}=k-i} (A + BU_*)^{j_1}(B\Delta U)(A + BU_*)^{j_2}(B\Delta U)...(A + BU_*)^{j_{i+1}} ,$$

$$\|D_1(k, \epsilon)\| \leq \sum_{i=2}^{k} \frac{k!}{i!(k-i)!} \|A + BU_*\|^{k-i} \|B\|^i \epsilon^{i-2} .$$

Thus we have:

$$\left\| \sum_{k \geq 1} \frac{t^k}{k!} D_1(k, \epsilon) \right\| \leq \sum_{k \geq 2} \sum_{i \geq 2} \frac{1}{i!(k-i)!} \leq 4 .$$

Define $E(t)$ and $E_1(\epsilon, t)$ as follows: for $0 \leq t \leq t_0$, let

$$E(t) = \sum_{k \geq 1} \frac{t^k}{k!} \sum_{i=0}^{k-1}(A + BU_*)^i (B\Delta U_*)(A + BU_*)^{k-1-i}, \quad E_1(\epsilon, t) = \sum_{k \geq 1} \frac{t^k}{k!} D_1(k, \epsilon) ,$$

and for $t \in [\frac{1}{2}t_0, t_0]$, $l \geq 1$, we inductively define $E(2^l t)$ and $E_1(2^l t)$ as follows:

$$E(2^l t) = e^{(A+BU_*)2^{l-1}t} E(2^{l-1}t) + E(2^{l-1}t)e^{(A+BU_*)2^{l-1}t} ,$$

$$E_1(\epsilon, 2^l t) = e^{(A+BU_*)2^{l-1}t} E_1(\epsilon, 2^{l-1}t) + E_1(\epsilon, 2^{l-1}t)e^{(A+BU_*)2^{l-1}t}$$
$$+ \left( E(2^{l-1}t) + \epsilon E_1(\epsilon, 2^{l-1}t) \right)^2 .$$

Then we have the relation that $e^{(A+BU_*+B\Delta U)t} - e^{(A+BU_*)t} = \epsilon E(t) + \epsilon^2 E_1(\epsilon, t)$.

Now we upper bound $\|E(t)\|$ and $\|E_1(\epsilon, t)\|$. When $t \leq t_0$:

$$\|E(t)\| \leq \sum_{k \geq 1} \frac{t^k}{k!} \sum_{i=0}^{k-1} \left\| (A + BU_*)^i (B\Delta U_*)(A + BU_*)^{k-1-i} \right\| \leq \sum_{k \geq 1} \frac{1}{(k-1)!} = e$$

For $t \geq t_0$, let $t = 2^{l_1} t_1$, with $l_1$ be an integer and $t_1 \in (\frac{1}{2} t_0, t_0]$, then

$$\|E(2^{l_1} t_1)\| = \left\| e^{(A+BU_*)2^{l_1-1} t_1} E(t) + E(t) e^{(A+BU_*)2^{l_1-1} t_1} \right\|$$

$$\leq 2e^{\alpha(A+BU_*)2^{l_1-1} t_1} \left\| E(2^{l_1-1} t_1) \right\|$$

$$\leq 2^{l_1} e^{1 + \alpha(A+BU_*)2^{l_1-2} t_0}$$

$$\leq \frac{4}{-\alpha(A + BU_*)t_0},$$

where the last inequality is because for any $x, a > 0$, $xe^{-ax} \leq \frac{1}{ae}$, and thus for any $t \geq 0$, $\|E(t)\| \leq C = \frac{4}{-\alpha(A+BU_*)t_0}$.

When $t \geq \frac{2}{-\alpha(A+BU_*)}$, we additionally have

$$\|E(t)\| \leq 2e^{\frac{1}{2}\alpha(A+BU_*)t} \left\| E(\frac{t}{2}) \right\| \leq \frac{4t}{t_0} e^{\frac{1}{2}\alpha(A+BU_*)t} \leq \frac{8}{-\alpha(A + BU_*)t_0} e^{\frac{1}{4}\alpha(A+BU_*)t}.$$

Now we consider $E_1(\epsilon, t)$. When $t \leq t_0$,

$$\|E_1(\epsilon, t)\| \leq \sum_{k \geq 1} \left\| \frac{t^k}{k!} D_1(k, \epsilon) \right\| \leq 4.$$

When $t > t_0$, with $t = 2^l t_1$ and $t_1 \in (\frac{1}{2} t_0, t_0]$, we obtain:

$$\left\| E_1(\epsilon, 2^l t_1) \right\| =$$

$$\left\| e^{(A+BU_*)2^{l-1} t_1} E_1(\epsilon, 2^{l-1} t_1) + E_1(\epsilon, 2^{l-1} t_1) e^{(A+BU_*)2^{l-1} t_1} + \left( E(2^{l-1} t_1) + \epsilon E_1(\epsilon, 2^{l-1} t_1) \right)^2 \right\|$$

$$\leq 2e^{\alpha(A+BU_*)2^{l-1} t_1} \left\| E_1(\epsilon, 2^{l-1} t_1) \right\| + \left\| E(2^{l-1} t_1) + \epsilon E_1(\epsilon, 2^{l-1} t_1) \right\|^2$$

$$\leq 2e^{\alpha(A+BU_*)2^{l-1} t_1} \left\| E_1(\epsilon, 2^{l-1} t_1) \right\| + 2 \left\| E(2^{l-1} t_1) \right\|^2 + 2\epsilon^2 \left\| E_1(\epsilon, 2^{l-1} t_1) \right\|^2.$$

Now, we show that $\left\| E_1(\epsilon, 2^l t_1) \right\|$ converges exponentially eventually. The proof consists of two parts: first, for $t$ which is not too large, $\|E_1(\epsilon, t)\|$ can be bounded uniformly over all possible $\Delta U$ and any constrained $\epsilon$. Then, for larger $t$ we can utilize the construction of $\|E_1(\epsilon, t)\|$ to estimate its convergence speed.

Let $\epsilon \leq \frac{-\alpha(A+BU_*)t_0}{(64C)^2}$, $l_0 = 1 + \lfloor \log_2 \frac{4}{-\alpha(A+BU_*)t_0} \rfloor$. We first inductively show that for any $l \leq l_0$, $\left\| E_1(\epsilon, 2^l t_1) \right\| \leq (2^{l+3} - 4)C^2$. The base case where $l = 0$ is certainly true. Suppose we already have $\left\| E_1(\epsilon, 2^{l-1} t_1) \right\| \leq (2^{l+2} - 4)C^2$. Then for the case of $l$, we obtain:

$$\left\| E_1(\epsilon, 2^l t_1) \right\| \leq 2 \left\| E_1(\epsilon, 2^{l-1} t_1) \right\| + 4C^2 \leq (2^{l+3} - 4)C^2,$$

where for the first inequality we use the inductive hypothesis that

$$\epsilon \left\| E_1(\epsilon, 2^{l-1} t_1) \right\| \leq 2^{l_0+3} C^2 \epsilon \leq \frac{64}{-\alpha(A + BU_*)t_0} C^2 \epsilon \leq C,$$

along with facts that $\left\| E(2^{l-1} t_1) \right\| \leq C$ and $2e^{\alpha(A+BU_*)2^{l-1} t_1} \leq 2$. Specifically, we have $\left\| E_1(\epsilon, 2^{l_0} t_1) \right\| \leq \frac{64C^2}{-\alpha(A+BU_*)t_0}$.

Now, we consider $l > l_0$. We first show that for all such $l$, $\left\|E_1(\epsilon, 2^l t_1)\right\| \leq \frac{64C^2}{-\alpha(A+BU_*)t_0}$. Since $2^{l-1}t_1 \geq 2^{l_0-1}t_0 \geq \frac{2}{-\alpha(A+BU_*)}$, we have $2e^{\alpha(A+BU_*)2^l t_1} \leq 2e^{-2}$, and thus

$$\left\|E_1(\epsilon, 2^l t_1)\right\| \leq 2e^{\alpha(A+BU_*)2^{l-1}t_1}\left\|E_1(\epsilon, 2^{l-1}t_1)\right\| + 2\left\|E(2^{l-1}t_1)\right\|^2 + 2\epsilon^2\left\|E_1(\epsilon, 2^{l-1}t_1)\right\|^2$$
$$\leq 2e^{-2}\left\|E_1(\epsilon, 2^{l-1}t)\right\| + 4C^2$$
$$\leq \frac{64C^2}{-\alpha(A+BU_*)t_0},$$

which holds for all $l \geq l_0$ with induction on $l$. Now we reuse the above expression and obtain that

$$\left\|E_1(\epsilon, 2^l t_1)\right\|$$
$$\leq 2e^{\alpha(A+BU_*)2^{l-1}t_1}\left\|E_1(\epsilon, 2^{l-1}t_1)\right\| + 2\left\|E(2^{l-1}t_1)\right\|^2 + 2\epsilon^2\left\|E_1(\epsilon, 2^{l-1}t_1)\right\|^2$$
$$\leq 2e^{-2^{l-l_0}}\frac{64C^2}{-\alpha(A+BU_*)t_0} + \frac{128}{\alpha^2(A+BU_*)t_0^2}e^{-2^{l-l_0-1}} + 2\epsilon^2\left\|E_1(\epsilon, 2^{l-1}t_1)\right\|^2 .$$

Let $l_*$ be the smaller integer greater than $l_0 + 1$ which satisfies:

$$2e^{-2^{l_*-l_0}}\frac{64C^2}{-\alpha(A+BU_*)t_0} + \frac{128}{\alpha^2(A+BU_*)t_0^2}e^{-2^{l_*-l_0-1}} \leq \frac{1}{4} .$$

Then by using the relation that $2\epsilon^2\left\|E_1(\epsilon, 2^{l-1}t_1)\right\|^2 \leq 2\epsilon^2\left(\frac{64C^2}{-\alpha(A+BU_*)t_0}\right)^2 \leq \frac{1}{4}$, we have:

$$\left\|E_1(\epsilon, 2^{l_*}t_1)\right\| \leq \frac{1}{2} .$$

Now we inductively show that for all $k \geq 0$,

$$\left\|E_1(\epsilon, 2^{l_*+k}t_1)\right\| \leq 2^{-2^k} .$$

By using the hypothesis for $k$ and $2\epsilon^2 \leq \frac{1}{4}$, we obtain:

$$\left\|E_1(\epsilon, 2^{l_*+k+1}t_1)\right\| \leq 2\epsilon^2\left\|E_1(\epsilon, 2^{l_*+k}t_1)\right\|^2 + \frac{1}{4}e^{-2^{k+l_*-l_0}+2^{l_*-l_0}}$$
$$\leq \frac{1}{4}2^{-2^{k+1}} + \frac{1}{4}e^{-2^{k+2}+2^2}$$
$$\leq 2^{-2^{k+1}},$$

leading to the claim. This means there exist some constants $C_1, c_1 > 0$ depending on $\alpha(A + BU_*)$ such that for all $t \geq 0$, $\|E_1(\epsilon, t)\| \leq C_1 e^{-c_1 t}$.

Finally, we consider $\int_{t\geq 0} e^{(A+BU)^{\mathrm{T}}t}(Q + U^{\mathrm{T}}RU)e^{(A+BU)t}dt$. Since $e^{(A+BU_*+\epsilon\Delta U)t} = e^{(A+BU_*)t} + \epsilon E(t) + \epsilon^2 E_1(\epsilon, t)$, with $\|E(t)\| \leq \frac{8}{-\alpha(A+BU_*)t_0}e^{\frac{1}{4}\alpha(A+BU_*)t}$ and bounded $E_1(\epsilon, t)$, we obtain:

$$\int_{t\geq 0} e^{(A+BU)^{\mathrm{T}}t}(Q + U^{\mathrm{T}}RU)e^{(A+BU)t}dt$$
$$= \int_{t\geq 0}(e^{(A+BU_*)^{\mathrm{T}}t} + \epsilon E^{\mathrm{T}}(t) + \epsilon^2 E_1^{\mathrm{T}}(\epsilon, t))(Q + U^{\mathrm{T}}RU)(e^{(A+BU_*)t} + \epsilon E(t) + \epsilon^2 E_1(\epsilon, t))dt$$
$$= \int_{t\geq 0} e^{(A+BU_*)^{\mathrm{T}}t}(Q + U_*^{\mathrm{T}}RU_*)e^{(A+BU_*)t}dt$$
$$+ \epsilon \int_{t\geq 0} E^{\mathrm{T}}(t)(Q + U_*^{\mathrm{T}}RU_*)e^{(A+BU_*)t} + e^{(A+BU_*)^{\mathrm{T}}t}(Q + U_*^{\mathrm{T}}RU_*)E(t)dt$$
$$+ \epsilon \int_{t\geq 0} e^{(A+BU_*)^{\mathrm{T}}t}\left(\Delta U^{\mathrm{T}}RU_* + U_*^{\mathrm{T}}R\Delta U\right)e^{(A+BU_*)t}dt$$
$$+ \mathcal{O}(\epsilon^2) .$$

Where the last term $\mathcal{O}(\epsilon^2)$ contains any terms with order at least $\epsilon^2$, whose norm is at most $C_2\epsilon^2$ for any $\epsilon \in [0, \epsilon_0)$ and $\|\Delta U\| = 1$, where the constant $C_2$ depends on $A, B, \alpha(A + BU_*)$ and $\epsilon_0$ is some small constant.

For any $\|\Delta U\| = 1$, define $V$ by

$$
V = \int_{t \geq 0} E^{\mathrm{T}}(t)(Q + U_*^{\mathrm{T}} R U_*) e^{(A + BU_*)t} + e^{(A + BU_*)^{\mathrm{T}} t}(Q + U_*^{\mathrm{T}} RU)E(t)dt
$$

$$
+ \int_{t \geq 0} e^{(A + BU_*)^{\mathrm{T}} t} \left( \Delta U^{\mathrm{T}} R U_* + U^{\mathrm{T}} R \Delta U \right) e^{(A + BU_*)t} dt,
$$

then $cost(U) = cost(U_*) + \epsilon \mathrm{tr}(V) + O(\epsilon^2)$.

Since $U_*$ minimizes $cost(U)$, $\mathrm{tr}(V) = \lim_{\epsilon \to 0} \epsilon^{-1}(cost(U_* + \epsilon \Delta U) - cost(U_*)) = 0$. Therefore, we obtain that $cost(U) = cost(U_*) + \mathcal{O}(\epsilon^2)$.

$\qquad\square$

## B.5 Proof of Lemma 19

In this section, we proof Lemma 19.

**Lemma 19.** *Let $U_*$ follows the same definition as in Lemma 18. Then, for some $\epsilon > 0$, there exist constants $C_2$ and $C_3$ (independent of $U$) such that for all $T > 0$ and any $U$ such that $\|U - U_*\| \leq \epsilon$,*

$$
|J_T - cost(U)T| \leq C_2 \|x\|_2^2 + C_3 .
$$

*Here $J_T$ is the expected cost of the policy that takes action by $U_t = U X_t$ ($t \in [0, T]$), with initial state $X_0 = x$.*

*Proof.* By definition of $J_T$, we have:

$$
J_T = \mathbb{E} \left[ \int_{t=0}^{T} \left( X_t^{\mathrm{T}} Q X_t + U_t^{\mathrm{T}} R U_t \right) dt \right] = \mathbb{E} \left[ \int_{t=0}^{T} X_t^{\mathrm{T}} (Q + U^{\mathrm{T}} RU) X_t dt \right] .
$$

Since the state transits according to $dX_t = AX_t dt + BU X_t dt + dW_t$, we can derive the expression of $X_t$ by $X_t = e^{(A + BU)t} X_0 + \int_{s=0}^{t} e^{(A + BU)(t - s)} dW_s$. Then by utilizing this expression we obtain:

$$
\begin{aligned}
& \mathbb{E} \left[ X_t^{\mathrm{T}} (Q + U^{\mathrm{T}} RU) X_t \right] \\
&= (e^{(A + BU)t} X_0)^{\mathrm{T}} (Q + U^{\mathrm{T}} RU) e^{(A + BU)t} X_0 \\
&\quad + 2\mathbb{E} \left[ (e^{(A + BU)t} X_0)^{\mathrm{T}} (Q + U^{\mathrm{T}} RU) \left( \int_{s=0}^{t} e^{(A + BU)(t - s)} dW_s \right) \right] \\
&\quad + \mathbb{E} \left[ \left( \int_{s=0}^{t} e^{(A + BU)(t - s)} dW_s \right)^{\mathrm{T}} (Q + U^{\mathrm{T}} RU) \left( \int_{s=0}^{t} e^{(A + BU)(t - s)} dW_s \right) \right] \\
&= X_0^{\mathrm{T}} e^{(A + BU)^{\mathrm{T}} t} (Q + U^{\mathrm{T}} RU) e^{(A + BU)t} X_0 \\
&\quad + \mathrm{tr} \left( \int_{s=0}^{t} e^{(A + BU)^{\mathrm{T}} s} (Q + U^{\mathrm{T}} RU) e^{(A + BU)s} ds \right) \\
&= X_0^{\mathrm{T}} e^{(A + BU)^{\mathrm{T}} t} (Q + U^{\mathrm{T}} RU) e^{(A + BU)t} X_0 \\
&\quad + \int_{s=0}^{t} \mathrm{tr} \left( e^{(A + BU)^{\mathrm{T}} s} (Q + U^{\mathrm{T}} RU) e^{(A + BU)s} \right) ds .
\end{aligned}
$$

Then, the expected cost on a trajectory lasting for time $T$ can be computed as:

$$\mathbb{E}\left[\int_{t=0}^{T} X_t^{\mathrm{T}}(Q + U^{\mathrm{T}}RU)X_t dt\right]$$

$$= \int_{t=0}^{T} \mathbb{E}\left[X_t^{\mathrm{T}}(Q + U^{\mathrm{T}}RU)X_t\right] dt$$

$$= \int_{t=0}^{T} X_0^{\mathrm{T}} e^{(A+BU)^{\mathrm{T}}t}(Q + U^{\mathrm{T}}RU)e^{(A+BU)t} X_0 dt$$

$$+ \int_{t=0}^{T} (T-t)tr\left(e^{(A+BU)^{\mathrm{T}}t}(Q + U^{\mathrm{T}}RU)e^{(A+BU)t}\right) dt$$

$$= \int_{t=0}^{T} X_0^{\mathrm{T}} e^{(A+BU)^{\mathrm{T}}t}(Q + U^{\mathrm{T}}RU)e^{(A+BU)t} X_0 dt + cost(U)T$$

$$- \int_{t=0}^{T} tr\left(e^{(A+BU)^{\mathrm{T}}t}(Q + U^{\mathrm{T}}RU)e^{(A+BU)t}\right) t dt$$

$$- T \int_{t=T}^{+\infty} tr\left(e^{(A+BU)^{\mathrm{T}}t}(Q + U^{\mathrm{T}}RU)e^{(A+BU)t}\right) dt \,.$$

Here the first term satisfies

$$\left|\int_{t=0}^{T} X_0^{\mathrm{T}} e^{(A+BU)^{\mathrm{T}}t}(Q + U^{\mathrm{T}}RU)e^{(A+BU)t} X_0 dt\right| \leq \int_{t\geq 0} e^{2\alpha(A+BU)t} \|X_0\|_2^2 dt$$

$$\leq \frac{1}{-2\alpha(A+BU)} \|X_0\|_2^2 \,,$$

and the latter two integral terms can be bounded as follows.

$$\left|\int_{t=0}^{T} tr\left(e^{(A+BU)^{\mathrm{T}}t}(Q + U^{\mathrm{T}}RU)e^{(A+BU)t}\right) t dt\right|$$

$$\leq \int_{t\geq 0} d \cdot e^{2\alpha(A+BU)t} \|Q + U^{\mathrm{T}}RU\| t dt$$

$$\leq \frac{d\|Q + U^{\mathrm{T}}RU\|}{4\alpha^2(A+BU)} \,,$$

$$\left|T \int_{t=T}^{+\infty} tr\left(e^{(A+BU)^{\mathrm{T}}t}(Q + U^{\mathrm{T}}RU)e^{(A+BU)t}\right) dt\right|$$

$$\leq T \int_{t\geq T} d \cdot e^{2\alpha(A+BU)t} \|Q + U^{\mathrm{T}}RU\| dt$$

$$\leq \frac{Td\|Q + U^{\mathrm{T}}RU\|}{-2\alpha(A+BU)} e^{2\alpha(A+BU)T}$$

$$\leq \frac{d\|Q + U^{\mathrm{T}}RU\|}{4\alpha^2(A+BU)} \,.$$

Therefore, for $C_2 \geq -\frac{1}{2\alpha(A+BU)}$ and $C_3 \geq \frac{d\|Q+U^{\mathrm{T}}RU\|}{2\alpha^2(A+BU)}$, we have

$$|J_T - cost(U)T| \leq C_2 \|x\|_2^2 + C_3 \,.$$

$\square$

## B.6  Proof of Lemma 21

Finally, we prove Lemma 21. In this part we suppose $T \geq T_0$, where $T_0 \geq 1$ is a constant depending on some hidden constants and $\|X_0\|_2^2$.

**Lemma 21.** *regret Let $U_t$ be the action applied as in Algorithm 3. Then there exists a constant $C \in poly(\kappa, M, \mu^{-1}, |\alpha(A + BK)|^{-1}, |\alpha(A + BK_*)|^{-1})$ such that for sufficiently large $T$:*

$$\mathbb{E}\left[\int_{t=0}^{\sqrt{T}} \left(X_t^{\mathrm{T}} Q X_t + U_t^{\mathrm{T}} R U_t\right) dt\right] \leq C \cdot \sqrt{T},$$

$$\mathbb{E}\left[\int_{t=\sqrt{T}}^{T} \left(X_t^{\mathrm{T}} Q X_t + U_t^{\mathrm{T}} R U_t\right) dt\right] \leq C \cdot \sqrt{T} + J_T^*.$$

Define the following events where the stabilizing controller $K$ might ever be applied during the exploitation phase. Let $\mathcal{E}_1 = \left\{\|X_{\sqrt{T}}\|_2 \geq \frac{1}{2}T^{1/5}\right\}$, $\mathcal{E}_2 = \left\{\|X_t\|_2 \geq T^{1/5} \ for \ some \ t \in [\sqrt{T}, T]\right\}$, and $\mathcal{E}_3 = \left\{\|\bar{K} - K_*\| \leq \epsilon_3\right\}$, where $\epsilon_3 > 0$ depends on the constant $\epsilon_0$ in Lemma 18, which will be determined later. In this part, we again let $C_1, C_2$ be the same as in (32), and denote $C_3$ be the constant $C_1$ in Lemma 18. We firstly analyze these three events.

**Upper Bound of** $\mathbb{P}[\mathcal{E}_1]$  By Lemma 17, we can find some constant $C_0$ depending on $\|A\|, \|B\|, \|K\|, d, p, h$ such that

$$\mathbb{P}\left[\|X_{\sqrt{T}}\|_2 \geq C_0\sqrt{\log(2T/\delta)}\right] \leq \delta.$$

This is because we have the recursive function of $\{X_{kh}\}$ that

$$X_{(k+1)h} = e^{(A+BK)h}X_{kh} + \int_{t=0}^{h} e^{(A+BK)(h-t)}dW_{kh+t} + \int_{t=0}^{h} e^{(A+BK)(h-t)}u_k dt,$$

from which we can derive that

$$X_{kh}$$
$$= e^{(A+BK)kh}X_0 + \int_{t=0}^{kh} e^{(A+BK)(kh-t)}dW_t + \sum_{i=0}^{k-1} e^{(A+BK)(k-i-1)h}\left(\int_{t=0}^{h} e^{(A+BK)t}dt\right)u_i.$$

Then, for sufficiently large $T$, $\left\|e^{(A+BK)\sqrt{T}}X_0\right\|_2$ can be bounded by 1, and from the proof in Lemma 17 we can apply similar idea to upper bound the norm of the last two terms. So we can obtain the probability bound on $\|X_{\sqrt{T}}\|_2$.

By setting $\delta = 2T \cdot e^{-\frac{T^{1/5}}{4C_0^2}}$, we obtain that $\mathbb{P}[\mathcal{E}_1] \leq 2T \cdot e^{-\frac{T^{1/5}}{4C_0^2}}$.

**Upper Bound of** $\mathbb{P}[\mathcal{E}_3^C]$   By (31), we obtain that, for $\epsilon_3 \leq \frac{C_1\epsilon_1}{T^{1/8}4C_2^2}$, we have:

$$\mathbb{P}\left[\|\bar{K} - K_*\| \geq x\right] \leq e^{-\frac{T^{1/2}x^2}{4C_1^2C_2^2}} \ \forall x \leq \epsilon_3,$$

and we also have: $\mathbb{P}[\mathcal{E}_3^C] \leq e^{-\frac{T^{1/2}\epsilon_3^2}{4C_1^2C_2^2}}$.

By setting $\epsilon_3 = \frac{C_1\epsilon_1}{T^{1/8}4C_2^2}$, we have: $\mathbb{P}[\mathcal{E}_3^C] \leq e^{-\frac{T^{1/4}\epsilon_1^2}{64C_2^2}}$.

**Upper Bound of** $\mathbb{P}[\mathcal{E}_2]$   Consider any $\|X_{\sqrt{T}}\|_2 \leq \frac{1}{2}T^{1/5}$ and any $\|\bar{K} - K_*\| \leq \epsilon_3$, we claim that $\mathbb{P}\left[\mathcal{E}_2 \big| X_{\sqrt{T}}, \bar{K}\right] \leq e^{-\Omega(T^{1/5})}$.

As what have discussed in Lemma 15 (see the discussion about stable margin near (30)), such $\bar{K}$ satisfies $\alpha(A + B\bar{K}) \leq \frac{1}{2}\alpha(A + BK_*)$.

Then by Lemma 17 we can derive that, for some constant $C$,

$$\mathbb{P}\left[\sup_{t\in[\sqrt{T},T]}\|X_t\|_2 - \|X_{\sqrt{T}}\|_2 \le \frac{1}{2}T^{1/5}\right] \le CTe^{-\frac{T^{1/5}}{C}} \le e^{-\Omega(T^{1/5})}.$$

Therefore,

$$\begin{aligned}
\mathbb{P}\left[\mathcal{E}_2\right] &\le 1 - \mathbb{P}\left[\mathcal{E}_1^C \cap \mathcal{E}_3\right] + e^{-\Omega(T^{1/5})}\mathbb{P}\left[\mathcal{E}_1^C \cap \mathcal{E}_3\right] \\
&\le \mathbb{P}\left[\mathcal{E}_1\right] + \mathbb{P}\left[\mathcal{E}_3^C\right] + e^{-\Omega(T^{1/5})} \\
&\le e^{-\Omega(T^{1/5})}.
\end{aligned}$$

Now we come to estimate the expected cost of Algorithm 3, as well as bound the regret. We separately calculate the cost during the two phases.

**Cost During Exploration Phase**  For $(k+1)h \le \sqrt{T}$ and $t \in [0, h]$, we have:

$$X_{kh+t} = e^{(A+BK)t}X_{kh} + \int_{s=kh}^{kh+t} e^{(A+BK)(kh+t-s)}dW_s + \left(\int_{s=0}^{t} e^{(A+BK)s}ds\right)u_k.$$

Then

$$\begin{aligned}
\mathbb{E}\left[X_{kh+t}^{\mathrm{T}}QX_{kh+t} + U_{kh+t}^{\mathrm{T}}RU_{kh+t}\right] &= \mathbb{E}\left[X_{kh+t}^{\mathrm{T}}(Q + K^{\mathrm{T}}RK)X_{kh+t} + u_k^{\mathrm{T}}Ru_k\right] + 2\mathbb{E}\left[X_{kh+t}^{\mathrm{T}}K^{\mathrm{T}}Ru_k\right] \\
&\le \mathbb{E}\left[X_{kh+t}^{\mathrm{T}}(Q + K^{\mathrm{T}}RK)X_{kh+t} + u_k^{\mathrm{T}}Ru_k\right] \\
&\quad + 2\mathbb{E}\left[u_k^{\mathrm{T}}\left(\int_{s=0}^{t}e^{(A+BK)s}ds\right)^{\mathrm{T}}K^{\mathrm{T}}Ru_k\right],
\end{aligned}$$

where the inequality is because $u_k$ is independent of $X_{kh}$ and $W_s (s \in [kh, kh+t])$.

For the first term, we first upper bound $\mathbb{E}\left[\|X_{kh+t}\|_2^2\right]$.

Denote $w_{k,t} = \int_{s=kh}^{kh+t} e^{(A+BK)(kh+t-s)}dW_s + \left(\int_{s=0}^{t} e^{(A+BK)s}ds\right)u_k$, which is a Gaussian variable with zero mean and is independent of $X_{kh}$. Then

$$\begin{aligned}
\mathbb{E}\left[\|X_{kh+t}\|_2^2\right] &= \mathbb{E}\left[\left\|e^{(A+BK)t}X_{kh} + w_{k,t}\right\|_2^2\right] \\
&= \mathbb{E}\left[\left\|e^{(A+BK)t}X_{kh}\right\|_2^2\right] + \mathbb{E}\left[\|w_{k,t}\|_2^2\right] \\
&\le \mathbb{E}\left[\|X_{kh}\|_2^2\right] + \mathbb{E}\left[\|w_{k,t}\|_2^2\right].
\end{aligned}$$

For $\mathbb{E}\left[\|X_{kh}\|_2^2\right]$, since

$$X_{kh} = e^{(A+BK)h}X_0 + \int_{t=0}^{kh} e^{(A+BK)(kh-t)}dW_t + \sum_{i=0}^{k-1} e^{(A+BK)(k-i-1)h}\left(\int_{t=0}^{h} e^{(A+BK)t}dt\right)u_i.$$

We have

$$\begin{aligned}
\mathbb{E}\left[\|X_{kh}\|_2^2\right] &= \left\|e^{(A+BK)kh}X_0\right\|_2^2 + \mathbb{E}\left[\left\|\int_{t=0}^{kh} e^{(A+BK)(kh-t)}dW_t\right\|_2^2\right] \\
&\quad + \sum_{i=0}^{k-1}\mathbb{E}\left[\left\|e^{(A+BK)(k-i-1)h}\left(\int_{t=0}^{h} e^{(A+BK)t}dt\right)u_i\right\|_2^2\right] \\
&\le e^{2\alpha(A+BK)\cdot kh}\|X_0\|_2^2 + tr\left(\int_{t=0}^{kh} e^{(A+BK)t}e^{(A+BK)^{\mathrm{T}}t}dt\right) \\
&\quad + \sum_{i=0}^{k-1} tr\left(\left[e^{(A+BK)ih}\left(\int_{t=0}^{h} e^{(A+BK)t}dt\right)\right]\left[e^{(A+BK)ih}\left(\int_{t=0}^{h} e^{(A+BK)t}dt\right)\right]^{\mathrm{T}}\right)
\end{aligned}$$

Therefore, we have

$$\mathbb{E}\left[\|X_{kh}\|_2^2\right] \leq e^{2\alpha(A+BK)\cdot kh}\|X_0\|_2^2 + \int_{t=0}^{kh} d \cdot e^{2\alpha(A+BK)t}dt + \sum_{i=0}^{k-1} d \cdot e^{2\alpha(A+BK)ih} \cdot h^2$$

$$\leq e^{2\alpha(A+BK)\cdot kh}\|X_0\|_2^2 + \frac{d}{-2\alpha(A+BK)} + \frac{dh^2}{1 - e^{2\alpha(A+BK)h}}$$

$$\leq C_3 + e^{2\alpha(A+BK)\cdot kh}\|X_0\|_2^2.$$

Where $C_3$ is a constant depending on $\alpha(A+BK)$ and $d$.

For the second term $\mathbb{E}\left[\|w_{k,t}\|_2^2\right]$, can follow the same process of the above bound and obtain $\mathbb{E}\left[\|w_{k,t}\|_2^2\right] \leq C_3$. Therefore, $\mathbb{E}\left[\|X_{kh+t}\|_2^2\right] \leq 2C_3$.

Now we can upper bound $\mathbb{E}\left[X_{kh+t}^{\mathrm{T}}QX_{kh+t} + U_{kh+t}^{\mathrm{T}}RU_{kh+t}\right]$. We have

$$\mathbb{E}\left[X_{kh+t}^{\mathrm{T}}(Q + K^{\mathrm{T}}RK)X_{kh+t}\right] \leq \mathbb{E}\left[\|Q + K^{\mathrm{T}}RK\|\,\|X_{kh+t}\|_2^2\right] \leq \|Q + K^{\mathrm{T}}RK\|\,\mathbb{E}\left[\|X_{kh+t}\|_2^2\right],$$

We also have $\mathbb{E}\left[u_k^{\mathrm{T}}Ru_k\right] = tr(R)$ and the following inequality:

$$\mathbb{E}\left[u_k^{\mathrm{T}}\left(\int_{s=0}^t e^{(A+BK)s}ds\right)^{\mathrm{T}} K^{\mathrm{T}}Ru_k\right] \leq (d+p) \cdot \left\|\left(\int_{s=0}^t e^{(A+BK)s}ds\right)^{\mathrm{T}} K^{\mathrm{T}}R\right\| \leq (d+p)h\|KR\|.$$

We can conclude that there exists constant $C_4$ depending on $A, B, K, Q, R, d, p, h$ such that

$$\mathbb{E}\left[X_{kh+t}^{\mathrm{T}}QX_{kh+t} + U_{kh+t}^{\mathrm{T}}RU_{kh+t}\right] \leq C_4\left(1 + e^{2\alpha(A+BK)\cdot(kh+t)}\|X_0\|_2^2\right), \forall k, t.$$

Therefore, the cost during exploration phase can be bounded as

$$\mathbb{E}\left[\int_{t=0}^{\sqrt{T}} \left(X_{kh+t}^{\mathrm{T}}QX_{kh+t} + U_{kh+t}^{\mathrm{T}}RU_{kh+t}\right)dt\right] \leq C_4\left(\sqrt{T} + \frac{\|X_0\|_2^2}{-2\alpha(A+BK)}\right). \qquad (35)$$

**Cost During Exploitation Phase** We first concentrate on $\mathcal{E}_2$, which is the hardest event for the analysis of the cost. Consider the following two cases:

**Case 1:** $\|X_{\sqrt{T}}\|_2 \geq T^{1/5}$. In this case, the action is applied by $U_t = KX_t, t \in [\sqrt{T}, T]$.

**Case 2:** $\|X_{\sqrt{T}}\|_2 < T^{1/5}$. In this case, the trajectory is unfortunately controlled by a bad controller, and suffers from large risk of diverging.

We first consider **Case 1**. By (**??**) we can derive that

$$X_t = e^{(A+BK)(t-\sqrt{T})}X_{\sqrt{T}} + \int_{s=\sqrt{T}}^t e^{(A+BK)(t-s)}dW_s.$$

Then, we have:

$$\mathbb{E}\left[X_t^{\mathrm{T}}QX_t + U_t^{\mathrm{T}}RU_t\right]$$
$$= \mathbb{E}\left[X_t^{\mathrm{T}}(Q + K^{\mathrm{T}}RK)X_t\right]$$
$$\leq \|Q + K^{\mathrm{T}}RK\|\,\mathbb{E}\left[\|X_t\|_2^2\right]$$
$$\leq \|Q + K^{\mathrm{T}}RK\|\left[\|X_{\sqrt{T}}\|_2^2 + \int_{s=\sqrt{T}}^t tr\left(e^{(A+BK)(t-s)}e^{(A+BK)^{\mathrm{T}}(t-s)}\right)dt\right]$$
$$\leq \|Q + K^{\mathrm{T}}RK\|\left[\|X_{\sqrt{T}}\|_2^2 + \int_{s=\sqrt{T}}^t d \cdot e^{2\alpha(A+BK)(t-s)}dt\right].$$

Therefore, for some constants $C_5, C_6$, we have:

$$\mathbb{E}\left[X_t^{\mathrm{T}}QX_t + U_t^{\mathrm{T}}RU_t\right] \leq C_5\|X_{\sqrt{T}}\|_2^2 + C_6.$$

Now we consider **Case 2**. Let $t_0 = \inf_t \{ \|X_t\|_2 \geq T^{1/5}, t \geq \sqrt{T} \}$, then $\|X_{t_0}\|_2 = T^{1/5}$ almost surely.

For $t \in [\sqrt{T}, t_0]$, since we always have

$$\|U_t\|_2 \leq \max \left\{ \|K\|, \|R^{-1}B^{\mathrm{T}}P\| \right\} \|X_t\|_2 \leq \left( \|K\| + \|R^{-1}B^{\mathrm{T}}\| T^{1/5} \right) T^{1/5},$$

the cost satisfies:

$$X_t^{\mathrm{T}}QX_t + U_t^{\mathrm{T}}RU_t \leq C_7 T^{4/5}.$$

Where $C_7$ is a constant depending on $B, R, K, P$.

For $t \in [t_0, T]$, the trajectory $X_t$ satisfies

$$X_t = e^{(A+BK)(t-t_0)}X_{t_0} + \int_{s=t_0}^t e^{(A+BK)(t-s)}dW_s.$$

Similar to the analysis for **Case 1**, we have:

$$\mathbb{E}\left[ X_t^{\mathrm{T}}QX_t + U_t^{\mathrm{T}}RU_t \right] \leq C_5 T^{2/5} + C_6$$

Combining them, we can conclude that for some constant $C_8$, no matter whether $\mathcal{E}_2$ happens, we always have:

$$\mathbb{E}\left[ X_t^{\mathrm{T}}QX_t + U_t^{\mathrm{T}}RU_t \right] \leq C_8 \left[ T^{4/5} + \|X_{\sqrt{T}}\|_2^2 \right] \ \forall t \in [\sqrt{T}, T].$$

Now we establish the upper bound for the regret. Since

$$1 = 1_{\mathcal{E}_1^C \cap \mathcal{E}_3} + 1_{\mathcal{E}_1} + 1_{\mathcal{E}_1^C \cap \mathcal{E}_3^C}$$

Then we can rewrite $\mathbb{E}\left[ \int_{t=\sqrt{T}}^T \left( X_t^{\mathrm{T}}QX_t + U_t^{\mathrm{T}}RU_t \right) dt \right]$ as

$$\mathbb{E}\left[ \int_{t=\sqrt{T}}^T \left( X_t^{\mathrm{T}}QX_t + U_t^{\mathrm{T}}RU_t \right) dt \right]$$

$$= \mathbb{E}\left[ \int_{t=\sqrt{T}}^T \left( X_t^{\mathrm{T}}QX_t + U_t^{\mathrm{T}}RU_t \right) dt \cdot 1_{\mathcal{E}_1^C \cap \mathcal{E}_3} \right]$$

$$+ \mathbb{E}\left[ \int_{t=\sqrt{T}}^T \left( X_t^{\mathrm{T}}QX_t + U_t^{\mathrm{T}}RU_t \right) dt \cdot 1_{\mathcal{E}_1} \right]$$

$$+ \mathbb{E}\left[ \int_{t=\sqrt{T}}^T \left( X_t^{\mathrm{T}}QX_t + U_t^{\mathrm{T}}RU_t \right) dt \cdot 1_{\mathcal{E}_1^C \cap \mathcal{E}_3^C} \right].$$

For the first term, we can upper bound it by

$$\mathbb{E}\left[ \int_{t=\sqrt{T}}^T \left( X_t^{\mathrm{T}}QX_t + U_t^{\mathrm{T}}RU_t \right) dt \cdot 1_{\mathcal{E}_1^C \cap \mathcal{E}_3} \right]$$

$$\leq \mathbb{E}\left[ \int_{t=\sqrt{T}}^T \left( X_t^{\mathrm{T}}QX_t + U_t^{\mathrm{T}}RU_t \right) dt \cdot 1_{\mathcal{E}_1^C \cap \mathcal{E}_2^C \cap \mathcal{E}_3} \right]$$

$$+ \mathbb{E}\left[ \int_{t=\sqrt{T}}^T \left( X_t^{\mathrm{T}}QX_t + U_t^{\mathrm{T}}RU_t \right) dt \cdot 1_{\mathcal{E}_1^C \cap \mathcal{E}_2} \right]$$

$$\leq \mathbb{E}\left[ \left( cost \left( R^{-1}B^{\mathrm{T}}P \right) T + C_9 \|X_{\sqrt{T}}\|_2^2 \right) \cdot 1_{\mathcal{E}_1^C \cap \mathcal{E}_3} \right] + \mathbb{E}\left[ C_8 \left( T^{4/5} + \|X_{\sqrt{T}}\|_2^2 \right) \cdot 1_{\mathcal{E}_1^C \cap \mathcal{E}_2} \right]$$

$$\leq C_9 T^{2/5} + cost(R^{-1}B^{\mathrm{T}}P_*)T + C_{10}T\mathbb{E}\left[ \|\bar{K} - K_*\|^2 \cdot 1_{\mathcal{E}_3} \right] + 2C_8 T^{4/5} \cdot \mathbb{E}\left[ 1_{\mathcal{E}_1^C \cap \mathcal{E}_2} \right].$$

Here the first inequality is because $1_{\mathcal{E}_1^C \cap \mathcal{E}_3} = 1_{\mathcal{E}_1^C \cap \mathcal{E}_2^C \cap \mathcal{E}_3} + 1_{\mathcal{E}_1^C \cap \mathcal{E}_2 \cap \mathcal{E}_3}$ and $1_{\mathcal{E}_1^C \cap \mathcal{E}_2 \cap \mathcal{E}_3} \le 1_{\mathcal{E}_1^C \cap \mathcal{E}_3}$. For the second inequality, the first term is because we can assume a situation that we do not change the dynamic when $\mathcal{E}_2$ happens, and that will not make the expectation smaller. By applying the results of Lemma 18 and Lemma 19 we can get this term, where the constant $C_9$ is related to constants in these two lemmas. The last inequality is obtained from these two lemmas and the definitions of $\mathcal{E}_1, \mathcal{E}_2, \mathcal{E}_3$.

As for $\mathbb{E}\left[\|\bar{K} - K_*\|^2 \cdot 1_{\mathcal{E}_3}\right]$, we use the bound that

$$\mathbb{P}\left[\|\bar{K} - K_*\| \ge x\right] \le e^{-\frac{T^{1/2} x^2}{4 C_1^2 C_2^2}} \ \forall x \le \epsilon_3,$$

and compute that

$$\mathbb{E}\left[\|\bar{K} - K_*\|^2 \cdot 1_{\mathcal{E}_3}\right]$$
$$\le \int_{x=0}^{\epsilon_3^2} \mathbb{P}\left[\|\bar{K} - K_*\|^2 \ge x\right] \cdot dx$$
$$\le \int_{x \ge 0} e^{-\frac{T^{1/2} x}{4 C_1^2 C_2^2}} dx$$
$$= \frac{4 C_1^2 C_2^2}{T^{1/2}}.$$

For $\mathbb{E}\left[1_{\mathcal{E}_1^C \cap \mathcal{E}_2}\right]$, we directly have $\mathbb{E}\left[1_{\mathcal{E}_1^C \cap \mathcal{E}_2}\right] \le \mathbb{P}\left[\mathcal{E}_2\right] \le e^{-\Omega(T^{1/5})}$. Combining these results and Lemma 19 we obtain that for some constant $C$,

$$\mathbb{E}\left[\int_{t=\sqrt{T}}^{T} \left(X_t^\mathrm{T} Q X_t + U_t^\mathrm{T} R U_t\right) dt \cdot 1_{\mathcal{E}_1^C \cap \mathcal{E}_3}\right] \le J_{\theta_*, T} + C\sqrt{T}.$$

For the second term $\mathbb{E}\left[\int_{t=\sqrt{T}}^{T} \left(X_t^\mathrm{T} Q X_t + U_t^\mathrm{T} R U_t\right) dt \cdot 1_{\mathcal{E}_1}\right]$, given any $X_{\sqrt{T}}$, we always have

$$\mathbb{E}\left[X_t^\mathrm{T} Q X_t + U_t^\mathrm{T} R U_t\right] \le C_8 \left[T^{4/5} + \|X_{\sqrt{T}}\|_2^2\right] \ \forall t \in [\sqrt{T}, T].$$

So we can upper bound $\mathbb{E}\left[\int_{t=\sqrt{T}}^{T} \left(X_t^\mathrm{T} Q X_t + U_t^\mathrm{T} R U_t\right) dt \cdot 1_{\mathcal{E}_1}\right]$ by

$$\mathbb{E}\left[\int_{t=\sqrt{T}}^{T} \left(X_t^\mathrm{T} Q X_t + U_t^\mathrm{T} R U_t\right) dt \cdot 1_{\mathcal{E}_1}\right]$$
$$\le C_8 T^{9/5} \mathbb{P}[\mathcal{E}_1] + C_8 T \mathbb{E}\left[\|X_{\sqrt{T}}\|_2^2 \cdot 1_{\mathcal{E}_1}\right]$$
$$\le O(1) + C_8 T \mathbb{E}\left[\|X_{\sqrt{T}}\|_2^2 \cdot 1_{\mathcal{E}_1}\right],$$

where for the last inequality we apply the upper bound of $\mathbb{P}[\mathcal{E}_1]$ shown before.

For $\mathbb{E}\left[\|X_{\sqrt{T}}\|_2^2 \cdot 1_{\mathcal{E}_1}\right]$, we can apply Lemma 17 and obtain that for some constant $c > 0$, for any $x \ge \frac{1}{2} T^{1/5}$, we have

$$\mathbb{P}\left[\|X_{\sqrt{T}}\|_2 \ge x\right] \le e^{-cx^2}.$$

Thus we have:

$$T \mathbb{E}\left[\|X_{\sqrt{T}}\|_2^2 \cdot 1_{\mathcal{E}_1}\right]$$
$$\le \frac{1}{4} T^{7/5} \mathbb{P}\left[\|X_{\sqrt{T}}\|_2 \ge \frac{1}{2} T^{1/5}\right] + T \int_{x \ge \frac{1}{4} T^{2/5}} \mathbb{P}\left[\|X_{\sqrt{T}}\|_2^2 \ge x\right] dx$$
$$\le \mathcal{O}(1).$$

Therefore, we have $\mathbb{E}\left[\int_{t=\sqrt{T}}^{T} \left(X_t^\mathrm{T} Q X_t + U_t^\mathrm{T} R U_t\right) dt \cdot 1_{\mathcal{E}_1}\right] \le \mathcal{O}(1)$

Finally, for the last term $\mathbb{E}\left[\int_{t=\sqrt{T}}^{T}\left(X_t^{\mathrm{T}}QX_t + U_t^{\mathrm{T}}RU_t\right)dt \cdot 1_{\mathcal{E}_1^C \cap \mathcal{E}_3^C}\right]$, when condition on any $\|X_{\sqrt{T}}\|_2 \leq \frac{1}{2}T^{1/5}$, estimator $(\hat{A}, \hat{B})$ and $X_{t_0}$, where $t_0 = \inf_{t \geq \sqrt{T}}(\|X_t\|_2 \geq T^{1/5})$, we still have:

$$\mathbb{E}\left[X_t^{\mathrm{T}}QX_t + U_t^{\mathrm{T}}RU_t\right] \leq C_8\left[T^{4/5} + \|X_{\sqrt{T}}\|_2^2\right] \leq 2C_8 T^{4/5}, \forall t \in [\sqrt{T}, T].$$

So we can upper bound it by

$$\begin{aligned}
&\mathbb{E}\left[\int_{t=\sqrt{T}}^{T}\left(X_t^{\mathrm{T}}QX_t + U_t^{\mathrm{T}}RU_t\right)dt \cdot 1_{\mathcal{E}_1^C \cap \mathcal{E}_3^C}\right]\\
&\leq 2C_8 T^{9/5}\mathbb{P}\left[\mathcal{E}_1^C \cap \mathcal{E}_3^C\right]\\
&\leq 2C_8 T^{9/5}\mathbb{P}[\mathcal{E}_3^C]\\
&\leq \mathcal{O}(1).
\end{aligned}$$

Combining them we finally obtain Lemma 21.

