# OpenReview forum: "Finite Sample Analyses for Continuous-time Linear Systems: System Identification and Online Control"
_NeurIPS.cc/2025/Conference — NeurIPS 2025 poster_

### Official Review · Reviewer_ZczL · 2025-06-30

**Clarity:** 3
**Significance:** 3
**Originality:** 3
**Rating:** 5
**Confidence:** 3

**Summary:**

The paper studies continuous time linear dynamical systems and makes the following contributions. Firstly, for the system identification problem with time horizon $T$, and assuming that the system is stable, an algorithm is obtained that achieves an estimation error of  $O(1/\sqrt{T})$ with only $O(T)$ samples. This method is also shown to be optimal up to constant factors. When a stabilizer is not known in advance, it is shown that $N$ independent short trajectories can be used  to obtain an estimate with error $O(1/\sqrt{N})$. These results lead to an online continuous linear control algorithm which is shown to achieve $O(\sqrt{T})$ regret.

**Questions:**

1. I do not see the dependence on the dimensions of $A,B$ appearing in the results, is this hidden inside the constant terms? If so, then I think it would be more clear if this could be explicitly shown. For discrete-time autonomous linear systems (with B = 0), we know that $T$ needs to be of the order $poly(n)$ where n is the dimension of $A$. Is it the same  for continuous time systems?

2. The system identification results require A to be stable. For discrete time autonomous linear systems, recent results have shown that consistent identification is possible even when the spectral radius of A is only required to be less than or equal to 1. Is it possible to use those techniques here as well?

**Ethical Concerns:**

["NO or VERY MINOR ethics concerns only"]

**Final Justification:**

The authors have clarified all the technical questions I had related to some parts of the paper. I am satisfied with the response and believe the paper is a good theoretical contribution towards the problem of learning continuous-time linear dynamical systems

**Limitations:**

Yes these have been discussed towards the end

**Quality:**

3

**Strengths And Weaknesses:**

Pros
---------

1.  The paper is written well overall with a clearly defined problem, notation, and summary of existing works. The writing is clear for the most part, and makes it easy to follow the ideas.

2.  The technique used for the system identification part (Alg. 1) seems quite natural, but it’s still non-trivial to carry out the full error analysis. The results are overall substantial, as upper and lower bounds are provided for system identification, along with an algorithm for online control. To my knowledge, these results appear to be novel as most recent works have addressed discrete-time linear systems. I think the results would be of interest for researchers working in this domain.

Cons
---------

It would have been helpful to provide the full dependence on the problem parameters in the statements of the theorems/lemmas. For e.g., in Theorem 2, $T$ is required to be polynomial in the parameters, but it would be worth displaying the actual dependence (hiding only the absolute constants). This is the case in other statements as well.

I also have some other questions which are separately outlined below.

---

> ### Author Rebuttal · Authors · 2025-07-31
>
> We extend our gratitude to the reviewer for the constructive suggestions and questions. We address the reviewer's concerns belows:
>
> >Q1: It would have been helpful to provide the full dependence on the problem parameters in the statements of the theorems/lemmas. For e.g., in Theorem 2, $T$ is required to be polynomial in the parameters, but it would be worth displaying the actual dependence (hiding only the absolute constants). This is the case in other statements as well.
>
> **A1:**  We thank the reviewer for the helpful suggestion. The dependencies on problem parameters in our theorems and lemmas are indeed polynomial, all with degrees less than 5. In the current version, we presented simplified statements in the main text. In the next revision, we will revise the main theorems to explicitly include the parameter dependencies to improve clarity and completeness.
>
> >Q2: I do not see the dependence on the dimensions of $A, B$ appearing in the results, is this hidden inside the constant terms? If so, then I think it would be more clear if this could be explicitly shown. For discrete-time autonomous linear systems (with $B=0$), we know that $T$ needs to be of the order $\text{poly}(n)$ where $n$ is the dimension of $A$. Is it the same for continuous time systems?
>
> **A2:** We thank the reviewer for the question. Regarding the dependence on the dimensions of $A \in \mathbb{R}^{d \times d}$ and $B \in \mathbb{R}^{d \times p}$, we assume $p \geq d$ to ensure the existence of a stabilizing controller. The required sample length $T$ is not simply a low-degree polynomial in $d$ and $p$, but must satisfy $T \geq \mathcal{\Omega}((d + p) \cdot h)$, where $h$ is the sampling interval. This ensures that the system parameters can be reliably recovered. In practice, $T$ should also be sufficiently large to guarantee small estimation errors. We agree that making the dimensional dependence more explicit would improve clarity, and we will revise our manuscript in the next version accordingly.
>
>
> >Q3: The system identification results require $A$ to be stable. For discrete time autonomous linear systems, recent results have shown that consistent identification is possible even when the spectral radius of $A$ is only required to be less than or equal to $1$. Is it possible to use those techniques here as well?
>
>
> **A3:** We thank the reviewer for the insightful question. Our approach establishes a bridge between continuous-time and discrete-time linear systems, which allows certain techniques developed for discrete-time settings to be adapted to the continuous-time case.
>
> To clarify the relationship: in the continuous-time system $$dX_{t}=AX_{t}dt+Bu_{t}dt+dW_{t}.$$
>
> The sign of $\text{Re}(\lambda(A))$ corresponds to different regimes in discrete-time systems. Specifically:
>
> $\text{Re}(\lambda(A)) < 0$ corresponds to a discrete system with spectral radius strictly less than 1 (i.e., stable),
>
> $\text{Re}(\lambda(A)) = 0$ corresponds to marginal stability (spectral radius 1),
>
> $\text{Re}(\lambda(A)) > 0$ corresponds to instability (spectral radius greater than 1).
>
> In our analysis, the asymptotic convergence rate remains the same across these regimes, up to constant factors, provided the system remains controllable and observable. Thus, while we assume stability for simplicity and to align with standard assumptions in continuous-time identification, we believe our framework could be extended to accommodate marginally stable systems by incorporating techniques from recent discrete-time literature.
>
>
> Finally, we thank the reviewer once again for the effort in providing us with valuable suggestions. We will continue to provide clarifications if the reviewer has any further questions.

---

> > ### Comment · Reviewer_ZczL · 2025-08-03
> > **Read the rebuttal**
> >
> > Thanks for clarifying my questions, I am satisfied with the response and will keep my current score.

---

> > > ### Author Response · Authors · 2025-08-03
> > > **Thank you!**
> > >
> > > We thank the reviewer for acknowledging our work!

---

### Official Review · Reviewer_tEBy · 2025-06-30

**Clarity:** 3
**Significance:** 2
**Originality:** 3
**Rating:** 4
**Confidence:** 3

**Summary:**

**Setting**. This paper studies the system identification and online control problems for continuous-time linear systems. Specifically, the system dynamics are given by: $d X_t = A X_t d t + B U_t d t + d W_t$, where $X_t$ and $U_t$ represent the state and action respectively, and $A, B$ are unknown system parameters. The control objective is to minimize the following continuous-time loss function:

$$
J_T = \mathbb{E}\left[\int_{t=0}^T \left(X_t^{\mathrm{T}} Q X_t + U_t^{\mathrm{T}} R U_t\right) d t\right]
$$

where $Q, R$ are known. Since the system is unknown, the parameters $A, B$ must be estimated online from observed data, and actions must be selected based on these estimates to optimize the loss. The whole process is interactive and online, so the objective is rewritten as minimizing the regret $R_T = J_T - J_T^*$.

**Methods**. The overall strategy consists of two parts. First, the continuous-time system is transformed into a discrete linear system:

$x_{k+1} = A^{\prime} x_k + B^{\prime} u_k + w_k$.

Second, the remaining problem can be viewed as a bandit problem: on one hand, exploration is needed to estimate the parameters more accurately; on the other hand, we need to select optimal actions based on the estimated system.

Thus, an explore-then-commit strategy is adopted: during the initial phase, a known stable controller $K$ is used to sample data. Then, based on the collected data, the system is estimated, and the optimal actions are selected using the estimated system and the known loss function.

**Questions:**

- In the second figure of the Experiments section, the red line is flat by normalization, but was the regret also flat during the early exploratory phase? That seems a bit strange— the regret should increase rapidly at the beginning and then stabilize later, due to the exploration then commit.

**Ethical Concerns:**

["NO or VERY MINOR ethics concerns only"]

**Final Justification:**

My initial concern was that the paper assumes a known stable controller for an unknown system, whereas in practice, obtaining such a controller also requires learning and can be viewed as part of the exploration process. The theoretical analysis in the paper did not initially account for the impact of this exploration phase on the overall performance. However, the authors’ rebuttal clarified that the cost of this phase is essentially of order $\log(T)$, and thus does not affect the final regret bounds. As a result, my main concern has been addressed. Thus I increase my score.

**Limitations:**

As mentioned in the weaknesses, I have concerns about the assumptions and setup of this work. The proposed method relies on a known stable controller for an unknown system, which seems contradictory. In an unknown environment, obtaining a stable controller typically requires some form of exploration or trial-and-error. This paper essentially ignores the impact of that exploration phase on the overall finite-sample analysis. A direct question would be: if we include the sampling used for exploration, is it still possible to achieve sub-linear finite-sample regret? Or does the analysis only hold in the asymptotic sense?

**Quality:**

3

**Strengths And Weaknesses:**

**Strength**

- The theoretical development appears solid and the content is detailed.
- It provides finite-time theoretical guarantees for continuous-time linear systems, while most prior work in control theory typically focuses on asymptotic convergence.

**Weakness**

- Assumption 3 seems a bit questionable: if the system dynamics are unknown, how can one find a stabilizing controller $K$? This implicitly requires Assumption 1 — namely, that the system matrix $A$ is already stable — otherwise finding such a controller would be very difficult. Therefore, this assumption feels rather strong.
- Although Section 4.4 proposes a strategy to find a stabilizing controller, this procedure itself involves estimating $A, B$, which is also part of exploration. However, this is not clearly discussed or accounted for as part of the exploration strategy.
- While the paper emphasizes its focus on continuous-time systems and providing theoretical guarantees in that setting, the core methodology still relies on first discretizing the system and then applying techniques from discrete-time analysis. The approach is not purely continuous in nature.

---

> ### Author Rebuttal · Authors · 2025-07-31
>
> We express our gratitude to the reviewer for the insightful comments. We address the reviewer's questions in more details as follows:
>
>
> >Q1: Assumption 3 seems a bit questionable: if the system dynamics are unknown, how can one find a stabilizing controller $K$? This implicitly requires Assumption 1 --- namely, that the system matrix $A$ is already stable --- otherwise finding such a controller would be very difficult. Therefore, this assumption feels rather strong.
>
> **A1:** We appreciate the reviewer’s comment. As noted in our manuscript (see Subsection 4.4, line 229), assuming access to a known stabilizing controller is standard in control with a single trajectory, and also in previous works[1][2].
>
> Moreover, our analysis in Section4.4 is dedicated to the problem when such a controller is not available. One can first apply Algorithm 2 using multiple independent trajectories to estimate $A, B$. With enough samples, the estimation errors $\Vert\hat{A} - A\Vert$ and $\Vert\hat{B} - B\Vert$ can be made sufficiently small. Using these estimates, we can compute a stabilizing controller $\hat{K}$ for $(\hat{A}, \hat{B})$ such that $\text{Re}(\lambda(\hat{A} + \hat{B}\hat{K})) \leq c$ for some $c < 0$. When the estimation errors are small enough, $\hat{K}$ also stabilizes the true system $(A, B)$. Therefore, Assumption 3 is practically feasible and does not require $A$ to be stable.
>
>
>
>
> >Q2: Although Section 4.4 proposes a strategy to find a stabilizing controller, this procedure itself involves estimating $A, B$, which is also part of exploration. However, this is not clearly discussed or accounted for as part of the exploration strategy.
>
> **A2:** We thank the reviewer for the suggestion. As discussed in A1, the procedure in Section 4.4 offers an alternative way to obtain a stabilizing controller through exploration using multiple trajectories. We will add the explanation as the reviewer suggested.
>
> In our main analysis, we assume a known stabilizer, which is standard in the literature and allows for a direct comparison with prior works [1][2] under the same assumption. Importantly, under this assumption, our algorithm achieves significantly lower regret than existing methods, as demonstrated both theoretically and empirically.
>
> >Q3: While the paper emphasizes its focus on continuous-time systems and providing theoretical guarantees in that setting, the core methodology still relies on first discretizing the system and then applying techniques from discrete-time analysis. The approach is not purely continuous in nature.
>
> **A3:** We thank the reviewer for the question. As computation can be only done with discrete samples, our work aim to close the gap between a continous world and a implementable estimation algorithm using discretely sampled data.
>
> We also note that our method performs an exact transformation, not an approximation, unlike prior work. In existing approaches to continuous-time system identification [1][2], discretization is typically based on approximations (see Section 5.1 of [1]):
> $$X_{t+h} \approx (I + hA)X_t + hBU_t + (W_{t+h} - W_t). $$
> This introduces a discretization error between the approximated and true dynamics. The error term, given by $(e^{hA} - I)/h - A$, is of order $\mathcal{O}(h)$. To ensure the discretization error does not dominate, $h$ must be set as $\mathcal{O}(\frac{1}{\sqrt{T}})$, leading to a super-linear sampling complexity $m = T/h = \mathcal{\Omega}(T^{3/2})$, and thus higher computational cost (see discussion in Section 4.1 of our manuscript).
>
> In contrast, our method avoids such approximations by using an identity-based transformation (see Lemma 1), eliminating the discretization error entirely. As a result, the sampling interval $h = \frac{1}{15}\kappa_A$ depends only on system parameters, not on $T$. This allows the sampling complexity to scale linearly with $T$, offering substantial computational benefits.
>
> Moreover, our regret analysis in Section 5 is carried out entirely in the continuous-time domain, and the proof of Theorem 6 uses tools specifically designed for continuous systems.
>
>
>
>
> >Q4: In the second figure of the Experiments section, the red line is flat by normalization, but was the regret also flat during the early exploratory phase? That seems a bit strange— the regret should increase rapidly at the beginning and then stabilize later, due to the exploration then commit.
>
> **A4:** We thank the reviewer for the insightful comment. The reason the red curve appears flat is due to the structure of the regret, which consists of two main phases: exploration and exploitation. The exploration phase incurs a large, dominant regret of order $O(\sqrt{T})$. Crucially, this regret is accumulated over a very short period relative to the total time horizon $T$. On the plot, this brief but steep increase is visually compressed into what appears as a large, near-vertical jump at the very beginning. This initial jump establishes a high baseline for the cumulative regret. Consequently, the subsequent, much slower growth during the exploitation phase appears negligible in comparison, rendering the overall curve relatively flat.
>
>
> >Q5: The proposed method relies on a known stable controller for an unknown system, which seems contradictory. In an unknown environment, obtaining a stable controller typically requires some form of exploration or trial-and-error. This paper essentially ignores the impact of that exploration phase on the overall finite-sample analysis. If we include the sampling used for exploration, is it still possible to achieve sub-linear finite-sample regret?
>
> **A5:** We thank the reviewer for the thoughtful question. First, the assumption of a known stabilizing controller is made for simplicity. As discussed in our manuscript (particularly in Section 4.4), a stabilizing controller can also be obtained by estimating system parameters using multiple independent short trajectories. This procedure effectively serves as an exploration phase.
>
> We now evaluate the regret incurred during this exploration phase. From Theorem 5, we know that with $\mathcal{O}(\kappa_A^2 \kappa_B^2 (d + p) \gamma^{-2} \log T)$ independent trajectories, we can ensure, with probability at least $1 - T^{-10}$, that the estimation error $\Vert\hat{A} - A\Vert + \Vert\hat{B} - B\Vert$ is at most $\gamma$. By choosing $\gamma$ to be the stability margin, we can guarantee that the controller computed for $(\hat{A}, \hat{B})$ also stabilizes the true system $(A, B)$ with high probability.
>
> From Assumption 2 in our paper, we know that each trajectory has a $T' = \mathcal{O}(1)$ run time. From the state transition formula (line 428), we have
> $$
> X_{t}=\int_{s=0}^{t}e^{A(t-s)}(BU_{s}ds+dW_s) ,
> $$
> and it follows that $\mathbb{E}[\Vert X_{t}\Vert^{2}_{2}]\leq e^{2\Vert A\Vert T'}\cdot(1+\Vert B\Vert^{2})T'h$. Hence, the expected cost per trajectory is at most$$\mathcal{O}\left(e^{2\Vert A\Vert T'}\cdot (1+\Vert B\Vert^{2})(\Vert Q\Vert +\Vert R\Vert)T'\right),$$which is $\mathcal{O}(1)$ and independent from the total run time $T$ in regret analysis.
>
> Combining this with the number of required trajectories, we conclude that the total cost of the initial exploration phase is sublinear in $T$. Therefore, the overall regret, including this phase, remains sublinear. We thank the reviewer again for raising this important question and will revise the manuscript accordingly to include this analysis.
>
> We thank the reviewer once again for the valuable and helpful suggestions. We would be happy to provide further clarifications if the reviewer has any additional questions.
>
>
> **References**
>
> [1] Faradonbeh, Mohamad Kazem Shirani, and Mohamad Sadegh Shirani Faradonbeh. "Online Reinforcement Learning in Stochastic Continuous-Time Systems." The Thirty Sixth Annual Conference on Learning Theory. PMLR, 2023.
>
> [2] Shirani Faradonbeh, Mohamad Kazem, Mohamad Sadegh Shirani Faradonbeh, and Mohsen Bayati. "Thompson sampling efficiently learns to control diffusion processes." Advances in Neural Information Processing Systems 35 (2022): 3871-3884.

---

> > ### Comment · Reviewer_tEBy · 2025-08-04
> >
> > Thank you for the detailed response. My question has been resolved. I believe the discussion in A5 should be included in the revised version of the paper. I will increase my score.

---

> > > ### Author Response · Authors · 2025-08-05
> > > **Thank you!**
> > >
> > > We thank the reviewer for the insightful suggestion. We will carefully revise our manuscript and include the discussion in the new version of our paper.

---

### Official Review · Reviewer_nQK5 · 2025-07-01

**Clarity:** 2
**Significance:** 3
**Originality:** 3
**Rating:** 5
**Confidence:** 4

**Summary:**

This paper developed a system identification algorithm for continuous-time stochastic linear systems. The proposed method is based on time-discretization with a properly designed sampling interval. The estimation accuracy is guaranteed by an upper bound (Theorem 2). Moreover, based on the established result for the system identification, the authors devised an online control algorithm for continuous-time systems. Its performance is guaranteed by $O(\sqrt{T})$ regret upper bound (Theorem 6). The effectiveness of the proposed methods is verified through numerical experiments.

**Questions:**

- In Theorem 4, the system parameters $A$ and $B$ are allowed to depend on the time horizon $T$. In fact, in the proof sketch, the authors consider stable matrices $A$ and $\bar{A}$ satisfying $ |A - \bar{A}| = 2c_1 / \sqrt{T} $. However, the true system parameters are independent of $T$. Therefore, deriving a lower bound based on $A$ depending on $T$ appears to lack meaningful interpretation. If there is a valid reason why such a $T$-dependent construction of A is relevant or meaningful in the context of this work, it should be explained clearly and convincingly in the manuscript. Otherwise, a lower bound should be derived under the assumption that $A$ is independent of $T$.

- In Section 4.4, the authors consider the situation where a stabilizer is not known in advance. Then, they propose to first find a stable controller. However, Theorem 5 only gives the estimation error of system parameters and does not establish a result about stability. Therefore, it is unclear how Theorem 5 is useful. I suggest using Theorem 5 for finding a stabilizer for online control of unstable systems and establishing $O(\sqrt{T})$ regret bound without Assumption 3-1, which assumes knowledge of a stabilizer $K$. Then, the usefulness of Theorem 5 is clarified.

The usefulness of two theorems (Theorems 4, 5) is unclear in the current version.
If this issue is properly addressed, I would gladly raise my review score.

**Ethical Concerns:**

["NO or VERY MINOR ethics concerns only"]

**Final Justification:**

The authors have addressed my concerns. Therefore, I increase my score.

**Limitations:**

yes

**Paper Formatting Concerns:**

I do not have any paper formatting concerns.

**Quality:**

3

**Strengths And Weaknesses:**

### Strengths:
The proposed algorithm can attain an expected regret of $O(\sqrt{T})$ with only $O(T)$ samples in time horizon $T$.
Even when a stabilizing controller for an unstable, unknown system is not known in advance, the authors establish an upper bound of the estimation error by using multiple short-interval trajectories.


### Weaknesses:
I would like to point out that your paper adopts the pseudo regret in Theorem 6, while the existing studies [23,30,31] on continuous-time online control instead used the regret definition based on trajectory-wise comparisons, namely $$ \int_0^T [c(x_t,K_t x_t) - c(x_t^{\star}, K_{\star} x_t^{\star})] dt, $$
where $x_t^{\star}$ is the optimal trajectory and $c(x_t,u_t)$ denotes the cost function. The analysis of the regret is generally more challenging than that of the pseudo-regret as it requires accounting for trajectory deviations under the learned and optimal policies. Therefore, the fact that the proposed method achieves $O(\sqrt{T})$ upper bound of the pseudo-regret, does not imply that it surpasses the performance guarantees of the existing studies [23,30,31]. The authors should clarify this distinction and carefully position their results in relation to those earlier studies.

---

> ### Author Rebuttal · Authors · 2025-07-31
>
> We thank the reviewer for the comments and constructive suggestions. We address the reviewer's questions in more detail as follows:
>
> >Q1: This paper adopts the pseudo regret in Theorem 6, while the existing studies[1,2] on continuous-time online control instead used the regret definition based on trajectory-wise comparisons. The proposed method achieves $\mathcal{O}(\sqrt{T})$ upper bound of the pseudo-regret, does not imply that it surpasses the performance guarantees of the existing studies[1,2]. The authors should clarify this distinction and carefully position their results in relation to those earlier studies.
>
> **A1:** We appreciate the reviewer’s comment. This is a very helpful point and we will add the clarifications below in our updated manuscript.
>
>
> We first clarify that existing results [1,2] achieve an $\mathcal{O}(\sqrt{T} \log T)$ bound for both the exact regret and the pseudo-regret. The cost over $[0,T]$ is defined as: $$Cost_{T}=\int_{t=0}^{T}X_{t}^TQX_{t}+U_{t}^TRU_{t}dt ,$$
>
> which can be decomposed as:
>
> $$
> Cost_{T}=J_{T}+\int_{t=0}^{T}X_{t}^{T}QX_{t}+U_{t}^{T}RU_{t}-\mathbb{E}[X_{t}^{T}QX_{t}+U_{t}^{T}RU_{t}]dt ,
> $$
> where $J_{T}=\mathbb{E}\left[\int_{t=0}^{T}X_{t}^{T}QX_{t}+U_{t}^{T}RU_{t}dt\right]$ is the pseudo-cost used in our analysis.
>
> The total regret can then be written as: $R_{T}=(J_{T}-J_{T}^{*})+(Cost_{T}-J_{T})$,
> where the first term is the pseudo-regret and the second term captures the randomness due to noise.
>
> Prior works only obtained $\mathcal{O}(\sqrt{T} \log T)$ bounds for the pseudo-regret term $J_T - J_T^*$, while we improve this to $\mathcal{O}(\sqrt{T})$. The second term, $\text{Cost}_T - J_T$, has zero mean and variance $\mathcal{O}(T)$ under any fixed stabilizing controller, and can be bounded by $\mathcal{O}(\sqrt{T})$ with high probability, or $\mathcal{O}(\sqrt{T} \log T)$ uniformly over $T$.
>
> We believe that the main analytical challenge lies in controlling the first term, and our method can be directly applied to derive a bound on the total regret. In contrast, the second term arises from the inherent randomness of the noise process and is unavoidable. Therefore, our improvement in the pseudo-regret term represents an improvement over previous methods under the same regret decomposition framework.
>
> >Q2: In Theorem 4, the system parameters $A$ and $B$ are allowed to depend on the time horizon $T$. In fact, in the proof sketch, the authors consider stable matrices $\bar{A}$ and $\bar{A}$ satisfying $|\bar{A} - A| = 2c_1 / \sqrt{T}$. However, the true system parameters are independent of $T$. Therefore, deriving a lower bound based on $A$ depending on $T$ appears to lack meaningful interpretation.
>
> **A2:** We thank the reviewer for the comment. Our upper bound shows that for fixed $T$, any transition matrix $A$ with bounded norm can be estimated with error $\mathcal{O}(1 / \sqrt{T})$. The lower bound establishes that for fixed $T$, there exists some $A$ satisfying the assumptions for which no algorithm can achieve estimation error better than $\mathcal{\Omega}(1 / \sqrt{T})$. The purpose of Theorem 4 is to demonstrate that our method is near-optimal, and allowing the adversarial choice of $(A, B)$ to depend on $T$ does not break the above message.
>
>
> >Q3: In Section 4.4, the authors consider the situation where a stabilizer is not known in advance. Then, they propose to first find a stable controller. However, Theorem 5 only gives the estimation error of system parameters and does not establish a result about stability. Therefore, it is unclear how Theorem 5 is useful.
>
> **A3:** We thank the reviewer for the helpful suggestion and we will add the following details in our updated manuscript. When a stabilizer is not known in advance, we can first apply Algorithm 2 using multiple independent trajectories to estimate $\hat{A}, \hat{B}$. With enough samples, the estimation errors $\Vert\hat{A} - A\Vert$ and $\Vert\hat{B} - B\Vert$ can be made sufficiently small. Based on these estimates, we can compute a controller $\hat{K}$ that stabilizes $(\hat{A}, \hat{B})$, i.e., $\text{Re}(\lambda(\hat{A} + \hat{B}\hat{K})) \leq c$ for some constant $c < 0$. When the estimation errors are small enough, $\hat{K}$ will also stabilize the true system $(A, B)$. Therefore, Algorithm 2 can be used to construct a stabilizing controller. We appreciate the reviewer’s suggestion and will clarify in the next version of our manuscript how Theorem 5 supports the construction of a stabilizer.
>
>
> We thank the reviewer once again for the valuable and helpful suggestions. We would be happy to provide further clarifications if the reviewer has any additional questions.
>
>
> **References**
>
> [1] Faradonbeh, Mohamad Kazem Shirani, and Mohamad Sadegh Shirani Faradonbeh. "Online Reinforcement Learning in Stochastic Continuous-Time Systems." The Thirty Sixth Annual Conference on Learning Theory. PMLR, 2023.
>
> [2] Shirani Faradonbeh, Mohamad Kazem, Mohamad Sadegh Shirani Faradonbeh, and Mohsen Bayati. "Thompson sampling efficiently learns to control diffusion processes." Advances in Neural Information Processing Systems 35 (2022): 3871-3884.

---

> > ### Comment · Reviewer_nQK5 · 2025-08-07
> >
> > Thank you for the clarification. However, I still have a concern regarding the interpretation of the lower bound in Theorem 4. Specifically, since the constant $ C $ in the upper bound (Theorem 2) depends on $ A $, allowing the construction of $A = A(T)$ depending on the time horizon $ T $ in the lower bound (Theorem 4) makes it unclear whether the upper and lower bounds are actually comparable. In addition, it is not clear from the statement of Theorem 4 whether the constants $ c_1 $ and $ c_2 $ in the lower bound depend on $ T $. Clarifying or addressing this point is essential to justify the claimed optimality of the proposed algorithm.

---

> > > ### Author Response · Authors · 2025-08-07
> > >
> > > We sincerely thank the reviewer for raising this important question. We acknowledge that in the construction of the lower bound, the matrix $A$ is allowed to depend on the time horizon $T$. However, as shown in Appendix A.6, the constructed matrices $A$ and $\bar{A}$ are given by $-I_d$ and $-I_d - U$, respectively, where $U$ has only one nonzero entry at position (1,1), equal to $\frac{1}{5\sqrt{T}}$, and zeros elsewhere.
> > >
> > > From this, we can compute the key constants involved in the upper bound:
> > >
> > > - The inverse stability margin $\frac{1}{|\alpha(A)|}$ equals 1 for $A = -I_d$, and is at most $\frac{1}{1 + \frac{1}{5\sqrt{T}}} \leq 1$ for $\bar{A}$.
> > > - The condition number $\kappa(A)$ equals 1 for $A = -I_d$, and is at most $1 + \frac{1}{5\sqrt{T}} \leq 2$ for $\bar{A}$.
> > >
> > > Both quantities are therefore upper bounded by universal constants, independent of $T$. This implies that the constant $C$ in Theorem 2 (the upper bound) does not depend on $T$ when applied to these instances, making the upper and lower bounds comparable in a meaningful way. In this regime, the gap between the lower and upper bounds remains only up to a constant factor.
> > >
> > > We will revise the presentation of Theorem 4 in the next version of the manuscript to make the setup and assumptions clearer, emphasize that under the mild restrictions $\frac{1}{|\alpha(A)|} \leq 1$ and $\kappa(A) \leq 2$, we are still able to construct matrices $A$ satisfying the conditions required for the lower bound. This ensures that the dependence on $A$ in the upper bound can be uniformly bounded by constants, thus maintaining a meaningful comparison between the upper and lower bounds.
> > >
> > > Moreover, regarding the constants $c_1$ and $c_2$ in Theorem 4, we clarify that both are absolute constants independent of $T$, as shown in Appendix A.6: $c_1 = \frac{1}{10}$ and $c_2 = \frac{1}{4e^3}$.
> > >
> > > We greatly appreciate the reviewer for pointing out that the statement of Theorem 4 could be made clearer. In the next revision of the manuscript, we will revise this part to explicitly clarify the independence of these constants from $T$ and add a discussion comparing the upper and lower bounds more directly.

---

> > > > ### Comment · Reviewer_nQK5 · 2025-08-07
> > > >
> > > > Thank you for the response. The authors have addressed my concerns, and I will increase my score.

---

> > > > > ### Author Response · Authors · 2025-08-07
> > > > > **Thank you!**
> > > > >
> > > > > We thank the reviewer for the insightful questions. We will carefully revise our manuscript and include the discussion in the new version of our paper.

---

### Official Review · Reviewer_YKfn · 2025-07-02

**Clarity:** 3
**Significance:** 3
**Originality:** 3
**Rating:** 5
**Confidence:** 2

**Summary:**

This paper develops non-asymptotic system-identification algorithms for continuous-time linear dynamical systems using only a finite number of noisy state observations. By discretizing at carefully chosen intervals and directly estimating the matrix exponential, the authors recover the continuous-time parameters without incurring discretization bias, achieving an estimation error of $O(1/\sqrt{T})$. They also prove a matching lower bound showing this rate is optimal up to constants. They also apply the above algorithm to online control regret analysis for continuous-time linear system.

**Questions:**

My main concern is the linear model specification. As the authors note in Section 6, many practical models are nonlinear. To what extent can the results in this paper be extended to nonlinear settings? It would also be useful to run robustness checks to assess the proposed method’s performance under nonlinear models.

The simulation study is limited. Adding simulations for higher-dimensional systems would help evaluate the stability of the proposed method.

How should one choose the sampling interval $h$? In theory, it can be set according to Assumption 1, but $\kappa_A$ is unknown in practice. Is there a data-driven procedure for selecting $h$?

Minor typo

Page 6, line 208: “we proof Theorem …”.

**Ethical Concerns:**

["NO or VERY MINOR ethics concerns only"]

**Final Justification:**

The authors have successfully addressed my concerns. Thus I increase my score.

**Limitations:**

See the weaknesses and questions for details.

**Paper Formatting Concerns:**

No formatting concerns.

**Quality:**

3

**Strengths And Weaknesses:**

Strengths

The paper provides the provable error bound for continuous-time system identification, which also matches information-theoretic lower bound. It applies the identification scheme to online LQR, achieving $O(\sqrt{T})$ regret.

Weaknesses

Experiments are restricted to small, simulated 3-dimensional systems; the performance and numerical stability of the method in higher dimensions or on real-world data remain untested. It would be helpful to comment/provide further simulations in higher dimensional setups.

Extension to adversarial or non-Gaussian noise: As mentioned in the paper, the analysis assumes stochastic (Brownian) disturbances; it is unclear how the method performs under adversarial or heavy-tailed noise models.

---

> ### Author Rebuttal · Authors · 2025-07-31
>
> We sincerely appreciate the reviewer’s insightful comments and suggestions. The minor typos have been corrected in the revised manuscript. Below, we provide responses to the main questions.
>
> >Q1: It would be helpful to provide further simulations in higher dimensional setups.
>
> **A1:** We appreciate the reviewer’s feedback. To address this, we conduct an experiment in a higher-dimensional setting with $d = p = 10$. Each entry of $A$ is sampled uniformly from $[-1, 1]$. The matrices $B$, $Q$, and $R$ are set as identity matrices.
>
> Since figures are not permitted in the rebuttal, we present the estimation errors $\Vert\hat{A} - A\Vert_F^2$, $\Vert\hat{B} - B\Vert_F^2$, and the normalized regret $R(T)/T^{1/2}$ in the tables below.
>
> |Running Time|Estimation Error of $A$|Estimation Error of $B$|
> |:----:|:----:|:----:|
> |5|2.80|1.06|
> |10|1.99|0.67|
> |15|1.62|0.53|
> |20|1.40|0.45|
> |25|1.25|0.40|
> |30|1.14|0.37|
> |35|1.07|0.34|
> |40|1.00|0.32|
> |45|0.91|0.29|
> |50|0.85|0.27|
>
> |Running Time|Normalized Regret of Our Algorithm|Normalized Regret of Baseline Algorithm([6])|
> |:----:|:----:|----:|
> |500|61.7|18.5|
> |1000|70.9|28.5|
> |1500|65.6|43.9|
> |2000|64.7|53.5|
> |2500|65.8|51.3|
> |3000|62.4|66.4|
> |3500|59.9|71.8|
> |4000|57.1|79.2|
>
> These results show that in high-dimensional settings, our algorithm achieves low estimation error and $\mathcal{O}(\sqrt{T})$ regret, outperforming the baseline algorithm which achieves $\mathcal{O}(\sqrt{T}\log T)$ regret when $T$ is large.
>
> >Q2: It is unclear how the method performs under adversarial noise models.
>
> **A2:** We thank the reviewer for this insightful question. While many online control works focus on adversarial noise [1,2,3,4], our research addresses stochastic noise. However, only stochastic noise models readily extend to continuous-time systems [5,6,7] (e.g., by modeling Gaussian noise as Brownian motion). Currently, a formal definition of adversarial noise in continuous systems is lacking, which precludes such an analysis in the continuous setting. We agree that extending adversarial noise setup to continuous-time systems is an important future direction.
>
> >Q3: It would also be useful to run robustness checks to assess the proposed method’s performance under nonlinear models.
>
> **A3:** We appreciate the reviewer’s suggestion. The key difference between linear and nonlinear models lies in the transition dynamics: linear systems admit explicit solutions, while general nonlinear transitions typically cannot be computed in closed form and must be approximated.
>
> The main contribution of our method compared to known ones is estimating the underlying state transition rule from discretized samples. This may still help reduce discretization bias when applied to mildly nonlinear systems. However, the accuracy then depends heavily on the convergence rate of this approximation, which is generally hard to quantify for nonlinear systems. Extending our method to nonlinear settings is an important direction and we leave it for future work.
>
> >Q4: How should one choose the sampling interval $h$? In theory, it can be set according to Assumption 1, but $\kappa_A$ is unknown in practice. Is there a data-driven procedure for selecting $h$?
>
> **A4:** We thank the reviewer for the insightful question. In practice, $\kappa_A$ can often be estimated based on physical insights or prior system knowledge—an assumption commonly made in system identification. Prior works on continuous-time identification also adopt assumptions on system parameters. For example, [5] assumes H.1, and [6] assumes the initial estimates $(A,B)$ are within specific error bounds of the true parameters $(A*, B*)$ (see Equation 7).
>
> When $\kappa_A$ is unknown, a conservative sampling interval $h$ can be used. In our experiments, we set $h = \frac{1}{30}$, which, while not necessarily optimal, guarantees convergence for both single and multiple trajectories. Different choices of $h$ affect only the constant factor, not the order of $T$. We clarify that $\kappa_A$ in Assumption 1 acts as a theoretical upper bound, and choosing a sufficiently small $h$ ensures $h < \frac{1}{15\kappa_A}$ is satisfied.
>
>
> Finally, we thank the reviewer once again for the effort in providing us with valuable and helpful suggestions. We will continue to provide clarifications if the reviewer has any further questions.
>
> **References**
>
> [1] Agarwal, Naman, et al. "Online control with adversarial disturbances." International Conference on Machine Learning. PMLR, 2019.
>
> [2] Hazan, Elad, Sham Kakade, and Karan Singh. "The nonstochastic control problem." Algorithmic Learning Theory. PMLR, 2020.
>
> [3] Simchowitz, Max, Karan Singh, and Elad Hazan. "Improper learning for non-stochastic control." Conference on Learning Theory. PMLR, 2020.
>
> [4] Agarwal, Naman, Elad Hazan, and Karan Singh. "Logarithmic regret for online control." Advances in Neural Information Processing Systems 32 (2019).
>
> [5] Basei, Matteo, et al. "Logarithmic regret for episodic continuous-time linear-quadratic reinforcement learning over a finite-time horizon." Journal of Machine Learning Research 23.178 (2022): 1-34.
>
> [6] Faradonbeh, Mohamad Kazem Shirani, and Mohamad Sadegh Shirani Faradonbeh. "Online Reinforcement Learning in Stochastic Continuous-Time Systems." The Thirty Sixth Annual Conference on Learning Theory. PMLR, 2023.
>
> [7] Shirani Faradonbeh, Mohamad Kazem, Mohamad Sadegh Shirani Faradonbeh, and Mohsen Bayati. "Thompson sampling efficiently learns to control diffusion processes." Advances in Neural Information Processing Systems 35 (2022): 3871-3884.

---

> > ### Comment · Reviewer_YKfn · 2025-08-05
> >
> > Thanks for the authors response! They have addressed my concerns, and I will therefore increase my score.

---

> > > ### Author Response · Authors · 2025-08-05
> > > **Thank you!**
> > >
> > > We thank the reviewer for acknowledging our work!

---

### Comment · Area_Chair_7PkE · 2025-08-08

Dear reviewers,

Thanks for engaging with the authors in a timely manner. Pleased with the discussions on this submission.

---

### Decision · Program_Chairs · 2025-09-17

**Decision:**

Accept (poster)

**Comment:**

This paper considers control and identification of continuous time linear systems, which however may only be observed at discrete points in time. Here, one must decouple the number of observations allowed (N) from horizon (T). The first result is that the lower bound on the estimation error scales as 1/T^1/2 even if the number of observations is allowed to scale arbitrarily. This is largely a comment about the sampling model/oracle being appropriate. With nonstandard sampling models, some past results can learn a system from arbitrary T by increasing N. The work also provides a 1/T^1/2 accurate estimator of the system matrices given N=O(T) observations. Here the main novelty is correctly inverting the discretization of induced discrete-time linear system, whereas the tradition first-order approximation results in N=T^3/2. The control result is relatively less impressive, improving upon past work by logarithmic factors, by essentially a consequence of the estimation result.

This paper has unanimous support from all reviewers. So we are happy to recommend to accept this paper.